# Uncertainty-Aware Instance Reweighting for Off-Policy Learning

Xiaoying Zhang[1]    Junpu Chen[2]    Hongning Wang[3]    Hong Xie[4]    Yang Liu[1]
John C.S. Lui[5]    Hang Li[1]

[1]ByteDance Research    [2]ChongQing University    [3]Tsinghua University
[4] Chongqing Institute of Green and Intelligent Technology, Chinese Academy of Science
[5] The Chinese University of Hong Kong
{zhangxiaoying.xy,yang.liu01,lihang.lh}@bytedance.com
{jumpchan98,hongx87,wang.hongn}@gmail.com
cslui@cse.cuhk.edu.hk

## Abstract

Off-policy learning, referring to the procedure of policy optimization with access only to logged feedback data, has shown importance in various real-world applications, such as search engines and recommender systems. While the ground-truth logging policy is usually unknown, previous work simply employs its estimated value for the off-policy learning, ignoring the negative impact from both high bias and high variance resulted from such an estimator. And such impact is often magnified on samples with small and inaccurately estimated logging probabilities. The contribution of this work is to explicitly model the uncertainty in the estimated logging policy, and propose an Uncertainty-aware Inverse Propensity Score estimator (UIPS) for improved off-policy learning, with a theoretical convergence guarantee. Experiment results on the synthetic and real-world recommendation datasets demonstrate that UIPS significantly improves the quality of the discovered policy, when compared against an extensive list of state-of-the-art baselines.

## 1   Introduction

In many real-world applications, including search engines [2], online advertisements [35], recommender systems [8, 22], only logged data is available for subsequent policy learning. For example, in recommender systems, various complex recommendation policies are optimized over logged user interactions (e.g., clicks or stay time) with items recommended by previous recommendation policies (referred to as the *logging policy*) [51, 14]. However, such logged data is often known to be biased, since the feedback on items where the logging policy did not take is unknown. This inevitably distorts the evaluation and optimization of a new policy when it differs from the logging policy.

Off-policy learning [41, 27] thus emerges as a preferred way to learn an improved policy only from the logged data, by addressing the mismatch between the learning and logging policies. One of the most commonly used off-policy learning methods is the Inverse Propensity Scoring (IPS) [8, 25], which assigns per-sample importance weight (i.e., propensity score) to the training objective on the logged data, so as to get an unbiased optimization objective in expectation. The importance weight in IPS is the probability ratio of taking an action between the learning and logging policies.

Unfortunately, the ground-truth logging policy is oftentimes unavailable to the learner in practice, due to reasons like legacy issues, i.e., it was not recorded in the data. Additionally, in specific situations like the healthcare domain [28] or two-stage recommender systems [8], access to the ground-truth logging policy is not feasible. One common treatment by many previous studies [35, 22, 8, 24] is to first estimate the logging policy using a supervised learning method (e.g., logistic regression,

37th Conference on Neural Information Processing Systems (NeurIPS 2023).

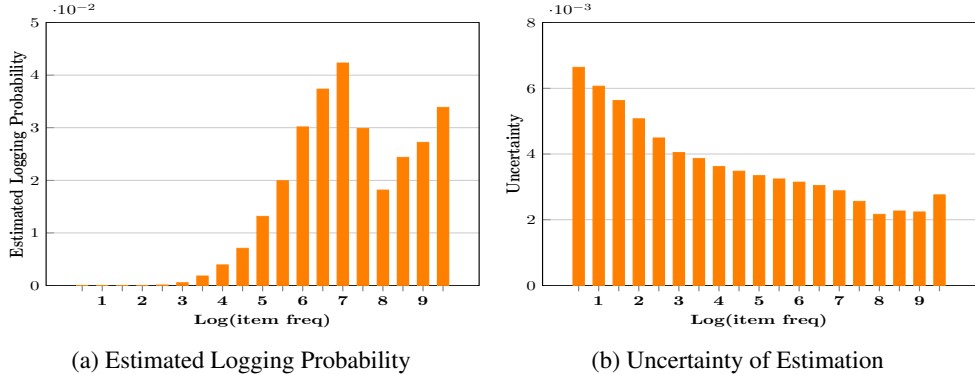

(a) Estimated Logging Probability      (b) Uncertainty of Estimation

Figure 1: Estimated logging policy and its uncertainty under different item frequency on KuaiRec.

neural networks, etc.), and then employ the estimated logging policy for off-policy learning. In this work, we first show that such an approximation results in a biased estimator which is sensitive to data with small estimated logging probabilities. Worse still, small estimated logging probabilities usually suggest there are limited related samples in the logged data, whose estimations can have high uncertainties, i.e., being wrong with a high probability. Figure 1 shows a piece of empirical evidence from a large-scale recommendation benchmark KuaiRec dataset [12], where items with lower frequencies in the logged dataset have lower estimated logging probabilities (via a neural network estimator) and higher uncertainties at the same time. The high bias and variance caused by these samples can greatly hinder the performance of subsequent off-policy learning. We defer detailed discussions of this result in Section 2.

In this work, we explicitly take the uncertainty of the estimated logging policy into consideration and design an Uncertainty-aware Inverse Propensity Score estimator (UIPS) for off-policy learning. UIPS reweighs the propensity score of each logged sample to control its impact on policy optimization, and learns an improved policy by alternating between: (1) Find the optimal weight that makes the estimator as accurate as possible, based on the uncertainty of the estimated logging policy; (2) Improve the policy by optimizing the resulting objective function. The optimal weight for each sample is obtained by minimizing the upper bound of the mean squared error (MSE) to the ground-truth policy evaluation, with a closed-form solution. Furthermore, UIPS ensures that off-policy learning converges to a stationary point where the true policy gradient is zero; while convergence may not be guaranteed when directly using the estimated logging policy. Extensive experiments on a synthetic and three real-world recommendation datasets against a rich set of state-of-the-art baselines demonstrate the power of UIPS. All data and code can be found in `https://github.com/Xiaoyinggit/UIPS.git`.

## 2   Preliminary: off-policy learning

We focus on the standard contextual bandit setup to explain the key concepts in UIPS. Following the convention [16, 29, 36], let $\boldsymbol{x} \in \mathcal{X} \subseteq R^d$ be a $d$-dimensional context vector drawn from an unknown distribution $p(\boldsymbol{x})$. Each context is associated with a finite set of actions denoted by $\mathcal{A}$, where $|\mathcal{A}| < \infty$. Let $\pi : \mathcal{A} \times \mathcal{X} \to [0, 1]$ denote a stochastic policy, such that $\pi(a|\boldsymbol{x})$ is the probability of selecting action $a$ under context $\boldsymbol{x}$ and $\sum_{a \in \mathcal{A}} \pi(a|\boldsymbol{x}) = 1$. Under a given context $\boldsymbol{x}$, the reward $r_{\boldsymbol{x},a}$ is only observed when action $a$ is chosen, i.e., bandit feedback. Without loss of generality, we assume $r_{\boldsymbol{x},a} \in [0, 1]$. Let $V(\pi)$ denote the expected reward of the policy $\pi$:

$$V(\pi) = \mathbb{E}_{\boldsymbol{x} \sim p(\boldsymbol{x}), a \sim \pi(a|\boldsymbol{x})}[r_{\boldsymbol{x},a}]. \tag{1}$$

We look for a policy $\pi(a|\boldsymbol{x})$ to maximize $V(\pi)$. In the rest, we denote $\mathbb{E}_{\boldsymbol{x} \sim p(\boldsymbol{x}), a \sim \pi(a|\boldsymbol{x})}[\cdot]$ as $\mathbb{E}_\pi[\cdot]$.

In off-policy learning, one can only access a set of logged feedback data $D := \{(\boldsymbol{x}_n, a_n, r_{\boldsymbol{x}_n,a_n}) | n \in [N]\}$. Given $\boldsymbol{x}_n$, the action $a_n$ was generated by a stochastic logging policy $\beta^*$, i.e., $a_n \sim \beta^*(a|\boldsymbol{x}_n)$, which is usually different from the learning policy $\pi(a|\boldsymbol{x})$ [24, 40, 8]. The actions $\{a_1, \ldots, a_N\}$ and their corresponding rewards $\{r_{\boldsymbol{x}_1,a_1}, \ldots, r_{\boldsymbol{x}_N,a_N}\}$ are generated independently given $\beta^*$. The main challenge is then to address the distributional discrepancy between $\beta^*(a|\boldsymbol{x})$ and $\pi(a|\boldsymbol{x})$, when optimizing $\pi(a|\boldsymbol{x})$ to maximize $V(\pi)$ with access only to the logged dataset $D$.

One of the most widely used methods to address the distribution shift between $\pi(a|\boldsymbol{x})$ and $\beta^*(a|\boldsymbol{x})$ is the Inverse Propensity Scoring (IPS) [8, 25]. One can easily get that:

$$V(\pi) = \mathbb{E}_{\beta^*}\left[\frac{\pi(a|\boldsymbol{x})}{\beta^*(a|\boldsymbol{x})}r_{\boldsymbol{x},a}\right],$$

yielding the following empirical estimator of $V(\pi)$:

$$\hat{V}_{\text{IPS}}(\pi) = \frac{1}{N}\sum_{n=1}^{N}\frac{\pi(a_n|\boldsymbol{x}_n)}{\beta^*(a_n|\boldsymbol{x}_n)}r_{\boldsymbol{x}_n,a_n}, \tag{2}$$

where $\pi(a_n|\boldsymbol{x}_n)/\beta^*(a_n|\boldsymbol{x}_n)$ is referred to as the propensity score. Various algorithms can be readily used for policy optimization under $\hat{V}_{\text{IPS}}(\pi)$, including value-based methods [33] and policy-based methods [19, 31, 42]. In this work, we adopt a well-known policy gradient algorithm, REINFORCE [42]. Assume the policy $\pi(a|\boldsymbol{x})$ is parameterized by $\boldsymbol{\vartheta}$, via the "log-trick", the gradient of $\hat{V}_{\text{IPS}}(\pi_{\boldsymbol{\vartheta}})$ with respect to $\boldsymbol{\vartheta}$ can be readily derived as,

$$\nabla_{\boldsymbol{\vartheta}}\hat{V}_{\text{IPS}}(\pi_{\boldsymbol{\vartheta}}) = \frac{1}{N}\sum_{n=1}^{N}\frac{\pi_{\boldsymbol{\vartheta}}(a_n|\boldsymbol{x}_n)}{\beta^*(a_n|\boldsymbol{x}_n)}r_{\boldsymbol{x}_n,a_n}\nabla_{\boldsymbol{\vartheta}}\log(\pi_{\boldsymbol{\vartheta}}(a_n|\boldsymbol{x}_n)).$$

**Approximation with an unknown logging policy**. In many real-world applications, the ground-truth logging probabilities, i.e., $\beta^*(a|\boldsymbol{x})$ of each observation $(\boldsymbol{x}, a)$ in $D$, are unknown. As a typical walk-around, previous work employs supervised learning methods such as logistic regression [30] and nerural networks [8] to estimate the logging policy, and replaces $\beta^*(a|\boldsymbol{x})$ with its estimated value $\hat{\beta}(a|\boldsymbol{x})$ to get the following BIPS estimator for policy learning:

$$\hat{V}_{\text{BIPS}}(\pi_{\boldsymbol{\vartheta}}) = \frac{1}{N}\sum_{n=1}^{N}\frac{\pi_{\boldsymbol{\vartheta}}(a_n|\boldsymbol{x}_n)}{\hat{\beta}(a_n|\boldsymbol{x}_n)}r_{\boldsymbol{x}_n,a_n}. \tag{3}$$

However, as shown in the following proposition, inaccurate $\hat{\beta}(a|\boldsymbol{x})$ leads to high bias and variance in BIPS. Worse still, smaller and inaccurate $\hat{\beta}(a|\boldsymbol{x})$ further enlarges this bias and variance.

**Proposition 2.1.** *The bias and variance of $\hat{V}_{\text{BIPS}}(\pi_{\boldsymbol{\vartheta}})$ can be derived as follows:*

$$\text{Bias}\left(\hat{V}_{\text{BIPS}}(\pi_{\boldsymbol{\vartheta}})\right) = \mathbb{E}_D\left[\hat{V}_{\text{BIPS}}(\pi_{\boldsymbol{\vartheta}}) - V(\pi_{\boldsymbol{\vartheta}})\right] = \mathbb{E}_{\pi_{\boldsymbol{\vartheta}}}\left[r_{\boldsymbol{x},a}\left(\frac{\beta^*(a|\boldsymbol{x})}{\hat{\beta}(a|\boldsymbol{x})} - 1\right)\right]$$

$$N \cdot \text{Var}_D\left(\hat{V}_{\text{BIPS}}(\pi_{\boldsymbol{\vartheta}})\right) = \text{Var}_{\pi_{\boldsymbol{\vartheta}}}\left(\frac{\beta^*(a|\boldsymbol{x})}{\hat{\beta}(a|\boldsymbol{x})}r_{\boldsymbol{x},a}\right) + \mathbb{E}_{\pi_{\boldsymbol{\vartheta}}}\left[\left(\frac{\pi_{\boldsymbol{\vartheta}}(a|\boldsymbol{x})}{\beta^*(a|\boldsymbol{x})} - 1\right)\frac{\beta^*(a|\boldsymbol{x})^2}{\hat{\beta}(a|\boldsymbol{x})^2}r_{\boldsymbol{x},a}^2\right]$$

However, smaller $\hat{\beta}(a|\boldsymbol{x})$ usually implies less number of related training samples in the logged data, and thus $\hat{\beta}(a|\boldsymbol{x})$ can be inaccurate with a higher probability. To make it more explicit, let us revisit the empirical results shown in Figure 1. We followed the method introduced in [8] to estimate the logging policy on KuaiRec dataset [12] and plotted the estimated $\hat{\beta}(a|\boldsymbol{x})$ and its corresponding uncertainties on items of different observation frequencies in the logged dataset. We adopted the method in [45] to measure the confidence interval of $\hat{\beta}(a|\boldsymbol{x})$ on each instance. A wider confidence interval, i.e., higher uncertainty in estimation, implies that with a high probability the true value may be further away from the empirical mean estimate. We can observe in Figure 1 that as item frequency decreases, the estimated logging probability also decreases, but the estimation uncertainty increases. This implies that a smaller $\hat{\beta}(a|\boldsymbol{x})$ is usually 1) more inaccurate and 2) associated with a higher uncertainty.

As a result, with high bias and variance caused by inaccurate $\hat{\beta}(a|\boldsymbol{x})$, it is erroneous to learn $\pi_{\boldsymbol{\vartheta}}(a|\boldsymbol{x})$ by simply optimizing $\hat{V}_{\text{BIPS}}(\pi_{\boldsymbol{\vartheta}})$. Furthermore, this approach may also hinder the convergence of off-policy learning, as discussed later in Section 3.2.

## 3 Uncertainty-aware off-policy learning

Our idea is to consider the uncertainty of the estimated logging policy by incorporating per-sample weight $\phi_{\boldsymbol{x},a}$, and perform policy learning by optimizing the following empirical estimator:

$$\hat{V}_{\text{UIPS}}(\pi_{\boldsymbol{\vartheta}}) = \frac{1}{N}\sum_{n=1}^{N}\frac{\pi_{\boldsymbol{\vartheta}}(a_n|\boldsymbol{x}_n)}{\hat{\beta}(a_n|\boldsymbol{x}_n)} \cdot \phi_{\boldsymbol{x}_n,a_n} \cdot r_{\boldsymbol{x}_n,a_n}. \tag{4}$$

Intuitively, one should assign lower weights to samples whose $\hat{\beta}(a|\boldsymbol{x})$ is small and far away from the ground-truth $\beta^*(a|\boldsymbol{x})$. We then divide off-policy optimization into two iterative steps:

- **Deriving the optimal instance weight:** Find the optimal $\phi_{\boldsymbol{x},a}$ to make $\hat{V}_{\mathrm{UIPS}}(\pi_{\boldsymbol{\vartheta}})$ approach its ground-truth $V(\pi)$ as closely as possible, so as to facilitate policy learning. The derived optimal weight is denoted as $\phi_{\boldsymbol{x},a}^*$ (see Theorem 3.2).
- **Policy improvement**: Update the policy $\pi_{\boldsymbol{\vartheta}}(a|\boldsymbol{x})$ using the following gradient:

$$\nabla_{\boldsymbol{\vartheta}} \hat{V}_{\mathrm{UIPS}}(\pi_{\boldsymbol{\vartheta}}) = \frac{1}{N} \sum_{n=1}^{N} \frac{\pi_{\boldsymbol{\vartheta}}(a_n|\boldsymbol{x}_n)}{\hat{\beta}(a_n|\boldsymbol{x}_n)} \cdot \phi_{\boldsymbol{x}_n,a_n}^* \cdot r_{\boldsymbol{x}_n,a_n} \nabla_{\boldsymbol{\vartheta}} \log(\pi_{\boldsymbol{\vartheta}}(a_n|\boldsymbol{x}_n)) \tag{5}$$

The whole algorithm framework and its computational cost, as well as important notations are summarized in Appendix 7.1.

### 3.1 Derive the optimal uncertainty-aware instance weight

We expect to find the optimal weight $\phi_{x,a}$ to make the empirical estimator $\hat{V}_{\mathrm{UIPS}}(\pi_{\boldsymbol{\vartheta}})$ as accurate as possible, taking into account the uncertainty in estimated logging probabilities. Intuitively, a high accuracy of the estimator is crucial for determining the correct direction of policy learning. We follow previous work [36, 29] and measure the mean squared error (MSE) of $\hat{V}_{\mathrm{UIPS}}(\pi_{\boldsymbol{\vartheta}})$ to the ground-truth policy value $V(\pi_{\boldsymbol{\vartheta}})$, which captures both the bias and variance of an estimator. A lower MSE indicates a more accurate estimator.

In UIPS, instead of directly minimizing the MSE, which is intractable, we find $\phi_{\boldsymbol{x},a}$ to minimize the upper bound of MSE. As we show later, the optimal $\phi_{\boldsymbol{x},a}$ has a closed-form solution which relates to both the value of $\pi_{\boldsymbol{\vartheta}}(a|\boldsymbol{x})/\hat{\beta}(a|\boldsymbol{x})$ and the estimation uncertainty of $\hat{\beta}(a|\boldsymbol{x})$.

**Theorem 3.1.** *The mean squared error (*MSE*) between $\hat{V}_{\mathrm{UIPS}}(\pi_{\boldsymbol{\vartheta}})$ and ground-truth estimator $V(\pi_{\boldsymbol{\vartheta}})$ is upper bounded as follows:*

$$\mathrm{MSE}\left(\hat{V}_{\mathrm{UIPS}}(\pi_{\boldsymbol{\vartheta}})\right) = \mathbb{E}_D\left[\left(\hat{V}_{\mathrm{UIPS}}(\pi_{\boldsymbol{\vartheta}}) - V(\pi_{\boldsymbol{\vartheta}})\right)^2\right] = \mathrm{Bias}\left(\hat{V}_{\mathrm{UIPS}}(\pi_{\boldsymbol{\vartheta}})\right)^2 + \mathrm{Var}\left(\hat{V}_{\mathrm{UIPS}}(\pi_{\boldsymbol{\vartheta}})\right)$$

$$\leq \mathbb{E}_{\pi_{\boldsymbol{\vartheta}}}\left[r_{\boldsymbol{x},a}^2 \frac{\pi_{\boldsymbol{\vartheta}}(a|\boldsymbol{x})}{\beta^*(a|\boldsymbol{x})}\right] \cdot \mathbb{E}_{\beta^*}\left[\left(\frac{\beta^*(a|\boldsymbol{x})}{\hat{\beta}(a|\boldsymbol{x})}\phi_{\boldsymbol{x},a} - 1\right)^2\right] + \mathbb{E}_{\beta^*}\left[\frac{\pi_{\boldsymbol{\vartheta}}(a|\boldsymbol{x})^2}{\hat{\beta}(a|\boldsymbol{x})^2}\phi_{\boldsymbol{x},a}^2\right].$$

As the first expectation term $\mathbb{E}_{\pi_{\boldsymbol{\vartheta}}}\left[r_{\boldsymbol{x},a}^2 \frac{\pi_{\boldsymbol{\vartheta}}(a|\boldsymbol{x})}{\beta^*(a|\boldsymbol{x})}\right]$ is a non-negative constant, we denote it as $\lambda \in [0, \infty)$ when searching for $\phi_{\boldsymbol{x},a}$. To minimize this upper bound of MSE, the optimal $\phi_{\boldsymbol{x},a}$ for each sample $(\boldsymbol{x}, a)$ should minimize the following,

$$\lambda \left(\frac{\beta^*(a|\boldsymbol{x})}{\hat{\beta}(a|\boldsymbol{x})}\phi_{\boldsymbol{x},a} - 1\right)^2 + \frac{\pi_{\boldsymbol{\vartheta}}(a|\boldsymbol{x})^2}{\hat{\beta}(a|\boldsymbol{x})^2}\phi_{\boldsymbol{x},a}^2. \tag{6}$$

An interesting observation is that setting $\phi_{\boldsymbol{x},a} = \frac{\hat{\beta}(a|\boldsymbol{x})}{\beta^*(a|\boldsymbol{x})}$, i.e., turning $\frac{\pi(a|\boldsymbol{x})}{\hat{\beta}(a|\boldsymbol{x})}\phi_{\boldsymbol{x},a}$ into $\frac{\pi(a|\boldsymbol{x})}{\beta^*(a|\boldsymbol{x})}$ does not result in the optimal solution of Eq.(6). This is because such a setting only reduces bias (i.e., the first term of Eq.(6)), but fails to control the second term, which is related to the variance. Moreover, we cannot directly minimize Eq.(6) due to the unknown $\beta^*(a|\boldsymbol{x})$. But it is possible to obtain a confidence interval which contains $\beta^*(a|\boldsymbol{x})$ with a high probability, when $\hat{\beta}(a|\boldsymbol{x})$ is obtained via a specific estimator, e.g., (generalized) linear model or kernel methods.

Following previous work [23, 16, 22], we adopt the realizable assumption that $\beta^*(a|\boldsymbol{x})$ can be represented by a softmax function applied over a parametric function $f_{\boldsymbol{\theta}^*}(\boldsymbol{x}, a)$. Moreover, the universal approximation theorem [18] states that a parametric function with sufficient capacity, when combined with a softmax function, can approximate any distribution. Then we have:

$$\beta^*(a|\boldsymbol{x}) \propto \exp(f_{\boldsymbol{\theta}^*}(\boldsymbol{x}, a)), \hat{\beta}(a|\boldsymbol{x}) \propto \exp(f_{\boldsymbol{\theta}}(\boldsymbol{x}, a)), \tag{7}$$

where $f_{\boldsymbol{\theta}}(\boldsymbol{x}, a)$ is an estimate of $f_{\boldsymbol{\theta}^*}(\boldsymbol{x}, a)$. Following the conventional definition of confidence interval [20], we define $\gamma$ and $U_{\boldsymbol{x},a}$ such that $|f_{\boldsymbol{\theta}^*}(\boldsymbol{x}, a) - f_{\boldsymbol{\theta}}(\boldsymbol{x}, a)| \leq \gamma U_{\boldsymbol{x},a}$ holds with probability

at least 1-$\delta$, where $\gamma$ is a function of $\delta$ (typically the smaller $\delta$ is, the larger $\gamma$ is). Then $\gamma U_{\boldsymbol{x},a}$ measures the width of confidence interval of $f_{\boldsymbol{\theta}}(\boldsymbol{x},a)$ against its ground-truth $f_{\boldsymbol{\theta}^*}(\boldsymbol{x},a)$. As derived in Appendix 7.2, with probability at least 1-$\delta$, we have $\beta^*(a|\boldsymbol{x}) \in \boldsymbol{B}_{\boldsymbol{x},a}$ and

$$\boldsymbol{B}_{\boldsymbol{x},a} = \left[ \frac{\hat{Z}\exp\left(-\gamma U_{\boldsymbol{x},a}\right)}{Z^*}\hat{\beta}(a|\boldsymbol{x}), \frac{\hat{Z}\exp\left(\gamma U_{\boldsymbol{x},a}\right)}{Z^*}\hat{\beta}(a|\boldsymbol{x}) \right],$$

where $Z^* = \sum_{a'} \exp(f_{\boldsymbol{\theta}^*}(a'|\boldsymbol{x}))$ and $\hat{Z} = \sum_{a'} \exp(f_{\theta}(a'|\boldsymbol{x}))$.

As $\beta^*(a|\boldsymbol{x})$ can be any value in $\boldsymbol{B}_{\boldsymbol{x},a}$ with high probability, we aim to find the optimal $\phi_{\boldsymbol{x},a}$ that minimizes the worst case of Eq.(6), thereby ensuring that $\hat{V}_{\mathrm{UIPS}}(\pi_{\boldsymbol{\vartheta}})$ approaches its ground-truth $V(\pi_{\boldsymbol{\vartheta}})$ under the sense of MSE, even in the worst possible scenarios. This ensures the subsequent policy improvement direction will not be much worse with high probability. Thus, we formulate the following optimization problem:

$$\min_{\phi_{\boldsymbol{x},a}} \max_{\beta_{\boldsymbol{x},a} \in \boldsymbol{B}_{\boldsymbol{x},a}} \lambda \left( \frac{\beta_{\boldsymbol{x},a}}{\hat{\beta}(a|\boldsymbol{x})}\phi_{\boldsymbol{x},a} - 1 \right)^2 + \frac{\pi_{\boldsymbol{\vartheta}}(a|\boldsymbol{x})^2}{\hat{\beta}(a|\boldsymbol{x})^2}\phi_{\boldsymbol{x},a}^2. \tag{8}$$

The following theorem derives a closed-form formula for the optimal solution of Eq.(8).

**Theorem 3.2.** *Let $\eta \in [\exp(-\gamma U_{\boldsymbol{x}}^{\max}), \exp(\gamma U_{\boldsymbol{x}}^{\max})]$, where $U_{\boldsymbol{x}}^{\max} = \max_a U_{\boldsymbol{x},a}$. The optimization problem in Eq.(8) has a closed-form solution:*

$$\phi_{\boldsymbol{x},a}^* = \min\left( \lambda \Big/ \left[ \frac{\lambda}{\eta}\exp\left(-\gamma U_{\boldsymbol{x},a}\right) + \frac{\eta\pi_{\boldsymbol{\vartheta}}(a|\boldsymbol{x})^2}{\hat{\beta}(a|\boldsymbol{x})^2\exp\left(-\gamma U_{\boldsymbol{x},a}\right)} \right], 2\eta\Big/\left[ \exp\left(\gamma U_{\boldsymbol{x},a}\right) + \exp\left(-\gamma U_{\boldsymbol{x},a}\right) \right] \right).$$

The following corollary demonstrates the advantage of UIPS. The detailed proof of Theorem 3.2 and Corollary 3.3 can be found in Appendix 7.8.

**Corollary 3.3.** *With $\phi_{\boldsymbol{x},a}^*$ derived in Theorem 3.2, $\hat{V}_{\mathrm{UIPS}}(\pi_{\boldsymbol{\vartheta}})$ in Eq.(4) achieves a smaller upper bound of MSE than $\hat{V}_{\mathrm{BIPS}}(\pi_{\boldsymbol{\vartheta}})$ in Eq. (3).*

**Insights about $\phi_{\boldsymbol{x},a}^*$.** The detailed analysis of the effect of $\phi_{\boldsymbol{x},a}^*$ can be found in Lemma 7.1 in Appendix 7.8. In summary, we have the following key findings,

- For samples whose largest possible propensity score is under control: i.e., $\frac{\pi_{\boldsymbol{\vartheta}}(a|\boldsymbol{x})}{\min \boldsymbol{B}_{\boldsymbol{x},a}} < \sqrt{\lambda}$, higher uncertainty implies smaller values of $\pi/\hat{\beta}$. This suggests samples of this type with positive rewards are underestimated, and the extend of underestimation increases with the estimation uncertainty. UIPS thus chooses to increase $\phi_{\boldsymbol{x},a}^*$ with uncertainty, to emphasize these long-tail positive samples.
- Conversely, for samples with large propensity scores, UIPS decreases $\phi_{\boldsymbol{x},a}^*$ as the uncertainty increases, so as to prevent their distortion in policy learning.

**Uncertainty estimation.** Now we describe how to calculate $U_{\boldsymbol{x},a}$, i.e., the uncertainty of the estimated $\hat{\beta}(a|\boldsymbol{x})$. In this work, we choose to estimate $\beta^*(a|\boldsymbol{x})$ using a neural network, because 1) its representation learning capacity has been proved in numerous studies, and 2) various ways [11, 45] can be leveraged to perform the uncertainty estimation in a neural network. We adopt [45] due to its computational efficiency and theoretical soundness. Following the proof of Theorem 4.4 in [45], given the logged dataset $D$, we can get with a high probability that there exists $\gamma$ such that:

$$|f_{\boldsymbol{\theta}}(\boldsymbol{x}_n, a_n) - f_{\boldsymbol{\theta}^*}(\boldsymbol{x}_n, a_n))| \leq \gamma\sqrt{\boldsymbol{g}(\boldsymbol{x}_n, a_n)^\top \boldsymbol{M}_D^{-1}\boldsymbol{g}(\boldsymbol{x}_n, a_n)}$$

where $\boldsymbol{g}(\boldsymbol{x}_n, a_n)$ is the gradient of $f_{\boldsymbol{\theta}}(\boldsymbol{x}_n, a_n)$ with respect to the neural network's last layer's parameter $\boldsymbol{\theta}_w \subset \boldsymbol{\theta}$, i.e., $\boldsymbol{g}(\boldsymbol{x}_n, a_n) = \nabla_{\boldsymbol{\theta}_w} f_{\boldsymbol{\theta}}(\boldsymbol{x}_n, a_n)$. And $\boldsymbol{M}_D = \sum_{n=1}^N \boldsymbol{g}(\boldsymbol{x}_n, a_n)\boldsymbol{g}(\boldsymbol{x}_n, a_n)^\top$, implying $U_{\boldsymbol{x}_n, a_n} = \sqrt{\boldsymbol{g}(\boldsymbol{x}_n, a_n)^\top \boldsymbol{M}_D^{-1}\boldsymbol{g}(\boldsymbol{x}_n, a_n)}$.

### 3.2 Convergence of policy learning under UIPS

The following theorem provides the convergence result for UIPS, which converges to a stationary point of the expected reward function. The proof is provided in Appendix 7.9.

**Theorem 3.4.** *Denote $G_{\max}$ and $\Phi$ as the maximum value of $\|\frac{\partial \pi_\vartheta(a|\boldsymbol{x})}{\partial \vartheta}\|$ and $\mathbb{E}_{\beta^*}\left[\frac{\pi_\vartheta^2(a|\boldsymbol{x})}{\hat{\beta}^2(a|\boldsymbol{x})}(\phi_{\boldsymbol{x},a}^*)^2\right]$ respectively, i.e., $\|\frac{\partial \pi_\vartheta(a|\boldsymbol{x})}{\partial \vartheta}\| \leq G_{\max}$ and $\mathbb{E}_{\beta^*}\left[\frac{\pi_\vartheta^2(a|\boldsymbol{x})}{\hat{\beta}^2(a|\boldsymbol{x})}(\phi_{\boldsymbol{x},a}^*)^2\right] \leq \Phi$. And denote $V_{\max}$ as the finite maximum expected reward that can be achieved, and $\varphi_{\max} = \max_{\boldsymbol{x},a}\left\{\left|\frac{\beta^*(a|\boldsymbol{x})}{\hat{\beta}(a|\boldsymbol{x})}\phi_{\boldsymbol{x},a}^* - 1\right|\right\}$. Assume that the expected reward of $\pi_\vartheta$, i.e., $V(\pi_\vartheta)$, is a differentiable and L-smooth function w.r.t $\vartheta$. Denote the policy parameters obtained by Eq.(5) at iteration $k \in [K]$ as $\vartheta_k$, then $\varphi_{\max} \in (0,1)$ and*

$$\frac{1}{K}\sum_{k=1}^{K} \mathbb{E}[\|\nabla V(\pi_{\vartheta_k})\|]^2 \leq \frac{2LV_{\max}}{K(1-\varphi_{\max})} + \left(L + \frac{2V_{\max}}{(1-\varphi_{\max})}\right)\frac{G_{\max}\sqrt{\Phi}}{\sqrt{K}},$$

*where $\nabla V(\pi_\vartheta)$ is the true policy gradient under ground-truth logging probability, i.e., $\nabla V(\pi_\vartheta) = E_{\beta^*}[\frac{\pi_\vartheta(a|\boldsymbol{x})}{\beta^*(a|\boldsymbol{x})}r_{\boldsymbol{x},a}\nabla_\vartheta \log(\pi_\vartheta(a|\boldsymbol{x}))]$.*

Theorem 3.4 shows that, as $K \to \infty$ and with $1/(1-\varphi_{\max})$ and $\Phi$ being controlled, UIPS leads policy update to converge to a stationary point where the true policy gradient $\nabla V(\pi_{\vartheta_k})$ is zero. And fortunately, UIPS is effective in controlling both $1/(1-\varphi_{\max})$ and $\Phi$. Specifically, we denote $\varphi_{\boldsymbol{x},a} = \left|\frac{\beta^*(a|\boldsymbol{x})}{\hat{\beta}(a|\boldsymbol{x})}\phi_{\boldsymbol{x},a}^* - 1\right|$ and $\Phi_{x,a} = \frac{\pi_\vartheta^2(a|\boldsymbol{x})}{\hat{\beta}^2(a|\boldsymbol{x})}(\phi_{\boldsymbol{x},a}^*)^2$. It is clear to note that $\lambda\varphi_{x,a}^2 + \Phi_{x,a}$ corresponds to the objective in Eq.(6) for deriving $\phi_{\boldsymbol{x},a}^*$ for each sample $(\boldsymbol{x}, a)$. In other words, UIPS selects $\{\phi_{\boldsymbol{x},a}^*\}$ to minimize $\varphi_{\max} = \max\{\varphi_{x,a}\}$ and $\Phi = \mathbb{E}_{\beta^*}[\Phi_{\boldsymbol{x},a}]$, which directly accelerate the policy converge to a stationary point with the true policy gradient being zero.

In the case of BIPS in Eq.(3), we have $\phi_{\boldsymbol{x},a} \equiv 1$. Although $\Phi$ may be large due to small logging probabilities, the more concerning issue is that the requirement $\varphi_{\max} \in (0,1)$ is no longer satisfied when $\beta^*(a|\boldsymbol{x}) \geq 2\hat{\beta}(a|\boldsymbol{x})$, which may happen with a non-negligible probability. Hence, the convergence of policy learning under $\hat{V}_{\text{BIPS}}$ is no better than that under UIPS.

## 4 Empirical evaluations

We evaluate UIPS on both synthetic data and three real-world datasets with unbiased collection. We compare UIPS with the following baselines, which can be grouped into five categories:

- **Cross-Entropy (CE)**: A supervised learning method with the cross-entropy loss over its softmax output. No off-policy correction is performed in this method.
- **BIPS-Cap** [8]: The off-policy learning solution under the BIPS estimator in Eq.(3). The estimated propensity scores are further suppressed to control variance, i.e., taking $\min\left(c, \frac{\pi_\vartheta(a|\boldsymbol{x})}{\hat{\beta}(a|\boldsymbol{x})}\right)$ as the propensity score. Setting $c$ to a small value can reduce variance, but introduces bias.
- **MinVar** & **stableVar** [46], **Shrinkage** [36]: This line of work improves off-policy evaluation by reweighing each sample. For example, MinVar and stableVar reweigh each sample by $\frac{h_{\boldsymbol{x},a}}{\sum_{a'}h_{\boldsymbol{x},a'}}$ with $h_{\boldsymbol{x},a} = \frac{\hat{\beta}(a|\boldsymbol{x})}{\pi_\vartheta(a|\boldsymbol{x})^2}$ and $h_{\boldsymbol{x},a} = \frac{\sqrt{\hat{\beta}(a|\boldsymbol{x})}}{\pi_\vartheta(a|\boldsymbol{x})}$ respectively, since they find that $\pi_\vartheta(a|\boldsymbol{x})^2/\hat{\beta}(a|\boldsymbol{x})$ is directly related to policy evaluation variance. Su et al. [36] propose to shrink the propensity score by $\lambda/(\lambda + \frac{\pi_\vartheta(a|\boldsymbol{x})^2}{\hat{\beta}(a|\boldsymbol{x})^2})$, which is a special case of our UIPS with $U_{\boldsymbol{x},a} = 0$ and $\eta = 1$. All these methods simply treat $\hat{\beta}(a|\boldsymbol{x})$ as $\beta^*(a|\boldsymbol{x})$, and none of them consider the uncertainty of $\hat{\beta}(a|\boldsymbol{x})$.
- **SNIPS** [39], **BanditNet** [16], **POEM** [38], **POXM** [23], **Adaptive** [22]: This line of work aims for more stable and accurate policy learning. For example, SNIPS normalizes the estimator by the sum of propensity scores in each batch. BanditNet extends SNIPS and leverages an additional Lagrangian term to normalize the estimator by an approximated sum of propensity scores of all samples. POEM jointly optimizes the estimator and its variance. POXM controls estimation variance by pruning samples with small logging probabilities. Adaptive proposes a new formulation to utilize negative samples.
- **ApproxKNN** [5] and **IPS-C-TS**: The line of work improves off-policy learning by applying calibration to estimated logging probabilities. ApproxKNN utilizes the K-Nearest Neighbor algorithm for calibration, which exhibits the lowest calibration error in [5]. IPS-C-TS, on the other hand, employs temperature scaling, a widely recognized and effective calibration method for probability distribution [13].

Table 1: Experiment results on synthetic datasets. The best and second best results are highlighted with **bold** and underline respectively. The $p$-value under the t-test between UIPS and the best baseline on each dataset is also provided.

| | $\tau=0.5$ | | | $\tau=1$ | | | $\tau=2$ | | |
|---|---|---|---|---|---|---|---|---|---|
| Algorithm | P@5 | R@5 | NDCG@5 | P@5 | R@5 | NDCG@5 | P@5 | R@5 | NDCG@5 |
| IPS-GT | $0.5589_{\pm 1e^{-3}}$ | $0.1582_{\pm 6e^{-4}}$ | $0.6093_{\pm 1e^{-3}}$ | $0.5526_{\pm 2e^{-3}}$ | $0.1565_{\pm 6e^{-4}}$ | $0.6007_{\pm 1e^{-3}}$ | $0.5531_{\pm 2e^{-3}}$ | $0.1557_{\pm 7e^{-4}}$ | $0.6037_{\pm 1e^{-3}}$ |
| CE | $0.5553_{\pm 6e^{-4}}$ | $0.1573_{\pm 2e^{-4}}$ | $0.6037_{\pm 5e^{-4}}$ | $0.5510_{\pm 6e^{-4}}$ | $0.1561_{\pm 2e^{-4}}$ | $0.5995_{\pm 4e^{-4}}$ | $0.5386_{\pm 2e^{-3}}$ | $0.1524_{\pm 7e^{-4}}$ | $0.5874_{\pm 2e^{-3}}$ |
| BIPS-Cap | $0.5515_{\pm 2e^{-3}}$ | $0.1553_{\pm 8e^{-4}}$ | $0.6031_{\pm 2e^{-3}}$ | $0.5526_{\pm 2e^{-3}}$ | $0.1561_{\pm 6e^{-4}}$ | $0.6016_{\pm 1e^{-3}}$ | $0.5409_{\pm 3e^{-3}}$ | $0.1529_{\pm 9e^{-4}}$ | $0.5901_{\pm 2e^{-3}}$ |
| MinVar | $0.5340_{\pm 2e^{-3}}$ | $0.1509_{\pm 6e^{-4}}$ | $0.5857_{\pm 2e^{-3}}$ | $0.5282_{\pm 2e^{-3}}$ | $0.1491_{\pm 7e^{-4}}$ | $0.5791_{\pm 2e^{-3}}$ | $0.5036_{\pm 4e^{-3}}$ | $0.1415_{\pm 1e^{-3}}$ | $0.5543_{\pm 3e^{-3}}$ |
| stableVar | $0.4577_{\pm 5e^{-3}}$ | $0.1310_{\pm 1e^{-3}}$ | $0.5111_{\pm 2e^{-3}}$ | $0.5373_{\pm 3e^{-3}}$ | $0.1523_{\pm 9e^{-4}}$ | $0.5866_{\pm 3e^{-3}}$ | $0.5279_{\pm 3e^{-3}}$ | $0.1492_{\pm 8e^{-4}}$ | $0.5781_{\pm 3e^{-3}}$ |
| Shrinkage | $0.5526_{\pm 2e^{-3}}$ | $0.1562_{\pm 7e^{-4}}$ | $0.6024_{\pm 1e^{-3}}$ | $0.5499_{\pm 4e^{-3}}$ | $0.1545_{\pm 1e^{-4}}$ | $0.6040_{\pm 3e^{-3}}$ | $0.5347_{\pm 2e^{-3}}$ | $0.1513_{\pm 6e^{-4}}$ | $0.5824_{\pm 2e^{-3}}$ |
| SNIPS | $0.2616_{\pm 6e^{-2}}$ | $0.0749_{\pm 2e^{-2}}$ | $0.3150_{\pm 7e^{-2}}$ | $0.3538_{\pm 5e^{-2}}$ | $0.0987_{\pm 1e^{-2}}$ | $0.4144_{\pm 6e^{-2}}$ | $0.4379_{\pm 3e^{-2}}$ | $0.1226_{\pm 9e^{-3}}$ | $0.5177_{\pm 3e^{-2}}$ |
| BanditNet | $0.4011_{\pm 3e^{-2}}$ | $0.1131_{\pm 8e^{-3}}$ | $0.4830_{\pm 2e^{-2}}$ | $0.3894_{\pm 4e^{-2}}$ | $0.1095_{\pm 1e^{-2}}$ | $0.4741_{\pm 3e^{-2}}$ | $0.4122_{\pm 3e^{-2}}$ | $0.1153_{\pm 8e^{-3}}$ | $0.4934_{\pm 3e^{-2}}$ |
| POEM | $0.5480_{\pm 2e^{-3}}$ | $0.1539_{\pm 8e^{-4}}$ | $0.6008_{\pm 2e^{-3}}$ | $0.5502_{\pm 2e^{-3}}$ | $0.1551_{\pm 6e^{-4}}$ | $0.6000_{\pm 2e^{-3}}$ | $0.5399_{\pm 2e^{-3}}$ | $0.1526_{\pm 8e^{-4}}$ | $0.5893_{\pm 2e^{-3}}$ |
| POXM | $0.4006_{\pm 3e^{-2}}$ | $0.1130_{\pm 8e^{-3}}$ | $0.4828_{\pm 2e^{-2}}$ | $0.3616_{\pm 4e^{-2}}$ | $0.1019_{\pm 1e^{-2}}$ | $0.4522_{\pm 4e^{-2}}$ | $0.3816_{\pm 4e^{-2}}$ | $0.1069_{\pm 1e^{-2}}$ | $0.4680_{\pm 4e^{-2}}$ |
| Adaptive | $0.3831_{\pm 2e^{-2}}$ | $0.1050_{\pm 4e^{-3}}$ | $0.4382_{\pm 2e^{-2}}$ | $0.4734_{\pm 4e^{-3}}$ | $0.1325_{\pm 1e^{-3}}$ | $0.5326_{\pm 3e^{-3}}$ | $0.3936_{\pm 1e^{-2}}$ | $0.1097_{\pm 4e^{-3}}$ | $0.4368_{\pm 2e^{-2}}$ |
| ApproxKNN | $0.5576_{\pm 1e^{-3}}$ | $0.1580_{\pm 4e^{-4}}$ | $0.6059_{\pm 2e^{-3}}$ | $0.5527_{\pm 9e^{-4}}$ | $0.1567_{\pm 1e^{-4}}$ | $0.6010_{\pm 1e^{-3}}$ | $0.5409_{\pm 2e^{-3}}$ | $0.1532_{\pm 6e^{-4}}$ | $0.5890_{\pm 1e^{-3}}$ |
| IPS-C-TS | $0.5565_{\pm 1e^{-3}}$ | $0.1577_{\pm 3e^{-4}}$ | $0.6048_{\pm 7e^{-4}}$ | $0.5517_{\pm 6e^{-4}}$ | $0.1563_{\pm 2e^{-4}}$ | $0.6002_{\pm 6e^{-4}}$ | $0.5393_{\pm 1e^{-3}}$ | $0.1526_{\pm 5e^{-4}}$ | $0.5879_{\pm 1e^{-3}}$ |
| UIPS-P | $0.4019_{\pm 3e^{-2}}$ | $0.1131_{\pm 1e^{-2}}$ | $0.4831_{\pm 3e^{-2}}$ | $0.3904_{\pm 4e^{-2}}$ | $0.1096_{\pm 1e^{-2}}$ | $0.4749_{\pm 3e^{-2}}$ | $0.4109_{\pm 3e^{-2}}$ | $0.1149_{\pm 1e^{-2}}$ | $0.4922_{\pm 3e^{-2}}$ |
| UIPS-O | $0.4135_{\pm 4e^{-2}}$ | $0.1167_{\pm 1e^{-2}}$ | $0.4954_{\pm 4e^{-2}}$ | $0.3896_{\pm 4e^{-2}}$ | $0.1096_{\pm 1e^{-2}}$ | $0.4739_{\pm 3e^{-2}}$ | $0.4519_{\pm 3e^{-2}}$ | $0.1268_{\pm 8e^{-3}}$ | $0.5296_{\pm 2e^{-2}}$ |
| UIPS | $\mathbf{0.5608}_{\pm 2e^{-3}}$ | $\mathbf{0.1589}_{\pm 8e^{-4}}$ | $\mathbf{0.6113}_{\pm 3e^{-3}}$ | $\mathbf{0.5572}_{\pm 2e^{-3}}$ | $\mathbf{0.1571}_{\pm 8e^{-4}}$ | $\mathbf{0.6074}_{\pm 2e^{-3}}$ | $\mathbf{0.5432}_{\pm 3e^{-3}}$ | $\mathbf{0.1534}_{\pm 8e^{-4}}$ | $\mathbf{0.5946}_{\pm 2e^{-3}}$ |
| p-value | $3e^{-3}$ | $1e^{-2}$ | $4e^{-5}$ | $2e^{-5}$ | $2e^{-1}$ | $4e^{-10}$ | $1e^{-1}$ | $5e^{-1}$ | $4e^{-2}$ |

Table 2: Performance under different uncertainties.

| | Actions on Samples with High Uncertainty | | | Actions on Samples with Low Uncertainty | | |
|---|---|---|---|---|---|---|
| Algorithm | P@5(RI) | R@5(RI) | NDCG@5(RI) | P@5(RI) | R@5(RI) | NDCG@5(RI) |
| CE | 0.5190 | 0.1231 | 0.5526 | 0.5913 | 0.1915 | 0.6549 |
| BIPS-Cap | 0.5117 (-1.41%) | 0.1202 (-2.33%) | 0.5488 (-0.68%) | 0.5913 (+0.00%) | 0.1903 (-0.64%) | 0.6574 (+0.39%) |
| Shrinkage | 0.5158 (-0.62%) | 0.1217 (-1.11%) | 0.5505 (-0.37%) | 0.5892 (-0.35%) | 0.1905 (-0.55%) | 0.6546 (-0.05%) |
| UIPS | 0.5222 (+0.61%) | 0.1237 (+0.50%) | 0.5568 (+0.77%) | 0.5994 (+1.38%) | 0.1940 (+1.28%) | 0.6658 (+1.66%) |

- **UIPS-P** and **UIPS-O**: These are two variants of our UIPS with different ways of leveraging uncertainties. UIPS-P directly penalizes samples whose estimated logging probabilities are of high uncertainty, i.e., taking $\phi_{\boldsymbol{x},a} = 1.0/\exp(\gamma U_{\boldsymbol{x},a})$, which follows previous work on offline reinforcement learning [43, 4]. UIPS-O adversarially uses the worst propensity score for policy learning, i.e., $\phi_{\boldsymbol{x},a} = 1.0/\exp(-\gamma U_{\boldsymbol{x},a})$.

## 4.1 Synthetic data

**Data generation.** Following previous work [24, 23], we generate a synthetic dataset by a supervise-to-bandit conversion on Wiki10-31K dataset [7], which is an extreme multi-label classification dataset. The Wiki10-31K dataset contains approximately 20K samples. Each sample is associated with a feature vector $\tilde{\boldsymbol{x}}$ of 101,938 dimensions and a label vector $\boldsymbol{y}_{\tilde{\boldsymbol{x}}}$ of 31K classes with more than one positive class. Let $\boldsymbol{y}_{\tilde{\boldsymbol{x}},a}$ denote the label of class $a$ under $\tilde{\boldsymbol{x}}$, and we take each class as an action. The huge action space creates great challenges in off-policy learning, e.g., sparse observations, and therefore better evaluates different methods.

We split the dataset into train, validation and test sets with size 11K:3K:6K. The test set is from the official split. Since the original feature vector $\tilde{\boldsymbol{x}}$ is too sparse, for ease of learning, we first embedded it to dimension $d$ by $\boldsymbol{x} = \boldsymbol{W}\tilde{\boldsymbol{x}}$, and synthesized the ground-truth logging policy $\beta^*(a|\boldsymbol{x})$ by:

$$\beta^*(a|\boldsymbol{x}) \propto \exp\left(\boldsymbol{x}^\top \boldsymbol{\theta}_a^*/\tau\right), \tag{9}$$

where $\boldsymbol{W}$ and $\{\boldsymbol{\theta}_a^*\}$ are pre-trained parameters by applying a logistic regression model on the train set, $\tau$ is a positive hyper-parameter that controls the skewness of logging distribution. A small value of $\tau$ leads to a near-deterministic logging policy, while a larger $\tau$ makes it flatter. More implementation details can be found in Appendix 7.3.

**Evaluation metrics.** To evaluate the learned policy $\pi_{\boldsymbol{\vartheta}}(a|\boldsymbol{x})$, we calculate Precision@K (P@K), Recall@K (R@K) and NDCG@K following previous work [23, 24]. Higher P@K, R@K and NDCG@K imply a better policy.

**Effectiveness of policy learning.** Table 1 shows the average performance and standard deviations of all algorithms under 10 random seeds on three synthetic datasets generated under different $\tau$. As the ground-truth logging policy is accessible on the synthetic datasets, we included a new baseline IPS-GT, which uses the IPS estimator with the ground-truth logging probabilities. We calculated $p$-value under t-test between UIPS and the best baseline to investigate the significance of improvement.

First, we can observe that UIPS achieved similar and even better performance than IPS-GT when $\tau = 0.5$ and $\tau = 1$, but performed worse than IPS-GT when $\tau = 2$. Despite using ground-truth

logging probabilities, IPS-GT still suffered from high variance caused by samples with small logging probabilities, which is the main cause of its worse performance when $\tau = 0.5$ and $\tau = 1$. In contrast, UIPS effectively controlled the negative impact of these high-variance samples, resulting in a better bias-variance trade-off.

With an increasing $\tau$, suggesting a decrease in the probability of selecting positive actions, most algorithms experienced a drop in performance. However, UIPS consistently outperformed all other algorithms across all three datasets and metrics. Interestingly, as $\tau$ decreases, the performance improvement of UIPS became even more pronounced, despite SNIPS, BanditNet, and POXM being designed to handle small logging probabilities of positive actions.

ApproxKNN and IPS-C-TS generally achieved better performance than BIPS-Cap, implying the effectiveness of calibration of estimated logging probabilities. However, UIPS still consistently outperformed both ApproxKNN and IPS-C-TS. The main reason is that calibration primarily focuses on adjusting the estimated probabilities to ensure on *average* the model's predictions are reliable and accurate. In contrast, UIPS specifically handles the impact from each *individual* sample in policy learning.

UIPS also consistently outperformed Shrinkage (a special case of UIPS with uncertainties always being zero) on all three datasets, demonstrating the benefits of considering the estimation uncertainty. Finally, blindly reweighing through uncertainties, regardless of their impact on the accuracy of the resulting estimator and the learned policy, ultimately resulted in poor performance, as demonstrated by UIPS-P and UIPS-O.

**Performance under different uncertainty levels.** As shown in Figure 1, low-frequency samples in the logged dataset suffer higher uncertainties in their propensity estimation. Thus, we divided the test set into two subsets according to the average frequency of associated actions, where the uncertainty in the subset associated with low-frequency actions is on average 8% higher than that in high-frequency actions. Table 2 shows the results on these two subsets when $\tau = 0.5$. In addition, we include the results of the top three baselines that directly utilize the estimated logging policy. Table 2 clearly demonstrates that only UIPS performed better than CE on the test set with low-frequency actions, implying the distortion of inaccurately estimated logging probabilities and the effectiveness of UIPS in efficiently handling them.

**Off-policy Evaluation.** We further inspected whether $\hat{V}_{\text{UIPS}}$ in Eq.(4) leads to more accurate off-policy evaluation. Following previous work [29, 46, 36], we evaluated the following $\epsilon$-greedy policy: $\pi(a|\boldsymbol{x}) = \frac{1-\epsilon}{|M_x|} \cdot \mathbb{I}\{a \in M_x\} + \epsilon/|\mathcal{A}|$, where $M_x$ contains all positive actions associated with instance $\boldsymbol{x}$. For each $\boldsymbol{x}$ in the test set, we randomly sample 100 actions following the logging policy in Eq.(9) to generate the logged dataset. Table 3 shows the MSE of the estimators to the ground-truth policy value under 20 different random seeds. From Table 3, one can observe that: 1) IPS-GT with a skewer logging policy (i.e., smaller $\tau$) leads to higher MSE, consistent with previous findings [29, 46, 36]; 2) inaccurate logging probabilities result in high bias and variance, leading to much larger MSE of BIPS compared to IPS-GT. Furthermore, this distortion is particularly pronounced when the ground-truth logging policy is skewed ($\tau = 0.5$); and 3) although all using the estimated logging policy, $\hat{V}_{\text{UIPS}}$ yields the smallest MSE, comparing to other baselines that are designed to improve over BIPS.

**Hyper-parameter Tuning.** Discussions about hyper-parameter tuning and performance of UIPS under different hyper-parameters can also be found in Appendix 7.3.1.

## 4.2 Real-world data

To demonstrate the effectiveness of UIPS in real-world scenarios, we evaluate it on three recommendation datasets: (1) Yahoo! R3[1]; (2) Coat[2]; (3) KuaiRec [12], for music, fashion and short-video recommendations respectively. All these datasets contain an unbiased test set collected from a randomized controlled trial where items are randomly selected. The statistics of the three datasets and implementation details, e.g., model architectures and dataset splits, can be found in Appendix 7.3.2.

Following [10], we take $K = 5$ on Yahoo! R3 and Coat datasets, and $K = 50$ on KuaiRec dataset. The $p$-value under the t-test between UIPS and the best baseline on each dataset is also reported to investigate the significance of improvement.

---

[1] `https://webscope.sandbox.yahoo.com/`
[2] `https://www.cs.cornell.edu/~schnabts/mnar/`

Table 3: MSE of different off-policy estimators. A lower MSE indicates a more accurate estimator.

| Algorithm | IPS-GT | BIPS | minVar | stableVar | Shrinage | UIPS |
|---|---|---|---|---|---|---|
| $\tau = 0.5$ | $0.0875\pm4e^{-4}$ | $15.786\pm1.51$ | $0.9021\pm7e^{-13}$ | $0.8612\pm5e^{-8}$ | $0.0718\pm5e^{-6}$ | $\mathbf{0.0210}\pm2e^{-6}$ |
| $\tau = 1.0$ | $0.0209\pm8e^{-5}$ | $0.5510\pm0.388$ | $0.9019\pm8e^{-12}$ | $0.8578\pm2e^{-7}$ | $0.1978\pm2e^{-5}$ | $\mathbf{0.0093}\pm1e^{-6}$ |
| $\tau = 2.0$ | $0.0020\pm6e^{-6}$ | $0.5669\pm0.013$ | $0.9015\pm5e^{-15}$ | $0.8342\pm5e^{-7}$ | $0.2952\pm3e^{-5}$ | $\mathbf{0.0043}\pm4e^{-7}$ |

Table 4: Experimental results on real-world datasets. The best and second best results are highlighted with **bold** and underline respectively. The p-value under the t-test between UIPS and the best baseline on each dataset is also provided.

| Algorithm | Yahoo | | | Coat | | | KuaiRec | | |
|---|---|---|---|---|---|---|---|---|---|
| | P@5 | R@5 | NDCG@5 | P@5 | R@5 | NDCG@5 | P@50 | R@50 | NDCG@50 |
| CE | $0.2819\pm2e^{-3}$ | $0.7594\pm6e^{-3}$ | $0.6073\pm7e^{-3}$ | $0.2799\pm5e^{-3}$ | $0.4618\pm1e^{-2}$ | $\underline{0.4529\pm7e^{-3}}$ | $0.8802\pm2e^{-3}$ | $0.0240\pm8e^{-5}$ | $0.8810\pm6e^{-3}$ |
| BIPS-Cap | $0.2808\pm2e^{-3}$ | $0.7576\pm5e^{-3}$ | $0.6099\pm8e^{-3}$ | $0.2758\pm6e^{-3}$ | $0.4582\pm7e^{-3}$ | $0.4399\pm9e^{-3}$ | $0.8750\pm3e^{-3}$ | $0.0238\pm7e^{-5}$ | $0.8788\pm5e^{-3}$ |
| MinVar | $0.2843\pm4e^{-3}$ | $\underline{0.7685\pm1e^{-2}}$ | $0.6168\pm1e^{-2}$ | $0.2813\pm3e^{-3}$ | $\underline{0.4668\pm9e^{-3}}$ | $0.4414\pm8e^{-3}$ | $0.8827\pm1e^{-3}$ | $0.0240\pm5e^{-5}$ | $0.8886\pm2e^{-3}$ |
| stableVar | $0.2787\pm2e^{-3}$ | $0.7499\pm7e^{-3}$ | $0.5919\pm7e^{-3}$ | $\underline{0.2840\pm3e^{-3}}$ | $0.4662\pm5e^{-3}$ | $0.4393\pm7e^{-3}$ | $0.8524\pm7e^{-3}$ | $0.0231\pm2e^{-4}$ | $0.8570\pm4e^{-3}$ |
| Shrinkage | $\underline{0.2843\pm3e^{-3}}$ | $0.7654\pm8e^{-3}$ | $\underline{0.6204\pm7e^{-3}}$ | $0.2790\pm5e^{-3}$ | $0.4636\pm4e^{-3}$ | $0.4464\pm1e^{-2}$ | $0.8744\pm3e^{-3}$ | $0.0238\pm9e^{-5}$ | $0.8771\pm6e^{-3}$ |
| SNIPS | $0.2222\pm4e^{-3}$ | $0.5828\pm1e^{-2}$ | $0.4357\pm1e^{-2}$ | $0.2643\pm7e^{-3}$ | $0.4287\pm1e^{-2}$ | $0.4009\pm9e^{-3}$ | $0.8411\pm6e^{-3}$ | $0.0228\pm2e^{-4}$ | $0.8431\pm6e^{-3}$ |
| BanditNet | $0.2413\pm8e^{-3}$ | $0.6442\pm2e^{-2}$ | $0.4988\pm2e^{-2}$ | $0.2781\pm8e^{-3}$ | $0.4527\pm1e^{-2}$ | $0.4251\pm1e^{-2}$ | $0.8758\pm2e^{-3}$ | $0.0239\pm2e^{-4}$ | $0.8810\pm4e^{-3}$ |
| POEM | $0.2732\pm3e^{-3}$ | $0.7357\pm1e^{-2}$ | $0.5880\pm1e^{-2}$ | $0.2791\pm4e^{-3}$ | $0.4566\pm6e^{-3}$ | $0.4375\pm6e^{-3}$ | $0.7785\pm1e^{-2}$ | $0.0210\pm2e^{-4}$ | $0.7779\pm6e^{-3}$ |
| POXM | $0.2250\pm5e^{-3}$ | $0.5940\pm1e^{-2}$ | $0.4542\pm2e^{-2}$ | $0.2663\pm6e^{-3}$ | $0.4308\pm9e^{-3}$ | $0.4006\pm1e^{-2}$ | $\underline{0.8962\pm1e^{-2}}$ | $\underline{0.0245\pm4e^{-4}}$ | $\underline{0.9041\pm1e^{-2}}$ |
| Adaptive | $0.2762\pm3e^{-3}$ | $0.7451\pm9e^{-3}$ | $0.5919\pm8e^{-3}$ | $0.2830\pm3e^{-3}$ | $0.4634\pm5e^{-3}$ | $0.4217\pm5e^{-3}$ | $0.8375\pm1e^{-2}$ | $0.0227\pm4e^{-4}$ | $0.8460\pm1e^{-2}$ |
| ApproxKNN | $0.2697\pm2e^{-2}$ | $0.7225\pm5e^{-3}$ | $0.5760\pm6e^{-3}$ | $0.2755\pm2e^{-3}$ | $0.4594\pm5e^{-3}$ | $0.4490\pm4e^{-3}$ | $0.8839\pm2e^{-6}$ | $0.0240\pm5e^{-5}$ | $0.8895\pm2e^{-3}$ |
| IPS-C-TS | $0.2816\pm2e^{-3}$ | $0.7582\pm5e^{-3}$ | $0.6114\pm5e^{-3}$ | $0.2799\pm3e^{-3}$ | $0.4625\pm7e^{-3}$ | $0.4462\pm6e^{-3}$ | $0.8781\pm3e^{-3}$ | $0.0239\pm1e^{-4}$ | $0.8749\pm3e^{-3}$ |
| UIPS-P | $0.1829\pm8e^{-3}$ | $0.4560\pm3e^{-2}$ | $0.3300\pm1e^{-2}$ | $0.2685\pm7e^{-3}$ | $0.4364\pm9e^{-3}$ | $0.4087\pm7e^{-3}$ | $0.8638\pm8e^{-3}$ | $0.0235\pm3e^{-4}$ | $0.8685\pm7e^{-3}$ |
| UIPS-O | $0.1947\pm3e^{-3}$ | $0.4959\pm1e^{-2}$ | $0.3600\pm8e^{-3}$ | $0.2657\pm5e^{-3}$ | $0.4306\pm9e^{-3}$ | $0.4146\pm9e^{-3}$ | $0.8651\pm8e^{-3}$ | $0.0235\pm2e^{-4}$ | $0.8697\pm7e^{-3}$ |
| UIPS | $\mathbf{0.2868}\pm2e^{-3}$ | $\mathbf{0.7742}\pm5e^{-3}$ | $\mathbf{0.6274}\pm5e^{-3}$ | $\mathbf{0.2877}\pm3e^{-3}$ | $\mathbf{0.4757}\pm5e^{-3}$ | $\mathbf{0.4576}\pm8e^{-3}$ | $\mathbf{0.9120}\pm1e^{-3}$ | $\mathbf{0.0250}\pm5e^{-5}$ | $\mathbf{0.9174}\pm7e^{-4}$ |
| p-value | $4e^{-2}$ | $1e^{-2}$ | $3e^{-2}$ | $2e^{-2}$ | $6e^{-4}$ | $5e^{-5}$ | $6e^{-4}$ | $6e^{-4}$ | $1e^{-3}$ |

We can first observe from Table 4 that on all three datasets, the proposed UIPS achieved the highest precision, recall and NDCG. Apparently, accurate estimation of logging probabilities in real-world scenarios with large action spaces and sparse interactions is challenging to achieve, causing BIPS-Cap to underperform CE. Additionally, calibration poses difficulties in such scenarios, leading to poor performance of ApproxKNN and IPS-C-TS across all three real-world datasets. BanditNet, POEM and POXM performed better in problems with a larger action space, while MinVar, stableVar and Shrinkage as well as Adaptive suited better for scenarios with a smaller action size. Again, UIPS outperformed Shrinkage, highlighting the importance of handling uncertainty in the estimated logging policy. But simply reweighing based on uncertainties, without considering their impact on the accuracy of the resulting estimator and the learned policy, led to poor performance, as shown by UIPS-P and UIPS-O.

### 4.3 Comparisons against other lines of off-policy learning

We discuss about the difference between UIPS and recent work with direct propensity estimation [34, 21], the DICE line of work [26, 47, 9] as well as the work on the distributionally robust off-policy learning [32, 17, 44] in Appendix 7.6, Appendix 7.5 and Appendix 7.7 respectively. Our empirical evaluations on recent work [26, 44, 34] from all three lines suggest that these lines of work cannot properly handle inaccuracies in the estimated logging probabilities that hinder policy improvement. Moreover, we also integrated the proposed UIPS with the doubly-robust (DR) estimator and our results suggest the accuracy of the imputation model greatly affected policy learning under DR, but UIPS still provides benefits. More details can be found in Appendix 7.4.

## 5 Related work

Our work is the first of its kind to account for the uncertainty of logging policy estimation in off-policy learning. The following two lines of work are most related to this work.

**Off-policy learning.** In many real-world applications, such as search engines and recommender systems, interactive online model update is expensive and risky [15]. Off-policy learning has therefore attracted great interest, since it can leverage the already logged feedback data [2, 8, 22]. The main challenge in off-policy learning is to address the mismatch between the logging policy and the learning policy. One common and widely-applied approach is to leverage the Inverse Propensity Scoring (IPS) method to correct the discrepancy between the two policies. And various methods are proposed to enhance IPS for more stabilizing learning [39, 37, 38] and improved variance control [23, 22], as well as extensions to more complex problems [40, 24].

However, all these solutions directly use the estimated logging policy for off-policy correction, leading to sub-optimal performance as shown in our experiments. A recent study on causal recommendation [10] also argues that propensity scores may not be correct due to unobserved confounders. They assume the effect of unobserved confounder for any sample can be bounded by a pre-defined hyper-parameter, and adversarially search for the worst-case propensity for learning. Mapping to off-policy learning, their solution is a special case of our UIPS-O variant with uncertainty as a pre-defined constant.

There were existing studies [34, 21, 26, 47, 9] also explore direct estimation of the propensity ratio to bypass estimating the logging policy. However, as discussed in Appendix 7.6 and Appendix 7.5, they demonstrate inferior performance compared to UIPS. This is primarily due to either the lack of consideration for the accuracy of the estimated propensity ratio, similar to the limitations of existing IPS-type algorithms in handling inaccurately estimated logging probabilities, or the degeneration to a specific IPS estimator that suffers high variance.

Recent work on distributionally robust off-policy evaluation and learning [32, 17, 44] also addresses uncertainty in off-policy learning. However, their approach to handling uncertainty and the underlying motivation differ significantly from ours, resulting in distinct techniques employed. Further details can be found in Appendix 7.7. Additionally, experiments conducted in Appendix 7.7 demonstrate that directly adapting methods from distributionally robust off-policy learning to handle inaccurately estimated logging probabilities leads to poor performance.

Off-policy learning can also be directly built on off-policy evaluation. Several work [36, 46] also propose to control the variance of the estimator caused by small logging probabilities through instance reweighing. Again, they directly use the estimated logging policy for correction, and thus performed worse than UIPS as observed in our experiments. A recent study [29] assumed additional structure in the action space and proposed the marginalized IPS. Instead, our work considers the uncertainty in the estimated logging policy and thus does not add any new assumptions about the problem space.

**Uncertainty-aware learning.** Estimation uncertainty has been extensively studied [45, 50, 1]. In the context of on-policy reinforcement leanring and bandits [1, 48, 49], the use of uncertainty aims to strike a balance between exploration and exploitation by adopting an optimistic approach (i.e., UCB in bandits). One the other hand, most reseach on offline reinforcement learning/bandits [43, 4, 6] tends to be more conservative, employing techniques such as Lower Confidence Bounds (LCB) or penalizing out-of-distribution states and actions based on uncertainty to address extrapolation errors. However, these principles differ fundamentally from UIPS, which directly minimizes the mean square error of off-policy evaluation. The closed-form solution of the resulting per-instance weight in UIPS reflects how uncertainty contributes to the policy evaluation error. Moreover, Our UIPS-O and UIPS-P baselines leverage uncertainties using the two aforementioned general principles respectively. However, empirical findings indicate that blindly penalizing or boosting samples based on uncertainty is problematic. Proper correction depends on both uncertainty in logging policy estimation and the actual value of estimated logging probabilities.

# 6   Conclusion

In this paper, we propose a Uncertainty-aware Inverse Propensity Score estimator (UIPS) to explicitly model the uncertainty of the estimated logging policy for improved off-policy learning. UIPS weighs each logged instance to reduce its policy evaluation error, where the optimal weights have a closed-form solution derived by minimizing the upper bound of the resulting estimator's mean squared error (MSE) to its ground-truth value. An improved policy is then obtained by optimizing the resulting estimation. Extensive experiments on synthetic and three real-world datasets as well as the theoretical convergence guarantee demonstrate the efficiency of UIPS.

As demonstrated in this work, explicitly modeling the uncertainty of the estimated logging policy is crucial for effective off-policy learning; but the best use of this uncertainty is not to simply down-weigh or drop instances with uncertain estimations, but to balance it with the actually estimated logging probabilities in a per-instance basis. As our future work, it is promising to investigate how UIPS can be extended to value-based learning methods, e.g., actor-critics. And on the other hand, it is also important to analyze how tight our upper bound analysis of policy evaluation error is; and if possible, find new ways to tighten it for improvements.

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

**Algorithm 1 UIPS**

---

1: **Input:** The logged dataset $D := \{(\boldsymbol{x}_n, a_n, r_{\boldsymbol{x}_n, a_n}) | n \in [N]\}$, the estimated logging policy model $\hat{\beta}(a|\boldsymbol{x}) = \frac{\exp(f_{\boldsymbol{\theta}}(\boldsymbol{x}, a))}{\sum_{a'} \exp(f_{\boldsymbol{\theta}}(\boldsymbol{x}, a'))}$, dimension $d$.
2: Initialize $\boldsymbol{M}_D = \boldsymbol{I}_{d \times d}$.
3: **for** $n = 1$ **to** $N$ **do**
4:      $\boldsymbol{M}_D = \boldsymbol{M}_D + \nabla_{\boldsymbol{\theta}} f_{\boldsymbol{\theta}}(\boldsymbol{x}_n, a_n) \nabla_{\boldsymbol{\theta}} f_{\boldsymbol{\theta}}(\boldsymbol{x}_n, a_n)^\top$         ▷ $\boldsymbol{M}_D$ for uncertainty calculation.
5: **end for**
6: **for** $n = 1$ **to** $N$ **do**
7:      $U_{\boldsymbol{x}_n, a_n} = \sqrt{\nabla_{\boldsymbol{\theta}} f_{\boldsymbol{\theta}}(\boldsymbol{x}_n, a_n)^\top \boldsymbol{M}_D^{-1} \nabla_{\boldsymbol{\theta}} f_{\boldsymbol{\theta}}(\boldsymbol{x}_n, a_n)}$
8: **end for**
9: **while** not converge **do**
10:      **for** $n = 1$ **to** $N$ **do**
11:          Calculate $\phi_{\boldsymbol{x}_n, a_n}^*$ as in Theorem 3.2
12:          Calculate gradients as in Eq.(5) and updating $\pi_{\boldsymbol{\vartheta}}(a|\boldsymbol{x})$.
13:      **end for**
14: **end while**
15: **Output:** The learnt policy $\pi_{\boldsymbol{\vartheta}}(a|\boldsymbol{x})$.

---

# 7 Appendix

## 7.1 Notations and Algorithm Framework.

For ease of reading, we list important notations in Table 5 and summarize the main framework of the proposed UIPS in Algorithm 1.

| Notation | Description |
|----------|-------------|
| $\mathcal{X}$ | context space |
| $\mathcal{A}$ | action set |
| $\boldsymbol{x} \in R^d$ | context vector |
| $a$ | action |
| $r_{\boldsymbol{x}, a}$ | reward |
| $\pi(a|\boldsymbol{x})$ | targeted policy to evaluate |
| $\beta^*(a|\boldsymbol{x})$ | the unknown ground-truth logging policy |
| $\hat{\beta}(a|\boldsymbol{x})$ | the estimated logging policy |
| $V(\pi)$ | value function |
| $D := \{(\boldsymbol{x}_n, a_n, r_{\boldsymbol{x}_n, a_n}) | n \in [N]\}$ | logged dataset containing $N$ samples |
| $\phi_{\boldsymbol{x}, a}^*$ | the optimal uncertainty-aware weight |
| $f_{\boldsymbol{\theta}^*}(\boldsymbol{x}, a)$ | the unknown ground-truth function that generates $\beta^*(a|\boldsymbol{x}) = \frac{\exp(f_{\boldsymbol{\theta}^*}(\boldsymbol{x}, a))}{\sum_{a'} \exp(f_{\boldsymbol{\theta}^*}(\boldsymbol{x}, a'))}$ |
| $f_{\boldsymbol{\theta}}(\boldsymbol{x}, a)$ | the estimate of $f_{\boldsymbol{\theta}^*}(\boldsymbol{x}, a)$ that generates $\hat{\beta}(a|\boldsymbol{x})$ |
| $\boldsymbol{B}_{\boldsymbol{x}, a}$ | confidence interval of $\hat{\beta}(a|\boldsymbol{x})$ |
| $U_{\boldsymbol{x}, a}$ | uncertainty defined as $|f_{\boldsymbol{\theta}^*}(\boldsymbol{x}, a) - f_{\boldsymbol{\theta}}(\boldsymbol{x}, a)| \leq \gamma U_{\boldsymbol{x}, a}$ |
| $\boldsymbol{g}(\boldsymbol{x}_n, a_n)$ | gradient of $f_{\boldsymbol{\theta}}(\boldsymbol{x}, a)$ regarding to the last layer. |

Table 5: Notations

**Computation Cost.** The additional computation cost of UIPS over IPS comes from two parts:

- Pre-calculate uncertainties (line 1-5 in Algorithm 1) : This part calculates uncertainty of the logging probability for each $(s, a)$ pair, and it *only needs to be executed once*. The computational cost of this step is $O(Nd^2 + d^3)$, where $O(Nd^2)$ is for calculating uncertainties in each $(s, a)$ pair and $O(d^3)$ is for matrix inverse.

- Calculate $\phi_{\boldsymbol{x}, a}^*$ during training (line 8 in Algorithm 1): It only takes $O(1)$ time, the same computational cost as calculating IPS score.

Note that calculating the logging probability for each sample, which is essential for both UIPS and IPS, takes $O(Nd|\mathcal{A}|)$ time. Since the dimension $d$ is usually much less than action size $|\mathcal{A}|$ and sample size $N$, UIPS does not introduce significant computational overhead compared to the original IPS solution.

## 7.2 Derivation of confidence interval of logging probability.

Given that $|f_{\boldsymbol{\theta}*}(\boldsymbol{x}, a) - f_{\boldsymbol{\theta}}(\boldsymbol{x}, a)| \leq \gamma U_{\boldsymbol{x},a}$ holds with probability at least $1 - \sigma$ and

$$\beta^*(a|\boldsymbol{x}) = \frac{\exp(f_{\boldsymbol{\theta}*}(\boldsymbol{x}, a))}{Z^*}, \hat{\beta}(a|\boldsymbol{x}) = \frac{\exp(f_{\boldsymbol{\theta}}(\boldsymbol{x}, a))}{\hat{Z}},$$

where $Z^* = \sum_{a'} \exp(f_{\boldsymbol{\theta}*}(a'|\boldsymbol{x}))$ and $\hat{Z} = \sum_{a'} \exp(f_{\boldsymbol{\theta}}(a'|\boldsymbol{x}))$, we can get that with probability at least $1 - \sigma$:

$$\begin{aligned}
&|f_{\boldsymbol{\theta}*}(\boldsymbol{x}, a) - f_{\boldsymbol{\theta}}(\boldsymbol{x}, a)| \leq \gamma U_{\boldsymbol{x},a}\\
\Leftrightarrow &f_{\boldsymbol{\theta}}(\boldsymbol{x}, a) - \gamma U_{\boldsymbol{x},a} \leq f_{\boldsymbol{\theta}*}(\boldsymbol{x}, a) \leq f_{\boldsymbol{\theta}}(\boldsymbol{x}, a) + \gamma U_{\boldsymbol{x},a}\\
\Leftrightarrow &\exp(f_{\boldsymbol{\theta}}(\boldsymbol{x}, a))\exp(-\gamma U_{\boldsymbol{x},a}) \leq \exp(f_{\boldsymbol{\theta}*}(\boldsymbol{x}, a)) \leq \exp(f_{\boldsymbol{\theta}}(\boldsymbol{x}, a))\exp(\gamma U_{\boldsymbol{x},a})\\
\overset{(1)}{\Leftrightarrow} &\hat{Z}\hat{\beta}(a|\boldsymbol{x})\exp(-\gamma U_{\boldsymbol{x},a}) \leq \exp(f_{\boldsymbol{\theta}*}(\boldsymbol{x}, a)) \leq \hat{Z}\hat{\beta}(a|\boldsymbol{x})\exp(\gamma U_{\boldsymbol{x},a})\\
\overset{(2)}{\Leftrightarrow} &\frac{\hat{Z}\exp(-\gamma U_{\boldsymbol{x},a})}{Z^*}\hat{\beta}(a|\boldsymbol{x}) \leq \beta^*(a|\boldsymbol{x}) \leq \frac{\hat{Z}\exp(\gamma U_{\boldsymbol{x},a})}{Z^*}\hat{\beta}(a|\boldsymbol{x})
\end{aligned}$$

The step labeled as $(1)$ is due to the modelling of $\hat{\beta}(a|\boldsymbol{x})$. And the step labeled as $(2)$ is because $Z^*$ is a positive constant independent of $\hat{\beta}(a|\boldsymbol{x})$. Thus with probability at least 1-$\delta$, we have $\beta^*(a|\boldsymbol{x}) \in \boldsymbol{B}_{\boldsymbol{x},a}$ and

$$\boldsymbol{B}_{\boldsymbol{x},a} = \left[\frac{\hat{Z}\exp(-\gamma U_{\boldsymbol{x},a})}{Z^*}\hat{\beta}(a|\boldsymbol{x}), \frac{\hat{Z}\exp(\gamma U_{\boldsymbol{x},a})}{Z^*}\hat{\beta}(a|\boldsymbol{x})\right].$$

## 7.3 Experiment details

### 7.3.1 Synthetic Data

**Data generation.** Given the ground-truth logging policy $\beta^*(a|\boldsymbol{x})$, we generate the logged dataset as follows. For each sample in train set, we first get the embedded context vector $\boldsymbol{x}$ from its original feature vector $\tilde{\boldsymbol{x}}$. We then sample an action $a$ according to $\beta^*(a|\boldsymbol{x})$, and obtain the reward $r_{\boldsymbol{x},a} = \boldsymbol{y}_{\tilde{\boldsymbol{x}},a}$, resulting bandit feedback $(\boldsymbol{x}, a, r_{\boldsymbol{x},a})$, where $\boldsymbol{y}_{\tilde{\boldsymbol{x}},a}$ is the label of class $a$ under the original feature vector $\tilde{\boldsymbol{x}}$. We repeat above process $N$ times to collect the logged dataset. In our experiments, we take $d = 64, N = 100$.

We model the logging policy as in Eq.(9), where $\{\boldsymbol{\theta}_a\}$ are the parameters to be estimated. To train the logging policy, we take all samples in the logged dataset $D$ as positive instances, and randomly sample non-selected actions as negative instances as in [8].

### 7.3.2 Real-world Data

**Statistics of datasets.** The statistics of three real-world recommendation datasets with unbiased data can be found in Table 6.

Table 6: The statistics of three real-world datasets.

| Dataset | #User | #Item | #Biased Data | #Unbiased Data |
|---------|-------|-------|--------------|----------------|
| Yahoo R3 | 15,400 | 1,000 | 311,704 | 54,000 |
| Coat | 290 | 300 | 6,960 | 4,640 |
| KuaiRec | 7,176 | 10,729 | 12,530,806 | 4,676,570 |

All these datasets contain a set of biased data collected from users' interactions on the platform, and a set of unbiased data collected from a randomized controlled trial where items are randomly selected. As in [10], on each dataset, the biased data is used for training, and the unbiased data is for testing, with a small part of unbiased data split for validation purpose (5% on Yahoo R3 and Coat,

and 15% on KuaiRec). We take the reward as 1 if : (1) the rating is larger than 3 in Yahoo! R3 and Coat datasets; (2) the user watched more than 70% of the video in KuaiRec. Otherwise, the reward is labeled as 0.

We adopted a two-tower neural network architecture to implement both the logging and learning policy, as shown in Figure 2. For the learning policy, the user representation and item representation are first modelled through two separate neural networks (i.e., the user tower and the item tower), and then their element-by-element product vector is projected to predict the user's preference for the item. We then re-use the user state generated from the user tower of the learning policy, and model the logging policy with another separate item tower, following [8]. We also block gradients to prevent the logging policy learning interfering the user state of the learning policy. In each learning epoch, we will first estimate the logging policy, and then take the estimated logging probabilities as well as their uncertainties to optimize the learning policy.

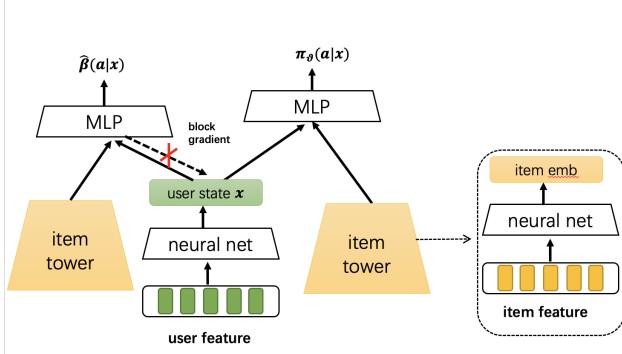

Figure 2: Model architecture of the logging and the learning policy in real-world datasets

### 7.3.3 Implementation details.

To facilitate hyper-parameter tuning, we disentangled two $\eta$s in two the terms of $\phi^*_{\boldsymbol{x},a}$ in Theorem 3.2, and introduce $\eta_1$ and $\eta_2$ to represent $\eta$ in the first and second term respectively in our implementation. This is due to the scale of $\eta$ in the first term is closely related to the scale of $\lambda$, with $\lambda/\sqrt{\eta}$ as the truly effective hyper-parameter. But the scale of $\eta$ in the second term is independent from $\lambda$.

Moreover, while $\lambda = \mathbb{E}_{\pi_{\boldsymbol{\vartheta}}}\left[r^2_{\boldsymbol{x},a}\frac{\pi_{\boldsymbol{\vartheta}}(a|\boldsymbol{x})}{\beta^*(a|\boldsymbol{x})}\right]$ depends on $\pi_\theta$ as discussed in Eq.(6), we cannot adaptively set the value of $\lambda$ since the ground-truth logging policy $\beta^*(a|\boldsymbol{x})$ is unknown. However, we have:

$$\lambda := E_\pi\left[r^2_{\boldsymbol{x},a}\frac{\pi_\theta(a|\boldsymbol{x})}{\beta^*(a|\boldsymbol{x})}\right] \leq \left(\sum_a \pi_\theta(a|\boldsymbol{x})^4\right)\left(\sum_a \frac{r^4_{\boldsymbol{x},a}}{\beta^*(a|\boldsymbol{x})^2}\right) \leq \left(\sum_a \frac{r^4_{\boldsymbol{x},a}}{\beta^*(a|\boldsymbol{x})^2}\right),$$

where the first inequality is due to the Cauchy–Schwarz inequality and the second inequality is because $\sum_a \pi_\theta(a|\boldsymbol{x})^4 \leq \sum_a \pi_\theta(a|\boldsymbol{x}) = 1$ with $\pi_\theta(a|\boldsymbol{x}) \in [0,1]$. We denote $\tilde{\lambda} = \left(\sum_a \frac{r^4_{\boldsymbol{x},a}}{\beta^*(a|\boldsymbol{x})^2}\right)$, which is dataset-specific constant and independent from $\pi_\theta$. By replacing $\lambda$ in Eq.(6) with $\tilde{\lambda}$, we can still minimize an upper bound of $\mathrm{MSE}(\hat{V}_{\mathrm{UIPS}}(\pi_{\boldsymbol{\vartheta}}))$, which ensures that the result of our analysis still holds. Thus, considering ease of computation and efficiency, we take a fixed $\lambda$ during our policy learning.

We then use grid search to select hyperparameters based on the model's performance on the validation dataset: the learning rate was searched in $\{1e^{-5}, 1e^{-4}, 1e^{-3}, 1e^{-2}\}$; $\lambda, \gamma, \eta_1$ were searched in $\{0.5, 0.1, 1, 2, 5, 10, 15, 20, 25, 30, 40, 50\}$. And $\eta_2$ was searched in $\{1, 10, 100, 1000\}$. For baseline algorithms, we also performed a similar grid search as mentioned above, and the search range follows the original papers.

**Ablation study: hyper-parameter tuning.** Although UIPS has four hyperparameters ($\lambda$, $\gamma$, $\eta_1$, and $\eta_2$), one only needs to carefully finetune two of them, i.e., $\gamma$ and $\eta_1^2/\lambda$, to obtain good performance of UIPS. This is because:

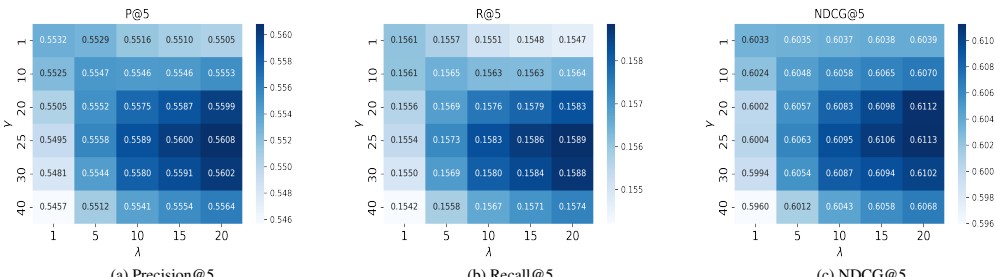

Figure 3: Effect of $\lambda$ and $\gamma$ on synthetic dataset with $\tau = 0.5$

- $\eta_2$ acts as a capping threshold to ensure $\phi^*_{\boldsymbol{x},a} \leq 2\eta_2$ holds even when the corresponding propensity scores are very low. Hence, it should be set to a reasonably large value (e.g., 100).

- The key component (i.e., the first term) of $\phi^*_{\boldsymbol{x},a}$ can be rewritten in the following way. While all $(\boldsymbol{x}, a)$ pairs will be multiplied by $\phi^*_{\boldsymbol{x},a}$, $\eta_1$ in the numerator will not affect final performance too much, and the key is to find a good value of $\eta_1^2/\lambda$ to balance the two terms in the denominator:

$$
\eta_1 / \left[ \exp\left(-\gamma U_{\boldsymbol{x},a}\right) + \frac{\eta_1^2/\lambda \cdot \pi_{\boldsymbol{\vartheta}}(a|\boldsymbol{x})^2}{\hat{\beta}(a|\boldsymbol{x})^2 \exp\left(-\gamma U_{\boldsymbol{x},a}\right)} \right].
$$

Thus with $\eta_1$ and $\eta_2$ fixed, the effect of hyper-parameter $\gamma$ and $\lambda$ on precision, recall as well as NDCG can be found in Figure 3. We can observe that to make UIPS excel, $\boldsymbol{B}_{\boldsymbol{x},a}$ needs to be of high confidence, e.g., $\gamma = 25$ performed the best on the dataset with $\tau = 0.5$. Moreover, the threshold $\sqrt{\lambda}/\eta_1$ cannot be too small or too large.

### 7.4 Experiments on the doubly robust estimators.

The doubly robust (DR) estimator [15], which is a hybrid of *direct method* (DM) estimator and *inverse propensity score* (IPS) estimator, is also widely used for off-policy evaluation. More specifically, let $\hat{\eta} : \mathcal{X} \times \mathcal{A} \rightarrow R$ be the imputation model in DM that estimates the reward of action $a$ under context vector $\boldsymbol{x}$, and $\hat{\beta}(a|\boldsymbol{x})$ be the estimated logging policy in the IPS estimator. The DR estimator evaluates policy $\pi$ based on the logged dataset $D := \{(\boldsymbol{x}_n, a_n, r_{\boldsymbol{x}_n, a_n}) | n \in [N]\}$, by:

$$
\hat{V}_{\mathrm{DR}}(\pi) = \hat{V}_{\mathrm{DM}}(\pi) + \frac{1}{N} \sum_{n=1}^{N} \frac{\pi(a_n|\boldsymbol{x}_n)}{\hat{\beta}(a_n|\boldsymbol{x}_n)} \left(r_{\boldsymbol{x}_n, a_n} - \hat{\eta}(\boldsymbol{x}_n, a_n)\right) \tag{10}
$$

where $\hat{V}_{\mathrm{DM}}(\pi)$ is the DM estimator:

$$
\hat{V}_{\mathrm{DM}}(\pi) = \frac{1}{N} \sum_{n=1}^{N} \sum_{a \in \mathcal{A}} \pi(a|\boldsymbol{x}_n) \hat{\eta}(\boldsymbol{x}_n, a). \tag{11}
$$

Again assume the policy $\pi(a|\boldsymbol{x})$ is parameterized by $\boldsymbol{\vartheta}$, the REINFORCE gradient of $\hat{V}_{\mathrm{DR}}(\pi_{\boldsymbol{\vartheta}})$ with respect to $\boldsymbol{\vartheta}$ can be readily derived as follows:

$$
\nabla_{\boldsymbol{\vartheta}} \hat{V}_{\mathrm{DR}}(\pi_{\boldsymbol{\vartheta}}) = \frac{1}{N} \sum_{n=1}^{N} \left( \sum_{a \in \mathcal{A}} \pi_{\boldsymbol{\vartheta}}(a|\boldsymbol{x}_n) \hat{\eta}(\boldsymbol{x}_n, a) \nabla_{\boldsymbol{\vartheta}} \log(\pi_{\boldsymbol{\vartheta}}(a|\boldsymbol{x}_n)) \right)
$$
$$
+ \frac{1}{N} \sum_{n=1}^{N} \left( \frac{\pi(a_n|\boldsymbol{x}_n)}{\hat{\beta}(a_n|\boldsymbol{x}_n)} \left(r_{\boldsymbol{x}_n, a_n} - \hat{\eta}(\boldsymbol{x}_n, a_n)\right) \nabla_{\boldsymbol{\vartheta}} \log(\pi_{\boldsymbol{\vartheta}}(a_n|\boldsymbol{x}_n)) \right). \tag{12}
$$

The imputation model $\hat{\eta}(\boldsymbol{x}, a)$ is pre-trained following previous work [22] with the same neural network architecture as the logging policy model. Besides the standard DR estimator, we also adapt

Table 7: Experiment results on synthetic datasets. The best and second best results are highlighted with **bold** and underline respectively. Two $p$-values are calculated: (1) $p$-value (UIPSDR): The $p$-value under the t-test between UIPSDR and the best DR baseline on each dataset; (2) $p$-value (UIPS): The $p$-value under the t-test between UIPS and the best DR baseline on each dataset.

| | $\tau = 0.5$ | | | $\tau = 1$ | | | $\tau = 2$ | | |
|---|---|---|---|---|---|---|---|---|---|
| Algorithm | P@5 | R@5 | NDCG@5 | P@5 | R@5 | NDCG@5 | P@5 | R@5 | NDCG@5 |
| BIPS-Cap | $0.5515_{\pm 2e^{-3}}$ | $0.1553_{\pm 8e^{-4}}$ | $0.6031_{\pm 2e^{-3}}$ | $0.5526_{\pm 2e^{-3}}$ | $0.1561_{\pm 6e^{-4}}$ | $0.6016_{\pm 1e^{-3}}$ | $0.5409_{\pm 3e^{-3}}$ | $0.1529_{\pm 9e^{-4}}$ | $0.5901_{\pm 2e^{-3}}$ |
| UIPS | $0.5589_{\pm 3e^{-3}}$ | $0.1583_{\pm 9e^{-4}}$ | $0.6095_{\pm 3e^{-3}}$ | $0.5572_{\pm 2e^{-3}}$ | $0.1571_{\pm 8e^{-4}}$ | $0.6074_{\pm 2e^{-3}}$ | $0.5432_{\pm 3e^{-3}}$ | $0.1534_{\pm 8e^{-4}}$ | $0.5946_{\pm 2e^{-3}}$ |
| DR | $0.3846_{\pm 3e^{-2}}$ | $0.1082_{\pm 8e^{-3}}$ | $0.4684_{\pm 3e^{-2}}$ | $0.3631_{\pm 3e^{-2}}$ | $0.1017_{\pm 9e^{-3}}$ | $0.4494_{\pm 3e^{-2}}$ | $0.3560_{\pm 3e^{-2}}$ | $0.0995_{\pm 7e^{-3}}$ | $0.4470_{\pm 3e^{-2}}$ |
| MinVarDR | $0.3212_{\pm 3e^{-2}}$ | $0.0908_{\pm 8e^{-3}}$ | $0.4062_{\pm 3e^{-2}}$ | $0.3240_{\pm 5e^{-2}}$ | $0.0903_{\pm 1e^{-2}}$ | $0.3905_{\pm 5e^{-2}}$ | $0.3234_{\pm 5e^{-2}}$ | $0.0910_{\pm 1e^{-2}}$ | $0.4059_{\pm 4e^{-2}}$ |
| ShrinkageDR | $0.4139_{\pm 2e^{-2}}$ | $0.1161_{\pm 7e^{-3}}$ | $0.4969_{\pm 3e^{-2}}$ | $0.3944_{\pm 3e^{-2}}$ | $0.1101_{\pm 8e^{-3}}$ | $0.4797_{\pm 2e^{-2}}$ | $0.4080_{\pm 3e^{-2}}$ | $0.1135_{\pm 7e^{-3}}$ | $0.4901_{\pm 2e^{-2}}$ |
| UIPSDR | **$0.4278_{\pm 2e^{-3}}$** | **$0.1200_{\pm 6e^{-3}}$** | **$0.5069_{\pm 2e^{-2}}$** | **$0.4008_{\pm 2e^{-2}}$** | **$0.1126_{\pm 7e^{-3}}$** | **$0.4847_{\pm 2e^{-2}}$** | **$0.4144_{\pm 2e^{-2}}$** | **$0.1162_{\pm 8e^{-3}}$** | **$0.4972_{\pm 2e^{-2}}$** |
| $p$-value (UIPSDR) | $2e^{-1}$ | $2e^{-1}$ | $3e^{-1}$ | $6e^{-1}$ | $4e^{-1}$ | $6e^{-1}$ | $6e^{-1}$ | $4e^{-1}$ | $5e^{-1}$ |
| $p$-value (UIPS) | $6e^{-13}$ | $4e^{-13}$ | $4e^{-12}$ | $2e^{-12}$ | $1e^{-12}$ | $5e^{-12}$ | $8e^{-12}$ | $8e^{-12}$ | $2e^{-11}$ |

Table 8: Experiment results on real-world unbiased datasets. The best and second best results are highlighted with **bold** and underline respectively. Two $p$-values are calculated: (1) $p$-value (UIPSDR): The $p$-value under the t-test between UIPSDR and the best DR baseline on each dataset; (2) $p$-value (UIPS): The $p$-value under the t-test between UIPS and the best DR baseline on each dataset.

| | Yahoo | | | Coat | | | KuaiRec | | |
|---|---|---|---|---|---|---|---|---|---|
| Algorithm | P@5 | R@5 | NDCG@5 | P@5 | R@5 | NDCG@5 | P@50 | R@50 | NDCG@50 |
| BIPS-Cap | $0.2808_{\pm 2e^{-3}}$ | $0.7576_{\pm 5e^{-3}}$ | $0.6099_{\pm 8e^{-3}}$ | $0.2758_{\pm 6e^{-3}}$ | $0.4582_{\pm 7e^{-3}}$ | $0.4399_{\pm 9e^{-3}}$ | $0.8750_{\pm 3e^{-3}}$ | $0.0238_{\pm 7e^{-5}}$ | $0.8788_{\pm 5e^{-3}}$ |
| UIPS | $0.2868_{\pm 2e^{-3}}$ | $0.7742_{\pm 5e^{-3}}$ | $0.6274_{\pm 5e^{-3}}$ | $0.2877_{\pm 3e^{-3}}$ | $0.4757_{\pm 5e^{-3}}$ | $0.4576_{\pm 8e^{-3}}$ | $0.9120_{\pm 1e^{-3}}$ | $0.0250_{\pm 5e^{-5}}$ | $0.9174_{\pm 7e^{-4}}$ |
| DR | $0.2670_{\pm 2e^{-3}}$ | $0.7174_{\pm 6e^{-3}}$ | $0.5636_{\pm 6e^{-3}}$ | $0.2884_{\pm 3e^{-3}}$ | $0.4760_{\pm 5e^{-3}}$ | $0.4541_{\pm 5e^{-3}}$ | $0.8794_{\pm 1e^{-2}}$ | $0.0240_{\pm 5e^{-4}}$ | $0.8824_{\pm 2e^{-2}}$ |
| MinVarDR | $0.2272_{\pm 5e^{-3}}$ | $0.5989_{\pm 1e^{-2}}$ | $0.4525_{\pm 1e^{-2}}$ | $0.2704_{\pm 4e^{-3}}$ | $0.4434_{\pm 9e^{-3}}$ | $0.4137_{\pm 6e^{-3}}$ | $0.8640_{\pm 7e^{-3}}$ | $0.0235_{\pm 2e^{-4}}$ | $0.8657_{\pm 7e^{-3}}$ |
| ShrinkageDR | $0.2697_{\pm 2e^{-3}}$ | $0.7226_{\pm 6e^{-3}}$ | $0.5713_{\pm 5e^{-3}}$ | $0.2895_{\pm 4e^{-3}}$ | $0.4749_{\pm 6e^{-3}}$ | $0.4526_{\pm 6e^{-3}}$ | $0.8778_{\pm 2e^{-3}}$ | $0.0239_{\pm 5e^{-4}}$ | $0.8800_{\pm 2e^{-2}}$ |
| UIPSDR | **$0.2721_{\pm 1e^{-3}}$** | **$0.7294_{\pm 6e^{-3}}$** | **$0.5750_{\pm 5e^{-3}}$** | **$0.2946_{\pm 4e^{-3}}$** | **$0.4854_{\pm 8e^{-3}}$** | **$0.4647_{\pm 8e^{-3}}$** | **$0.8849_{\pm 1e^{-2}}$** | **$0.0242_{\pm 4e^{-4}}$** | **$0.8896_{\pm 1e^{-2}}$** |
| $p$-value (UIPSDR) | $1e^{-2}$ | $2e^{-2}$ | $1e^{-1}$ | $7e^{-3}$ | $5e^{-3}$ | $2e^{-3}$ | $4e^{-1}$ | $4e^{-1}$ | $3e^{-1}$ |
| $p$-value (UIPS) | $1e^{-12}$ | $6e^{-14}$ | $6e^{-15}$ | $3e^{-1}$ | $8e^{-1}$ | $1e^{-1}$ | $2e^{-6}$ | $2e^{-6}$ | $1e^{-3}$ |

UIPS and the best two baselines on off-policy evaluation estimator (i.e., MinVar and Shrinkage based on the results in Table 1 and Table 4) to doubly robust setting using the same imputation model.

Table 7 and Table 8 report the empirical performance of the learned policy on the synthetic datasets and three real-world datasets respectively. For ease of comparison, we also include the experiment results of BIPS-Cap and UIPS on each dataset in these two tables. Two $p$-values are also provided: (1) $p$-value (UIPSDR): The $p$-value under the t-test between UIPSDR and the best DR baseline on each dataset; (2) $p$-value (UIPS): The $p$-value under the t-test between UIPS and the best DR baseline on each dataset. From Table 7 and Table 8, we can first observe that DR cannot consistently outperform BIPS-Cap: It outperformed BIPS-Cap on the Coat and KuaiRec dataset, while achieving much worse performance on the synthetic datasets and Yahoo dataset. This is because the imputation model also plays a very important role in gradient calculation as shown in Eq.(12). Its accuracy greatly affects policy learning. When the imputation model is sufficiently accurate, for example, on the Coat dataset with only 300 actions, incorporating the DM estimator not only led to better performance of DR over IPS, but also improved performance of UIPSDR over UIPS. And in particular, in this situation UIPSDR performed better than DR with the gain being statistically significant. When the imputation model is not accurate enough, for example, on the KuaiRec dataset with a large action space but sparse reward feedback, DR is still worse than UIPS, and UIPSDR also performs worse than UIPS due to the distortion of the imputation model.

## 7.5 Difference against DICE-type algorithms.

The DICE line of work [26, 47, 9] is proposed for off-policy correction in the multi-step RL setting with the environment following the Markov Decision Process (MDP) assumption. Although the DICE line of work does not require the knowledge of the logging policy, it is fundamentally different from our work. Given a logged dataset $D := \{(s_t, a_t, r_{s_t, a_t}, s_{t+1})\}$ collected from an unknown logging policy $\beta^*(a_t | s_t)$, DICE-type algorithms propose to directly estimate the discounted stationary distribution correction $w_{\pi/D}(s_t, a_t) = \frac{d^\pi(s_t, a_t)}{d^D(s_t, a_t)}$ to replace the product-based off-policy correction weight $\prod_t \frac{\pi(a_t | s_t)}{\beta^*(a_t | s_t)}$, which suffers high variance due to the series of products. While the estimation of $w_{\pi/D}(s_t, a_t)$ is agnostic to the logging policy, it highly depends on two assumptions: (1) Environment follows an MDP, i.e., $s_{t+1} \sim T(s_t, a_t)$ with $T(\cdot, \cdot)$ denoting the state transition function; (2) Each logged sample should be of a state-action-next-state tuple $(s_t, a_t, r_{s_t, a_t}, s_{t+1})$ that contains state transition information.

Table 9: Empirical performance of DICE-S on synthetic datasets.

| Algorithm | $\tau=0.5$ | | | $\tau=1$ | | | $\tau=2$ | | |
|---|---|---|---|---|---|---|---|---|---|
| | P@5 | R@5 | NDCG@5 | P@5 | R@5 | NDCG@5 | P@5 | R@5 | NDCG@5 |
| BIPS-Cap | $0.5515_{\pm 2e^{-3}}$ | $0.1553_{\pm 8e^{-4}}$ | $0.6031_{\pm 2e^{-3}}$ | $0.5526_{\pm 2e^{-3}}$ | $0.1561_{\pm 6e^{-4}}$ | $0.6016_{\pm 1e^{-3}}$ | $0.5409_{\pm 3e^{-3}}$ | $0.1529_{\pm 9e^{-4}}$ | $0.5901_{\pm 2e^{-3}}$ |
| DICE-S | $0.5416_{\pm 4e^{-3}}$ | $0.1520_{\pm 1e^{-2}}$ | $0.5968_{\pm 4e^{-3}}$ | $0.5508_{\pm 2e^{-3}}$ | $0.1553_{\pm 7e^{-4}}$ | $0.6010_{\pm 2e^{-3}}$ | $0.5403_{\pm 2e^{-3}}$ | $0.1526_{\pm 7e^{-4}}$ | $0.5903_{\pm 1e^{-3}}$ |
| UIPS | $0.5589_{\pm 3e^{-3}}$ | $0.1583_{\pm 9e^{-4}}$ | $0.6095_{\pm 3e^{-3}}$ | $0.5572_{\pm 2e^{-3}}$ | $0.1571_{\pm 8e^{-4}}$ | $0.6074_{\pm 2e^{-3}}$ | $0.5432_{\pm 3e^{-3}}$ | $0.1534_{\pm 8e^{-4}}$ | $0.5946_{\pm 2e^{-3}}$ |

Table 10: Empirical performance of DICE-S on real-world datasets.

| Algorithm | Yahoo | | | Coat | | | KuaiRec | | |
|---|---|---|---|---|---|---|---|---|---|
| | P@5 | R@5 | NDCG@5 | P@5 | R@5 | NDCG@5 | P@50 | R@50 | NDCG@50 |
| BIPS-Cap | $0.2808_{\pm 2e^{-3}}$ | $0.7576_{\pm 5e^{-3}}$ | $0.6099_{\pm 8e^{-3}}$ | $0.2758_{\pm 6e^{-3}}$ | $0.4582_{\pm 7e^{-3}}$ | $0.4399_{\pm 9e^{-3}}$ | $0.8750_{\pm 3e^{-3}}$ | $0.0238_{\pm 7e^{-5}}$ | $0.8788_{\pm 5e^{-3}}$ |
| DICE-S | $0.2618_{\pm 4e^{-3}}$ | $0.7010_{\pm 1e^{-2}}$ | $0.5627_{\pm 9e^{-3}}$ | $0.2686_{\pm 5e^{-3}}$ | $0.4422_{\pm 6e^{-3}}$ | $0.4265_{\pm 5e^{-3}}$ | $0.8842_{\pm 2e^{-3}}$ | $0.0241_{\pm 6e^{-5}}$ | $0.8908_{\pm 2e^{-3}}$ |
| UIPS | $0.2868_{\pm 2e^{-3}}$ | $0.7742_{\pm 5e^{-3}}$ | $0.6274_{\pm 5e^{-3}}$ | $0.2877_{\pm 3e^{-3}}$ | $0.4757_{\pm 5e^{-3}}$ | $0.4576_{\pm 8e^{-3}}$ | $0.9120_{\pm 1e^{-3}}$ | $0.0250_{\pm 5e^{-5}}$ | $0.9174_{\pm 7e^{-4}}$ |

We further inspected how DICE-type algorithms would work in the contextual bandit setting, which can be taken as a one-state MDP. We take DualDICE [1] as an example for illustration, and similar conclusions can be drawn for other DICE-type algorithms. We can easily derive that in the contextual bandit setting, DualDICE degenerates to the IPS estimator that approximates the unknown ground-truth logging policy $\beta^*(a|s)$ with its empirical estimate from the given logged dataset $D := \{(s_i, a_i, r_{s_i, a_i})|i \in [N]\}$, which inherits all limitations of IPS estimator, such as high variance on instances with low support in $D$.

More specifically, recall that DualDICE estimates the discounted stationary distribution correction by optimizing the following objective function (Eq.(8) in [26]):

$$\min_{x:S \times A \to C} \quad \frac{1}{2} E_{(s,a) \sim d^D}[x(s,a)^2] - E_{(s,a) \sim d^\pi}[x(s,a)],$$

with the optimizer $x^*(s,a) = w_{\pi/D}(s,a)$. In a multi-step RL setting, one cannot directly calculate the second expectation as $d^\pi(s,a)$ is inaccessible, thus DualDICE takes the change-of-variable trick to address the above optimization problem. However, in the contextual bandit setting, let $p(s)$ denote the state distribution, $d^\pi(s,a) = p(s)\pi(a|s)$ and $d^D(s,a) = p(s)\beta^*(a|s)$. The optimization problem can be rewritten as :

$$\min_{x:S \times A \to C} \quad E_{s \sim p(s)} \left[ \frac{1}{2} E_{a \sim \beta^*(a|s)}[x(s,a)^2] - E_{a \sim \pi(a|s)}[x(s,a)] \right].$$

With the logged dataset $D$, the empirical estimate of above optimization problem is :

$$\min_{x:S \times A \to C} \quad \sum_s \frac{N_s}{N} \left( \sum_a \frac{N_{s,a}}{2N_s} x(s,a)^2 - \sum_a \pi(a|s)x(s,a) \right),$$

yielding $x^*(s,a) = \pi(a|s)N_s/N_{s,a}$, where $N_s$ and $N_{s,a}$ denote the number of logged samples with $s_i = s$ and $(s_i = s, a_i = a)$ respectively. This is actually equivalent to the IPS estimator with $N_{s,a}/N_s$ as the estimate of $\beta^*(a|s)$. We refer to the above estimator as DICE-S, and Table 9 and Table 10 show the empirical performance of the policy learned through DICE-S on the synthetic and real-world datasets respectively. Similar as BIPS-Cap described in Section 4, we also clip propensity scores to control variance. One can observe from Table 9 and Table 10 that DICE-S performs worse than BIPS-Cap in all datasets except the KuaiRec dataset. Additionally, DICE-S underperforms compared to UIPS in all datasets. This is due to the fact that although DICE-S is unbiased, it only becomes accurate with numerous logged samples, and suffers high variance when logged samples are limited, resulting in its poor performance in our experiments.

## 7.6 Comparison against work on direct propensity estimation.

In addition to DICE, several other works [34, 21] propose methods to directly estimate the propensity ratio without requiring a behavior policy. These methods then use the propensity estimates to prioritize instances in the replay buffer for better TD learning. To compare its effectiveness, we also included a new baseline called IPS-LFIW, which implements the approach proposed in [1] to directly estimate the propensity ratio for off-policy learning.

The average performance and standard deviations of IPS-LFIW and UIPS on three synthetic datasets with different $\tau$ are reported in Table 11. Recall that smaller $\tau$ indicates a more skewed ground-truth logging policy. The p-value under the t-test between UIPS and IPS-LFIW is also provided to

Table 11: Empirical performance of IPS-LFIW and UIPS on synthetic datasets

| Algorithm | $\tau = 0.5$ | | | $\tau = 1$ | | | $\tau = 2$ | | |
|---|---|---|---|---|---|---|---|---|---|
| | P@5 | R@5 | NDCG@5 | P@5 | R@5 | NDCG@5 | P@5 | R@5 | NDCG@5 |
| IPS-LFIW | $0.5542_{\pm 1e^{-3}}$ | $0.1568_{\pm 4e^{-4}}$ | $0.6033_{\pm 1e^{-3}}$ | $0.5472_{\pm 1e^{-3}}$ | $0.1549_{\pm 5e^{-4}}$ | $0.5975_{\pm 1e^{-3}}$ | $0.5255_{\pm 6e^{-3}}$ | $0.1485_{\pm 1e^{-3}}$ | $0.5769_{\pm 5e^{-3}}$ |
| UIPS | $\mathbf{0.5608}_{\pm 2e^{-3}}$ | $\mathbf{0.1589}_{\pm 8e^{-4}}$ | $\mathbf{0.6113}_{\pm 3e^{-3}}$ | $\mathbf{0.5572}_{\pm 2e^{-3}}$ | $\mathbf{0.1571}_{\pm 8e^{-4}}$ | $\mathbf{0.6074}_{\pm 2e^{-3}}$ | $\mathbf{0.5432}_{\pm 3e^{-3}}$ | $\mathbf{0.1534}_{\pm 8e^{-4}}$ | $\mathbf{0.5946}_{\pm 2e^{-3}}$ |
| p-value | $2e^{-6}$ | $5e^{-6}$ | $6e^{-9}$ | $2e^{-8}$ | $2e^{-6}$ | $4e^{-10}$ | $4e^{-7}$ | $8e^{-7}$ | $1e^{-7}$ |

investigate the significance of improvements. Notably, UIPS consistently outperformed IPS-LFIW with statistically significant improvements. One major reason for the worse performance of IPS-LFIW is that it does not consider the accuracy of the estimated propensity ratio, in a direct analogy to failing to handle uncertainty in the estimated logging probabilities in existing IPS-type algorithms.

## 7.7 Comparison against distributionally robust off-policy evaluation and learning.

Our work is fundamentally different from the line of work on distributionally robust off-policy evaluation and learning [32, 17, 44]. This results in different objectives that guide the use of min-max optimization.

Specifically, the work in [32, 17, 44] assumes unknown changes exist between their training and deployment environments, such as user preference drift or unforeseen events during policy execution. Thus they choose to maximize the policy value (e.g., $\hat{V}_{\text{IPS}}(\pi)$) in the worst environment within an uncertainty set around the training environment,

$$\max_{\pi} \min_{\mathcal{U}} \quad \hat{V}_{\text{IPS}}(\pi).$$

As a result, their uncertainty set is created by introducing a small perturbation to the training environment. For example, the work in [32, 17] searches the worst environment $\mathcal{P}_{\infty}$ in the $\sigma$-close perturbed environments around the training environment $\mathcal{P}_0$ (Eq.(1) in [17]):

$$\mathcal{U}(\sigma) = \{\mathcal{P}_1 : \mathcal{P}_1 << \mathcal{P}_0 \quad \text{and} \quad D_{KL}(\mathcal{P}_1 || \mathcal{P}_0) \leq \sigma\}.$$

And the work in [44] adversarially perturbs the known ground-truth logging policy $\pi_0(\cdot|\cdot)$ in searching for the worst case (Eq.(4) in [44]):

$$\mathcal{U}(\alpha) = \{\pi_u : \max_{a,x} \quad \max\{\frac{\pi_u(a|x)}{\pi_0(a|x)}, \frac{\pi_0(a|x)}{\pi_u(a|x)}\} \leq e^{\alpha}\}.$$

In contrast, UIPS assumes the training and deployment environments stay the same, but the ground-truth logging policy is unknown. To control the high bias and high variance caused by inaccurately and small estimated logging probabilities, UIPS explicitly models the uncertainty in the estimated logging policy by incorporating a per-sample weight $\phi_{\boldsymbol{x},a}$ as discussed in Eq.(4). In order to make $\hat{V}_{\text{UIPS}}(\pi_{\boldsymbol{\vartheta}})$ as accurate as possible despite the unknown ground-truth logging policy $\beta^*(\cdot|\cdot)$, UIPS solves a min-max optimization problem in Eq.(8). This optimization problem seeks to find the optimal $\phi_{\boldsymbol{x},a}$ that minimizes the upper bound of the mean squared error (MSE) of $\hat{V}_{\text{UIPS}}$ to its ground-truth value, within an uncertainty set of the unknown ground-truth logging policy $\beta^*(\cdot|\cdot)$. The closed-form solution for the min-max optimization is also derived as in Theorem 3.2.

Furthermore, observing that work in [44] also performs optimization over an uncertainty set of the logging policy, we further adapted their method to handle the inaccuracy of the estimated logging policy, by taking $\pi_0(\cdot|\cdot)$ as the estimated logging policy, i.e., $\pi_0(\cdot|\cdot) = \hat{\beta}(\cdot|\cdot)$. We name the adapted methods as IPS-UN. Table 12 demonstrates the performance of the learned policy under IPS-UN on three synthetic datasets. The results suggest that directly applying IPS-UN to handle inaccurately estimated logging probabilities is not be a feasible solution to our problem. One important reason for the worse performance of IPS-UN is that it strives to optimize for the worst potential environment, which might not be the case in our experiment datatsets. On the other hand, UIPS assumes the training and deployment environments stay same and strives to identify the optimal policy with an unknown ground-truth logging policy.

## 7.8 Theoretical Proofs.

**Proof of Proposition 2.1:**

Table 12: Empirical performance of IPS-UN on synthetic datasets.

| | $\tau = 0.5$ | | | $\tau = 1$ | | | $\tau = 2$ | | |
|---|---|---|---|---|---|---|---|---|---|
| Algorithm | P@5 | R@5 | NDCG@5 | P@5 | R@5 | NDCG@5 | P@5 | R@5 | NDCG@5 |
| BIPS-Cap | $0.5515_{\pm 2e^{-3}}$ | $0.1553_{\pm 8e^{-4}}$ | $0.6031_{\pm 2e^{-3}}$ | $0.5526_{\pm 2e^{-3}}$ | $0.1561_{\pm 6e^{-4}}$ | $0.6016_{\pm 1e^{-3}}$ | $0.5409_{\pm 3e^{-3}}$ | $0.1529_{\pm 9e^{-4}}$ | $0.5901_{\pm 2e^{-3}}$ |
| IPS-UN | $0.4089_{\pm 3e^{-2}}$ | $0.1152_{\pm 8e^{-3}}$ | $0.4916_{\pm 2e^{-2}}$ | $0.3911_{\pm 3e^{-3}}$ | $0.1100_{\pm 9e^{-3}}$ | $0.4754_{\pm 3e^{-2}}$ | $0.3599_{\pm 4e^{-2}}$ | $0.1009_{\pm 1e^{-2}}$ | $0.4353_{\pm 4e^{-2}}$ |
| UIPS | $0.5589_{\pm 3e^{-3}}$ | $0.1583_{\pm 9e^{-4}}$ | $0.6095_{\pm 3e^{-3}}$ | $0.5572_{\pm 2e^{-3}}$ | $0.1571_{\pm 8e^{-4}}$ | $0.6074_{\pm 2e^{-3}}$ | $0.5432_{\pm 3e^{-3}}$ | $0.1534_{\pm 8e^{-4}}$ | $0.5946_{\pm 2e^{-3}}$ |

*Proof.* Because of the linearity of expectation, we have $\mathbb{E}_D\left[\hat{V}_{\text{BIPS}}(\pi_{\boldsymbol{\vartheta}})\right] = \mathbb{E}_{\beta^*}\left[\frac{\pi_{\boldsymbol{\vartheta}}(a|\boldsymbol{x})}{\hat{\beta}(a|\boldsymbol{x})}r_{\boldsymbol{x},a}\right]$, and thus:

$$
\begin{aligned}
\text{Bias}\left(\hat{V}_{\text{BIPS}}(\pi_{\boldsymbol{\vartheta}})\right) &= \mathbb{E}_D\left[\hat{V}_{\text{BIPS}}(\pi_{\boldsymbol{\vartheta}}) - V(\pi_{\boldsymbol{\vartheta}})\right] \\
&= \mathbb{E}_{\beta^*}\left[\frac{\pi_{\boldsymbol{\vartheta}}(a|\boldsymbol{x})}{\hat{\beta}(a|\boldsymbol{x})}r_{\boldsymbol{x},a}\right] - \mathbb{E}_{\beta^*}\left[\frac{\pi_{\boldsymbol{\vartheta}}(a|\boldsymbol{x})}{\beta^*(a|\boldsymbol{x})}r_{\boldsymbol{x},a}\right] \\
&= \mathbb{E}_{\beta^*}\left[\frac{\pi_{\boldsymbol{\vartheta}}(a|\boldsymbol{x})}{\beta^*(a|\boldsymbol{x})}r_{\boldsymbol{x},a}\left(\frac{\beta^*(a|\boldsymbol{x})}{\hat{\beta}(a|\boldsymbol{x})}-1\right)\right] \\
&= \mathbb{E}_{\pi_{\boldsymbol{\vartheta}}}\left[r_{\boldsymbol{x},a}\left(\frac{\beta^*(a|\boldsymbol{x})}{\hat{\beta}(a|\boldsymbol{x})}-1\right)\right].
\end{aligned}
\tag{13}
$$

Since samples are independently sampled from logging policy, the variance can be computed as:

$$
\text{Var}_D\left(\hat{V}_{\text{BIPS}}(\pi_{\boldsymbol{\vartheta}})\right) = \frac{1}{N}\text{Var}_{\beta^*}\left(\frac{\pi_{\boldsymbol{\vartheta}}(a|\boldsymbol{x})}{\hat{\beta}(a|\boldsymbol{x})}r_{\boldsymbol{x},a}\right).
$$

By re-scaling, we get:

$$
N \cdot \text{Var}_D\left(\hat{V}_{\text{BIPS}}(\pi_{\boldsymbol{\vartheta}})\right) = \text{Var}_{\beta^*}\left(\frac{\pi_{\boldsymbol{\vartheta}}(a|\boldsymbol{x})}{\hat{\beta}(a|\boldsymbol{x})}r_{\boldsymbol{x},a}\right)
\tag{14}
$$

$$
= \mathbb{E}_{\beta^*}\left[\frac{\pi_{\boldsymbol{\vartheta}}(a|\boldsymbol{x})^2}{\hat{\beta}(a|\boldsymbol{x})^2}r_{\boldsymbol{x},a}^2\right] - \left(\mathbb{E}_{\beta^*}\left[\frac{\pi_{\boldsymbol{\vartheta}}(a|\boldsymbol{x})}{\hat{\beta}(a|\boldsymbol{x})}r_{\boldsymbol{x},a}\right]\right)^2
$$

$$
= \mathbb{E}_{\pi_{\boldsymbol{\vartheta}}}\left[\frac{\pi_{\boldsymbol{\vartheta}}(a|\boldsymbol{x})}{\beta^*(a|\boldsymbol{x})}\cdot\frac{\beta^*(a|\boldsymbol{x})^2}{\hat{\beta}(a|\boldsymbol{x})^2}r_{\boldsymbol{x},a}^2\right] - \left(\mathbb{E}_{\pi_{\boldsymbol{\vartheta}}}\left[\frac{\beta^*(a|\boldsymbol{x})}{\hat{\beta}(a|\boldsymbol{x})}r_{\boldsymbol{x},a}\right]\right)^2
$$

This completes the proof. $\qquad\square$

**Proof of Theorem 3.1:**

*Proof.* We can get:

$$
\begin{aligned}
\text{MSE}\left(\hat{V}_{\text{UIPS}}(\pi_{\boldsymbol{\vartheta}})\right) &= \mathbb{E}_D\left[\left(\hat{V}_{\text{UIPS}}(\pi_{\boldsymbol{\vartheta}}) - V(\pi_{\boldsymbol{\vartheta}})\right)^2\right] \\
&= \left(\mathbb{E}_D\left[\hat{V}_{\text{UIPS}}(\pi_{\boldsymbol{\vartheta}}) - V(\pi_{\boldsymbol{\vartheta}})\right]\right)^2 + \text{Var}_D\left(\hat{V}_{\text{UIPS}}(\pi_{\boldsymbol{\vartheta}}) - V(\pi_{\boldsymbol{\vartheta}})\right) \\
&= \left(\mathbb{E}_D\left[\hat{V}_{\text{UIPS}}(\pi_{\boldsymbol{\vartheta}}) - V(\pi_{\boldsymbol{\vartheta}})\right]\right)^2 + \text{Var}_D\left(\hat{V}_{\text{UIPS}}(\pi_{\boldsymbol{\vartheta}})\right) \\
&= \text{Bias}(\hat{V}_{\text{UIPS}}(\pi_{\boldsymbol{\vartheta}}))^2 + \text{Var}(\hat{V}_{\text{UIPS}}(\pi_{\boldsymbol{\vartheta}})).
\end{aligned}
$$

We first bound the bias term:

$$\text{Bias}(\hat{V}_{\text{UIPS}}(\pi_{\boldsymbol{\vartheta}})) = \mathbb{E}_D\left[\hat{V}_{\text{UIPS}}(\pi_{\boldsymbol{\vartheta}}) - V(\pi_{\boldsymbol{\vartheta}})\right]$$

$$\overset{(1)}{=} \mathbb{E}_{\beta^*}\left[\frac{\pi_{\boldsymbol{\vartheta}}(a|\boldsymbol{x})}{\hat{\beta}(a|\boldsymbol{x})}\phi_{\boldsymbol{x},a}r_{\boldsymbol{x},a}\right] - V(\pi_{\boldsymbol{\vartheta}})$$

$$= \mathbb{E}_{\beta^*}\left[\frac{\pi_{\boldsymbol{\vartheta}}(a|\boldsymbol{x})}{\hat{\beta}(a|\boldsymbol{x})}\phi_{\boldsymbol{x},a}r_{\boldsymbol{x},a} - \frac{\pi_{\boldsymbol{\vartheta}}(a|\boldsymbol{x})}{\beta^*(a|\boldsymbol{x})}r_{\boldsymbol{x},a}\right]$$

$$= \mathbb{E}_{\beta^*}\left[r_{\boldsymbol{x},a}\frac{\pi_{\boldsymbol{\vartheta}}(a|\boldsymbol{x})}{\beta^*(a|\boldsymbol{x})}\cdot\left(\frac{\beta^*(a|\boldsymbol{x})}{\hat{\beta}(a|\boldsymbol{x})}\phi_{\boldsymbol{x},a} - 1\right)\right]$$

$$\overset{(2)}{\leq} \sqrt{\mathbb{E}_{\pi_{\boldsymbol{\vartheta}}}\left[r^2_{\boldsymbol{x},a}\frac{\pi_{\boldsymbol{\vartheta}}(a|\boldsymbol{x})}{\beta^*(a|\boldsymbol{x})}\right]} \cdot \sqrt{\mathbb{E}_{\beta^*}\left[\left(\frac{\beta^*(a|\boldsymbol{x})}{\hat{\beta}(a|\boldsymbol{x})}\phi_{\boldsymbol{x},a} - 1\right)^2\right]}$$

Step (1) follows the linearity of expectation and step (2) is due to the Cauchy-Schwarz inequality. We then bound the variance term:

$$\text{Var}(\hat{V}_{\text{UIPS}}(\pi_{\boldsymbol{\vartheta}})) = \frac{1}{N}\text{Var}_{\beta^*}\left(\frac{\pi_{\boldsymbol{\vartheta}}(a|\boldsymbol{x})}{\hat{\beta}(a|\boldsymbol{x})}\phi_{\boldsymbol{x},a}r_{\boldsymbol{x},a}\right)$$

$$= \frac{1}{N}\left(\mathbb{E}_{\beta^*}\left[\frac{\pi_{\boldsymbol{\vartheta}}(a|\boldsymbol{x})^2}{\hat{\beta}(a|\boldsymbol{x})^2}\phi^2_{\boldsymbol{x},a}r^2_{\boldsymbol{x},a}\right] - \left(\mathbb{E}_{\beta^*}\left[\frac{\pi_{\boldsymbol{\vartheta}}(a|\boldsymbol{x})}{\hat{\beta}(a|\boldsymbol{x})}\phi_{\boldsymbol{x},a}r_{\boldsymbol{x},a}\right]\right)^2\right)$$

$$\leq \frac{1}{N}\mathbb{E}_{\beta^*}\left[\frac{\pi_{\boldsymbol{\vartheta}}(a|\boldsymbol{x})^2}{\hat{\beta}(a|\boldsymbol{x})^2}\phi^2_{\boldsymbol{x},a}r^2_{\boldsymbol{x},a}\right] \leq \mathbb{E}_{\beta^*}\left[\frac{\pi_{\boldsymbol{\vartheta}}(a|\boldsymbol{x})^2}{\hat{\beta}(a|\boldsymbol{x})^2}\phi^2_{\boldsymbol{x},a}\right]$$

Combining the bound of bias and variance completes the proof. $\qquad\square$

**Proof of Theorem 3.2:**

*Proof.* We first define several notations:

- $T(\phi_{\boldsymbol{x},a}, \beta_{\boldsymbol{x},a}) = \lambda\mathbb{E}_{\beta^*}\left[\left(\frac{\beta_{\boldsymbol{x},a}}{\hat{\beta}(a|\boldsymbol{x})}\phi_{\boldsymbol{x},a} - 1\right)^2\right] + \mathbb{E}_{\beta^*}\left[\frac{\pi_{\boldsymbol{\vartheta}}(a|\boldsymbol{x})^2}{\hat{\beta}(a|\boldsymbol{x})^2}\phi^2_{\boldsymbol{x},a}\right]$.

- $\tilde{T}(\phi_{\boldsymbol{x},a}) = \max_{\beta_{\boldsymbol{x},a}\in\boldsymbol{B}_{\boldsymbol{x},a}} T(\phi_{\boldsymbol{x},a}, \beta_{\boldsymbol{x},a})$ denotes the maximum value of inner optimization problem.

- $T^* = \min_{\phi_{\boldsymbol{x},a}}\tilde{T}(\phi_{\boldsymbol{x},a}) = \min_{\phi_{\boldsymbol{x},a}}\max_{\beta_{\boldsymbol{x},a}\in\boldsymbol{B}_{\boldsymbol{x},a}} T(\phi_{\boldsymbol{x},a}, \beta_{\boldsymbol{x},a})$ denote the optimal min-max value. And $\phi^*_{\boldsymbol{x},a} = \arg\min_{\phi_{\boldsymbol{x},a}}\tilde{T}(\phi_{\boldsymbol{x},a})$ .

- $\boldsymbol{B}^-_{\boldsymbol{x},a} := \frac{\hat{Z}\exp(-\gamma U_{\boldsymbol{x},a})}{Z^*}\hat{\beta}(a|\boldsymbol{x})$, and $\boldsymbol{B}^+_{\boldsymbol{x},a} := \frac{\hat{Z}\exp(\gamma U_{\boldsymbol{x},a})}{Z^*}\hat{\beta}(a|\boldsymbol{x})$.

We first find the maximum value of the inner optimization problem, i.e., $\tilde{T}(\phi_{\boldsymbol{x},a})$ for any fixed $\phi_{\boldsymbol{x},a}$. And there are three cases shown in Figure 4:

**Case I:** When $\frac{\hat{\beta}(a|\boldsymbol{x})}{\phi_{\boldsymbol{x},a}} \geq \boldsymbol{B}^+_{\boldsymbol{x},a}$, , $\tilde{T}(\phi_{\boldsymbol{x},a})$ achieves the maximum value at $\beta_{\boldsymbol{x},a} = \boldsymbol{B}^-_{\boldsymbol{x},a}$. In other words, $\tilde{T}(\phi_{\boldsymbol{x},a}) = T(\phi_{\boldsymbol{x},a}, \boldsymbol{B}^-_{\boldsymbol{x},a})$ when $\phi_{\boldsymbol{x},a} \leq \frac{\hat{\beta}(a|\boldsymbol{x})}{\boldsymbol{B}^+_{\boldsymbol{x},a}}$.

**Case II:** When $\boldsymbol{B}^-_{\boldsymbol{x},a} \leq \frac{\hat{\beta}(a|\boldsymbol{x})}{\phi_{\boldsymbol{x},a}} \leq \boldsymbol{B}^+_{\boldsymbol{x},a}$, i.e., $\frac{Z^*\exp(-\gamma U_{\boldsymbol{x},a})}{\hat{Z}} \leq \phi_{\boldsymbol{x},a} \leq \frac{Z^*\exp(\gamma U_{\boldsymbol{x},a})}{\hat{Z}}$, then $\tilde{T}(\phi_{\boldsymbol{x},a})$ will be the maximum between $T(\phi_{\boldsymbol{x},a}, \boldsymbol{B}^-_{\boldsymbol{x},a})$ and $T(\phi_{\boldsymbol{x},a}, \boldsymbol{B}^+_{\boldsymbol{x},a})$.

More specifically, when $\frac{\hat{\beta}(a|\boldsymbol{x})}{\phi_{\boldsymbol{x},a}} \leq \frac{\boldsymbol{B}^+_{\boldsymbol{x},a}+\boldsymbol{B}^-_{\boldsymbol{x},a}}{2}$, i.e., $\phi_{\boldsymbol{x},a} \geq \frac{2\hat{\beta}(a|\boldsymbol{x})}{\boldsymbol{B}^+_{\boldsymbol{x},a}+\boldsymbol{B}^-_{\boldsymbol{x},a}}$, $\tilde{T}(\phi_{\boldsymbol{x},a}) = T(\phi_{\boldsymbol{x},a}, \boldsymbol{B}^+_{\boldsymbol{x},a})$. Otherwise when $\phi_{\boldsymbol{x},a} < \frac{2\hat{\beta}(a|\boldsymbol{x})}{\boldsymbol{B}^+_{\boldsymbol{x},a}+\boldsymbol{B}^-_{\boldsymbol{x},a}}$, $\tilde{T}(\phi_{\boldsymbol{x},a}) = T(\phi_{\boldsymbol{x},a}, \boldsymbol{B}^-_{\boldsymbol{x},a})$.

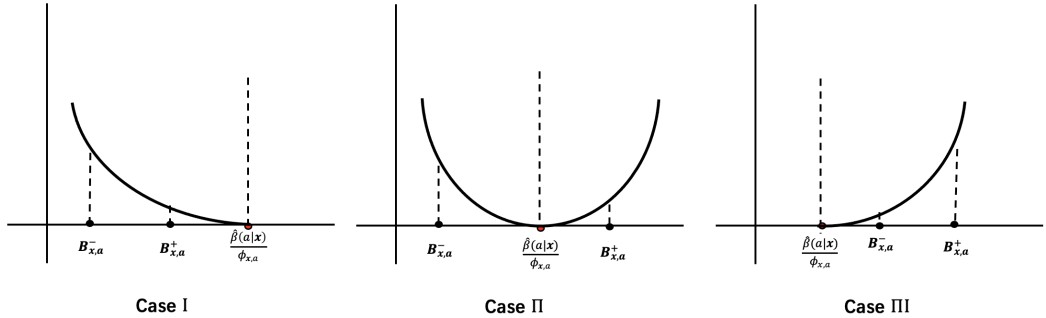

Figure 4: Three cases for maximizing the inner optimization problem.

**Case III:** When $\phi_{\boldsymbol{x},a} \geq \frac{\hat{\beta}(a|\boldsymbol{x})}{\boldsymbol{B}_{\boldsymbol{x},a}^-}$. implying $\frac{\hat{\beta}(a|\boldsymbol{x})}{\phi_{\boldsymbol{x},a}} \leq \boldsymbol{B}_{\boldsymbol{x},a}^-$, $\tilde{T}(\phi_{\boldsymbol{x},a}) = T(\phi_{\boldsymbol{x},a}, \boldsymbol{B}_{\boldsymbol{x},a}^+)$.

Overall, we get that:

$$\tilde{T}(\phi_{\boldsymbol{x},a}) = \begin{cases} T(\phi_{\boldsymbol{x},a}, \boldsymbol{B}_{\boldsymbol{x},a}^-), & \phi_{\boldsymbol{x},a} \in (-\infty, \frac{2\hat{\beta}(a|\boldsymbol{x})}{\boldsymbol{B}_{\boldsymbol{x},a}^+ + \boldsymbol{B}_{\boldsymbol{x},a}^-}] \\ T(\phi_{\boldsymbol{x},a}, \boldsymbol{B}_{\boldsymbol{x},a}^+) & \phi_{\boldsymbol{x},a} \in [\frac{2\hat{\beta}(a|\boldsymbol{x})}{\boldsymbol{B}_{\boldsymbol{x},a}^+ + \boldsymbol{B}_{\boldsymbol{x},a}^-}, \infty) \end{cases} \tag{15}$$

Next we try to find the minimum value of $\tilde{T}(\phi_{\boldsymbol{x},a})$. We first observe that without considering constraint on $\phi_{\boldsymbol{x},a}$, when

$$\phi_{\boldsymbol{x},a}^+ = \frac{\lambda}{\lambda \frac{\boldsymbol{B}_{\boldsymbol{x},a}^+}{\hat{\beta}(a|\boldsymbol{x})} + \frac{\pi_{\vartheta}(a|\boldsymbol{x})^2}{\hat{\beta}(a|\boldsymbol{x})\boldsymbol{B}_{\boldsymbol{x},a}^+}}$$

$T(\phi_{\boldsymbol{x},a}, \boldsymbol{B}_{\boldsymbol{x},a}^+)$ achieves the global minimum value. However, $\phi_{\boldsymbol{x},a}^+ \leq \frac{\hat{\beta}(a|\boldsymbol{x})}{\boldsymbol{B}_{\boldsymbol{x},a}^+} \leq \frac{2\hat{\beta}(a|\boldsymbol{x})}{\boldsymbol{B}_{\boldsymbol{x},a}^+ + \boldsymbol{B}_{\boldsymbol{x},a}^-}$, which implies when $\phi_{\boldsymbol{x},a} \in \left[\frac{2\hat{\beta}(a|\boldsymbol{x})}{\boldsymbol{B}_{\boldsymbol{x},a}^+ + \boldsymbol{B}_{\boldsymbol{x},a}^-}, \infty\right)$, the minimum value of $T(\phi_{\boldsymbol{x},a}, \boldsymbol{B}_{\boldsymbol{x},a}^+)$ is achieved at $\frac{2\hat{\beta}(a|\boldsymbol{x})}{\boldsymbol{B}_{\boldsymbol{x},a}^+ + \boldsymbol{B}_{\boldsymbol{x},a}^-}$.

On the other hand, without considering any constraint on $\phi_{\boldsymbol{x},a}$, the global minimum value of $T(\phi_{\boldsymbol{x},a}, \boldsymbol{B}_{\boldsymbol{x},a}^-)$ is achieved at:

$$\phi_{\boldsymbol{x},a}^- = \frac{\lambda}{\lambda \frac{\boldsymbol{B}_{\boldsymbol{x},a}^-}{\hat{\beta}(a|\boldsymbol{x})} + \frac{\pi_{\vartheta}(a|\boldsymbol{x})^2}{\hat{\beta}(a|\boldsymbol{x})\boldsymbol{B}_{\boldsymbol{x},a}^-}}. \tag{16}$$

Thus if $\phi_{\boldsymbol{x},a}^- \leq \frac{2\hat{\beta}(a|\boldsymbol{x})}{\boldsymbol{B}_{\boldsymbol{x},a}^+ + \boldsymbol{B}_{\boldsymbol{x},a}^-}$, $\phi_{\boldsymbol{x},a}^* = \phi_{\boldsymbol{x},a}^-$, since $T(\phi_{\boldsymbol{x},a}^-, \boldsymbol{B}_{\boldsymbol{x},a}^-) \leq T(\frac{2\hat{\beta}(a|\boldsymbol{x})}{\boldsymbol{B}_{\boldsymbol{x},a}^+ + \boldsymbol{B}_{\boldsymbol{x},a}^-}, \boldsymbol{B}_{\boldsymbol{x},a}^-) = T(\frac{2\hat{\beta}(a|\boldsymbol{x})}{\boldsymbol{B}_{\boldsymbol{x},a}^+ + \boldsymbol{B}_{\boldsymbol{x},a}^-}, \boldsymbol{B}_{\boldsymbol{x},a}^+)$. Otherwise, when $\phi_{\boldsymbol{x},a}^- > \frac{2\hat{\beta}(a|\boldsymbol{x})}{\boldsymbol{B}_{\boldsymbol{x},a}^+ + \boldsymbol{B}_{\boldsymbol{x},a}^-}$, the minimum value of $T(\phi_{\boldsymbol{x},a}, \boldsymbol{B}_{\boldsymbol{x},a}^-)$ is also achieved at $\frac{2\hat{\beta}(a|\boldsymbol{x})}{\boldsymbol{B}_{\boldsymbol{x},a}^+ + \boldsymbol{B}_{\boldsymbol{x},a}^-}$, implying $\phi_{\boldsymbol{x},a}^* = \frac{2\hat{\beta}(a|\boldsymbol{x})}{\boldsymbol{B}_{\boldsymbol{x},a}^+ + \boldsymbol{B}_{\boldsymbol{x},a}^-}$.

Overall,

$$\phi_{\boldsymbol{x},a}^* = \min\left(\frac{\lambda}{\lambda \frac{\boldsymbol{B}_{\boldsymbol{x},a}^-}{\hat{\beta}(a|\boldsymbol{x})} + \frac{\pi_{\vartheta}(a|\boldsymbol{x})^2}{\hat{\beta}(a|\boldsymbol{x})\boldsymbol{B}_{\boldsymbol{x},a}^-}}, \frac{2\hat{\beta}(a|\boldsymbol{x})}{\boldsymbol{B}_{\boldsymbol{x},a}^+ + \boldsymbol{B}_{\boldsymbol{x},a}^-}\right) \tag{17}$$

Let $\eta = \frac{Z^*}{\hat{Z}}$, we can get

$$\phi_{\boldsymbol{x},a}^* = \min\left(\frac{\lambda}{\frac{\lambda}{\eta} \exp\left(-\gamma U_{\boldsymbol{x},a}\right) + \frac{\eta \pi_{\vartheta}(a|\boldsymbol{x})^2}{\hat{\beta}(a|\boldsymbol{x})^2 \exp(-\gamma U_{\boldsymbol{x},a})}}, \frac{2\eta}{\exp\left(\gamma U_{\boldsymbol{x},a}\right) + \exp\left(-\gamma U_{\boldsymbol{x},a}\right)}\right). \tag{18}$$

Note that $\eta \in [\exp(-\gamma U_s^{\max}), \exp(\gamma U_s^{\max})]$, since $\hat{Z}\exp(-\gamma U_s^{\max}) \leq Z^* = \sum_{a'}\exp(f_{\theta^*}(a'|\boldsymbol{x})) \leq \hat{Z}\exp(U_s^{\max})$.

This completes the proof . $\qquad\square$

**Proof of Corollary 3.3:**

*Proof.* Following the similar procedure as in Theorem 3.1, we can derive the mean squared error (MSE) between $\hat{V}_{\mathrm{BIPS}}(\pi_{\boldsymbol{\vartheta}})$ and ground-truth estimator $V(\pi_{\boldsymbol{\vartheta}})$ is upper bounded as follows:

$$\mathrm{MSE}\left(\hat{V}_{\mathrm{BIPS}}(\pi_{\boldsymbol{\vartheta}})\right) = \mathbb{E}_D\left[\left(\hat{V}_{\mathrm{BIPS}}(\pi_{\boldsymbol{\vartheta}}) - V(\pi_{\boldsymbol{\vartheta}})\right)^2\right]$$

$$\leq \mathbb{E}_{\pi_{\boldsymbol{\vartheta}}}\left[r_{\boldsymbol{x},a}^2\frac{\pi_{\boldsymbol{\vartheta}}(a|\boldsymbol{x})}{\beta^*(a|\boldsymbol{x})}\right]\cdot\mathbb{E}_{\beta^*}\left[\left(\frac{\beta^*(a|\boldsymbol{x})}{\hat{\beta}(a|\boldsymbol{x})} - 1\right)^2\right] + \mathbb{E}_{\beta^*}\left[\frac{\pi_{\boldsymbol{\vartheta}}(a|\boldsymbol{x})^2}{\hat{\beta}(a|\boldsymbol{x})^2}\right].$$

When $\beta^*(a|\boldsymbol{x}) \in [\boldsymbol{B}_{\boldsymbol{x},a}^-, \boldsymbol{B}_{\boldsymbol{x},a}^+]$, with $\boldsymbol{B}_{\boldsymbol{x},a}^- := \frac{\hat{Z}\exp(-\gamma U_{\boldsymbol{x},a})}{Z^*}\hat{\beta}(a|\boldsymbol{x})$ and $\boldsymbol{B}_{\boldsymbol{x},a}^+ := \frac{\hat{Z}\exp(\gamma U_{\boldsymbol{x},a})}{Z^*}\hat{\beta}(a|\boldsymbol{x})$, $\mathrm{MSE}\left(\hat{V}_{\mathrm{BIPS}}(\pi_{\boldsymbol{\vartheta}})\right)$ can be further upper bounded as follows.

For ease of illustration, we set $\lambda = \mathbb{E}_{\pi_{\boldsymbol{\vartheta}}}\left[r_{\boldsymbol{x},a}^2\frac{\pi_{\boldsymbol{\vartheta}}(a|\boldsymbol{x})}{\beta^*(a|\boldsymbol{x})}\right]$ and

$$T(\phi_{\boldsymbol{x},a},\beta_{\boldsymbol{x},a}) = \lambda\mathbb{E}_{\beta^*}\left[\left(\frac{\beta_{\boldsymbol{x},a}}{\hat{\beta}(a|\boldsymbol{x})}\phi_{\boldsymbol{x},a} - 1\right)^2\right] + \mathbb{E}_{\beta^*}\left[\frac{\pi_{\boldsymbol{\vartheta}}(a|\boldsymbol{x})^2}{\hat{\beta}(a|\boldsymbol{x})^2}\phi_{\boldsymbol{x},a}^2\right].$$

Let $T_{\mathrm{BIPS}}^*$ denote the upper bound of $\mathrm{MSE}\left(\hat{V}_{\mathrm{BIPS}}(\pi_{\boldsymbol{\vartheta}})\right)$, then we can derive that

$$T_{\mathrm{BIPS}}^* = \begin{cases} T(1, \boldsymbol{B}_{\boldsymbol{x},a}^+), & \hat{\beta}(a|\boldsymbol{x}) \leq \frac{\boldsymbol{B}_{\boldsymbol{x},a}^+ + \boldsymbol{B}_{\boldsymbol{x},a}^-}{2} \\ T(1, \boldsymbol{B}_{\boldsymbol{x},a}^-), & \hat{\beta}(a|\boldsymbol{x}) > \frac{\boldsymbol{B}_{\boldsymbol{x},a}^+ + \boldsymbol{B}_{\boldsymbol{x},a}^-}{2}. \end{cases} \tag{19}$$

Recall that in Theorem 3.2, we show that the upper bound of $\mathrm{MSE}\left(\hat{V}_{\mathrm{UIPS}}(\pi_{\boldsymbol{\vartheta}})\right)$ is as follows:

$$T_{\mathrm{UIPS}}^* = \begin{cases} T(\phi_{\boldsymbol{x},a}^-, \boldsymbol{B}_{\boldsymbol{x},a}^-), & \phi_{\boldsymbol{x},a}^- < \frac{2\hat{\beta}(a|\boldsymbol{x})}{\boldsymbol{B}_{\boldsymbol{x},a}^+ + \boldsymbol{B}_{\boldsymbol{x},a}^-} \\ T(\frac{2\hat{\beta}(a|\boldsymbol{x})}{\boldsymbol{B}_{\boldsymbol{x},a}^+ + \boldsymbol{B}_{\boldsymbol{x},a}^-}, \boldsymbol{B}_{\boldsymbol{x},a}^+), & \phi_{\boldsymbol{x},a}^- \geq \frac{2\hat{\beta}(a|\boldsymbol{x})}{\boldsymbol{B}_{\boldsymbol{x},a}^+ + \boldsymbol{B}_{\boldsymbol{x},a}^-}. \end{cases} \tag{20}$$

where $\phi_{\boldsymbol{x},a}^-$ is defined in Eq.(16). Next we show that $T_{\mathrm{UIPS}}^* \geq T_{\mathrm{BIPS}}^*$.

- When $\hat{\beta}(a|\boldsymbol{x}) \leq \frac{\boldsymbol{B}_{\boldsymbol{x},a}^+ + \boldsymbol{B}_{\boldsymbol{x},a}^-}{2}$, we can get $T_{\mathrm{BIPS}}^* = T(1, \boldsymbol{B}_{\boldsymbol{x},a}^+)$ and $\frac{2\hat{\beta}(a|\boldsymbol{x})}{\boldsymbol{B}_{\boldsymbol{x},a}^+ + \boldsymbol{B}_{\boldsymbol{x},a}^-} \leq 1$. If $\phi_{\boldsymbol{x},a}^- < \frac{2\hat{\beta}(a|\boldsymbol{x})}{\boldsymbol{B}_{\boldsymbol{x},a}^+ + \boldsymbol{B}_{\boldsymbol{x},a}^-}$, then $T_{\mathrm{UIPS}}^* = T(\phi_{\boldsymbol{x},a}^-, \boldsymbol{B}_{\boldsymbol{x},a}^-) < T(\frac{2\hat{\beta}(a|\boldsymbol{x})}{\boldsymbol{B}_{\boldsymbol{x},a}^+ + \boldsymbol{B}_{\boldsymbol{x},a}^-}, \boldsymbol{B}_{\boldsymbol{x},a}^-) = T(\frac{2\hat{\beta}(a|\boldsymbol{x})}{\boldsymbol{B}_{\boldsymbol{x},a}^+ + \boldsymbol{B}_{\boldsymbol{x},a}^-}, \boldsymbol{B}_{\boldsymbol{x},a}^+) \leq T(1, \boldsymbol{B}_{\boldsymbol{x},a}^+)$. And if $\phi_{\boldsymbol{x},a}^- \geq \frac{2\hat{\beta}(a|\boldsymbol{x})}{\boldsymbol{B}_{\boldsymbol{x},a}^+ + \boldsymbol{B}_{\boldsymbol{x},a}^-}$, $T_{\mathrm{UIPS}}^* = T(\frac{2\hat{\beta}(a|\boldsymbol{x})}{\boldsymbol{B}_{\boldsymbol{x},a}^+ + \boldsymbol{B}_{\boldsymbol{x},a}^-}, \boldsymbol{B}_{\boldsymbol{x},a}^+) \leq T(1, \boldsymbol{B}_{\boldsymbol{x},a}^+)$;

- When $\hat{\beta}(a|\boldsymbol{x}) > \frac{\boldsymbol{B}_{\boldsymbol{x},a}^+ + \boldsymbol{B}_{\boldsymbol{x},a}^-}{2}$, we can get $T_{\mathrm{BIPS}}^* = T(1, \boldsymbol{B}_{\boldsymbol{x},a}^-)$ and $\frac{2\hat{\beta}(a|\boldsymbol{x})}{\boldsymbol{B}_{\boldsymbol{x},a}^+ + \boldsymbol{B}_{\boldsymbol{x},a}^-} > 1$. If $\phi_{\boldsymbol{x},a}^- < \frac{2\hat{\beta}(a|\boldsymbol{x})}{\boldsymbol{B}_{\boldsymbol{x},a}^+ + \boldsymbol{B}_{\boldsymbol{x},a}^-}$, then $T_{\mathrm{UIPS}}^* = T(\phi_{\boldsymbol{x},a}^-, \boldsymbol{B}_{\boldsymbol{x},a}^-) \leq T(1, \boldsymbol{B}_{\boldsymbol{x},a}^-)$, since $\phi_{\boldsymbol{x},a}^-$ is a global minimum. Otherwise when $\phi_{\boldsymbol{x},a}^- \geq \frac{2\hat{\beta}(a|\boldsymbol{x})}{\boldsymbol{B}_{\boldsymbol{x},a}^+ + \boldsymbol{B}_{\boldsymbol{x},a}^-} > 1$, $T_{\mathrm{UIPS}}^* = T(\frac{2\hat{\beta}(a|\boldsymbol{x})}{\boldsymbol{B}_{\boldsymbol{x},a}^+ + \boldsymbol{B}_{\boldsymbol{x},a}^-}, \boldsymbol{B}_{\boldsymbol{x},a}^+) = T(\frac{2\hat{\beta}(a|\boldsymbol{x})}{\boldsymbol{B}_{\boldsymbol{x},a}^+ + \boldsymbol{B}_{\boldsymbol{x},a}^-}, \boldsymbol{B}_{\boldsymbol{x},a}^-) < T(1, \boldsymbol{B}_{\boldsymbol{x},a}^-)$.

In both cases, we have $T_{\mathrm{UIPS}}^* \leq T_{\mathrm{BIPS}}^*$, thus completing the proof.

$\qquad\square$

**Lemma 7.1.** *Under fixed $\pi_{\vartheta}(a|\boldsymbol{x})$ and $\hat{\beta}(a|\boldsymbol{x})$, and $\alpha_{\boldsymbol{x},a} = \sqrt{\frac{\lambda}{2\eta^2} - \frac{\lambda(1-\eta)}{\eta^2}\exp(-2\gamma U_{\boldsymbol{x},a})}$, we have the following observations:*

- *If $\frac{\pi_{\vartheta}(a|\boldsymbol{x})}{\hat{\beta}(a|\boldsymbol{x})} \leq \alpha_{\boldsymbol{x},a}$, $\phi_{\boldsymbol{x},a}^* = 2\eta/\left[\exp(\gamma U_{\boldsymbol{x},a}) + \exp(-\gamma U_{\boldsymbol{x},a})\right]$. Otherwise $\phi_{\boldsymbol{x},a}^* = \lambda/\left[\frac{\lambda}{\eta}\exp\left(-\gamma U_{\boldsymbol{x},a}\right) + \frac{\eta\pi_{\vartheta}(a|\boldsymbol{x})^2}{\hat{\beta}^2(a|\boldsymbol{x})\exp(-\gamma U_{\boldsymbol{x},a})}\right]$. In other words, $\phi_{\boldsymbol{x},a}^* \leq 2\eta$ always holds.*
- *If $\alpha_{\boldsymbol{x},a} \leq \frac{\pi_{\vartheta}(a|\boldsymbol{x})}{\hat{\beta}(a|\boldsymbol{x})}$ and $\frac{\pi_{\vartheta}(a|\boldsymbol{x})}{\boldsymbol{B}_{\boldsymbol{x},a}^-} < \sqrt{\lambda}$, larger $U_{\boldsymbol{x},a}$ brings larger $\phi_{\boldsymbol{x},a}^*$.*
- *Otherwise $\phi_{\boldsymbol{x},a}^*$ decreases as $U_{\boldsymbol{x},a}$ increases.*

*Proof.* The following inequality validates the first observation:

$$\frac{\lambda}{\frac{\lambda}{\eta}\exp\left(-\gamma U_{\boldsymbol{x},a}\right) + \frac{\eta\pi_{\vartheta}(a|\boldsymbol{x})^2}{\hat{\beta}^2(a|\boldsymbol{x})\exp(-\gamma U_{\boldsymbol{x},a})}} \leq \frac{2\eta}{\exp\left(\gamma U_{\boldsymbol{x},a}\right) + \exp\left(-\gamma U_{\boldsymbol{x},a}\right)}$$

For the second and third observations, $\alpha_{\boldsymbol{x},a} \leq \frac{\sqrt{\lambda}}{\sqrt{2}\eta} \leq \frac{\sqrt{\lambda}}{\eta}$. Let $\mathcal{L}(u) = \frac{\lambda}{\eta}\exp(-\gamma u) + \frac{\eta\pi_{\vartheta}(a|\boldsymbol{x})^2}{\hat{\beta}(a|\boldsymbol{x})^2\exp(-\gamma u)}$, we can have:

$$\nabla_u\mathcal{L}(u) = -\gamma\frac{\lambda}{\eta}\exp(-\gamma u) + \gamma\frac{\eta\pi_{\vartheta}(a|\boldsymbol{x})^2}{\hat{\beta}(a|\boldsymbol{x})^2}\exp(\gamma u)$$

To make $\nabla_u\mathcal{L}(u) \geq 0$, we need $u \geq \frac{1}{\gamma}\log\left(\frac{\sqrt{\lambda}\hat{\beta}(a|\boldsymbol{x})}{\eta\pi_{\vartheta}(a|\boldsymbol{x})}\right)$. This implies when $U_{\boldsymbol{x},a} \geq \frac{1}{\gamma}\log\left(\frac{\sqrt{\lambda}\hat{\beta}(a|\boldsymbol{x})}{\eta\pi_{\vartheta}(a|\boldsymbol{x})}\right)$, $\phi_{\boldsymbol{x},a}^*$ will decrease as $U_{\boldsymbol{x},a}$ increases; otherwise as $U_{\boldsymbol{x},a}$ increases, $\phi_{\boldsymbol{x},a}^*$ also increases.

More specifically, when $\alpha_{\boldsymbol{x},a} \leq \frac{\pi_{\vartheta}(a|\boldsymbol{x})}{\hat{\beta}(a|\boldsymbol{x})}$, we have:

$$U_{\boldsymbol{x},a} \leq \frac{1}{\gamma}\log\left(\frac{\sqrt{\lambda}\hat{\beta}(a|\boldsymbol{x})}{\eta\pi_{\vartheta}(a|\boldsymbol{x})}\right) \Leftrightarrow \frac{\pi_{\vartheta}(a|\boldsymbol{x})}{\boldsymbol{B}_{\boldsymbol{x},a}^-} < \sqrt{\lambda} \quad \Leftrightarrow \quad \frac{\pi_{\vartheta}(a|\boldsymbol{x})}{\hat{\beta}(a|\boldsymbol{x})} < \frac{\sqrt{\lambda}}{\eta}\exp(-\gamma U_{\boldsymbol{x},a}).$$

In other words, for these samples, higher uncertainty implies smaller value of $\pi/\hat{\beta}$, and UIPS tends to boost such safe sample with higher $\phi_{\boldsymbol{x},a}^*$.

In other cases, $\phi_{\boldsymbol{x},a}^*$ decreases as $U_{\boldsymbol{x},a}$ increases. This completes the proof. $\square$

### 7.9 Convergence Analysis

Next we provide the convergence analysis of policy improvement under UIPS.

**Definition 7.2.** A function $f : \mathbb{R}^d \to \mathbb{R}$ is $L$-smooth when $\|\nabla f(x) - \nabla f(y)\|_2 \leq L\|x - y\|_2$, for all $x, y \in \mathbb{R}^d$.

We first state and prove a general result, which serves as the basis to complete our convergence analysis of policy improvement under UIPS. The proof is a special case of convergence proof of stochastic gradient descent with biased gradients in [3]. Suppose we have a differentiable function $f : \mathbb{R}^d \to \mathbb{R}$, which is $L$-smooth, and attains a finite minimum value $f^* := \min_{x \in \mathbb{R}^d} f(x)$.

Suppose we cannot directly assess the gradient $\nabla f(x)$. Instead we can only assess a noisy but unbiased gradient $\zeta(x) \in \mathbb{R}^d$ at the given $x$ of the function $\tilde{f}(x)$.

Let $b(x) = \nabla\tilde{f}(x) - \nabla f(x)$ denote the difference between $\nabla\tilde{f}(x)$ and $\nabla f(x)$, and $\delta(x) = \zeta(x) - \nabla\tilde{f}(x)$ denote the noise in gradients. We assume that:

$$\|b(x)\| \leq \varphi\|\nabla f(x)\| \quad \text{and} \quad \mathbb{E}[\delta(x)] = 0 \quad \text{and} \quad \mathbb{E}[\|\delta(x)\|^2|x] \leq M\mathbb{E}[\|\nabla\tilde{f}(x)\|^2] + \sigma^2 \quad (21)$$

where the constants $\varphi$ and $M$ satisfy $0 < \varphi < 1$ and $M \geq 0$ respectively. When running stochastic gradient descent algorithms, i.e. $x_{k+1} = x_k - \eta_k\zeta(x_k)$, we have the following guarantee on the convergence of $x_k$ to an approximate stationary point of $f$.

**Theorem 7.3.** *Suppose $f(\cdot)$ is differentiable and L-smooth, and the assessed approximate gradient meets the conditions in Eq.* (21) *with parameters $(\sigma_k, \varphi_k)$ at iteration $k$. Denote $\sigma_{\max} = \max_k \sigma_k$ and $\varphi_{\max} = \max_k \varphi_k$. Set the stepsizes $\{\eta_k\}$ to $\eta_k = \min\{\frac{1}{(M+1)L}, 1/(\sigma_{\max}\sqrt{K})\}$, after $K$ iterations, the stochastic gradient descent satisfies :*

$$\frac{1}{K}\sum_{k=1}^{K}\mathbb{E}\left[\|\nabla f(x_k)\|^2\right] \leq \frac{2L(f(x_1)-f^*)}{K(1-\varphi_{\max})} + \left(L + \frac{2(f(x_1)-f^*)}{(1-\varphi_{\max})}\right)\frac{\sigma_{\max}}{\sqrt{K}}$$

*Proof.* With $\eta_k \leq \frac{1}{(M+1)L}$, we have:

$$f(x_{k+1}) \leq f(x_k) + \langle \nabla f(x_k), x_{k+1}-x_k\rangle + \frac{L}{2}\|x_{k+1}-x_k\|^2 \tag{22}$$

$$= f(x_k) - \eta_k\langle \nabla f(x_k), \zeta(x_k)\rangle + \frac{L\eta_k^2}{2}\|\zeta(x_k)\|^2$$

$$= f(x_k) - \eta_k\langle \nabla f(x_k), \delta(x_k)+b(x_k)+\nabla f(x_k)\rangle + \frac{L\eta_k^2}{2}\|\delta(x_k)+b(x_k)+\nabla f(x_k)\|^2$$

By taking expectations on both side, we have:

$$\mathbb{E}[f(x_{k+1})] \leq \mathbb{E}[f(x_k)] - \eta_k\mathbb{E}[\langle\nabla f(x_k),\delta(x_k)\rangle] - \eta_k\mathbb{E}[\langle\nabla f(x_k),b(x)+\nabla f(x_k)\rangle] + \frac{L\eta_k^2}{2}\mathbb{E}[\|\delta(x_k)+b(x_k)+\nabla f(x_k)\|^2]$$

$$\overset{(1)}{\leq} \mathbb{E}[f(x_k)] - \eta_k\mathbb{E}[\langle\nabla f(x_k),b(x)+\nabla f(x_k)\rangle] + \frac{L\eta_k^2}{2}\left(\mathbb{E}[\|\delta(x_k)\|^2] + \mathbb{E}[\|b(x)+\nabla f(x_k)\|^2]\right)$$

$$\leq \mathbb{E}[f(x_k)] - \eta_k\mathbb{E}[\langle\nabla f(x_k),b(x)+\nabla f(x_k)\rangle] + \frac{L\eta_k^2}{2}\left((M+1)\mathbb{E}[\|b(x)+\nabla f(x_k)\|^2] + \sigma_k^2\right)$$

$$\overset{(2)}{\leq} \mathbb{E}[f(x_k)] + \frac{\eta_k}{2}\mathbb{E}\left[\left(-2\langle\nabla f(x_k),b(x)+\nabla f(x_k)\rangle + \|b(x)+\nabla f(x_k)\|^2\right)\right] + \frac{L\eta_k^2}{2}\sigma_k^2$$

$$\leq \mathbb{E}[f(x_k)] + \frac{\eta_k}{2}\mathbb{E}\left[\left(-\|\nabla f(x_k)\|^2 + \|b(x)\|^2\right)\right] + \frac{L\eta_k^2}{2}\sigma_k^2$$

$$\overset{(3)}{\leq} \mathbb{E}[f(x_k)] + \frac{\eta_k}{2}(\varphi_k-1)\mathbb{E}[\|\nabla f(x_k)\|^2] + \frac{L\eta_k^2}{2}\sigma_k^2$$

where the inequality labeled as (1) is due to $\mathbb{E}[\delta(x)] = 0$, inequality labeled as (2) is due to $\eta_k \leq \frac{1}{(M+1)L}$, and inequality labeled as (3) is due to $\|b(x_k)\| \leq \varphi_k\|\nabla f(x_k)\|$.

By summing over iterations $k = 1, 2, \ldots, K$ and re-arranging the terms, we obtain:

$$\frac{1}{2}\sum_{k=1}^{K}(1-\varphi_k)\eta_k\mathbb{E}[\|\nabla f(x_k)\|^2] \leq f(x_1) - \mathbb{E}[f(x_{K+1})] + \frac{L}{2}\sum_{k=1}^{K}\eta_k^2\sigma_k^2$$

$$\leq f(x_1) - f^* + \frac{L}{2}\sum_{k=1}^{K}\eta_k^2\sigma_k^2$$

where the last inequality follows from $f(x_{K+1}) \geq f^*$. Since $\eta_k = \min\{\frac{1}{(M+1)L}, \frac{1}{\sigma_{\max}\sqrt{K}}\}, \forall k = 1, \ldots, K$, we can obtain:

$$(1-\varphi_{\max})\eta_1\sum_{k=1}^{K}\mathbb{E}[\|\nabla f(x_k)\|^2] \leq 2(f(x_1)-f^*) + LK\eta_1^2\sigma_{\max}^2$$

Dividing both sides of the above inequality by $K\eta_1(1-\varphi_{\max})$, we obtain the following,

$$\frac{1}{K}\sum_{k=1}^{K}\mathbb{E}[\|\nabla f(x_k)\|^2] \leq \frac{2(f(x_1)-f^*) + LK\eta_1^2\sigma_{\max}^2}{K\eta_1(1-\varphi_{\max})}$$

$$\leq \frac{2(f(x_1)-f^*)}{K(1-\varphi_{\max})}\max\{L, \sigma_{\max}\sqrt{K}\} + L\sigma_{\max}^2\frac{1}{\sigma_{\max}\sqrt{K}}$$

$$\leq \frac{2L(f(x_1)-f^*)}{K(1-\varphi_{\max})} + \left(L + \frac{2(f(x_1)-f^*)}{(1-\varphi_{\max})}\right)\frac{\sigma_{\max}}{\sqrt{K}}$$

$\square$

Given this general result, we can now prove Theorem 3.4 by showing that the gradient in UIPS meets the requirements in Theorem 7.3.

**Proof of Theorem 3.4:**

*Proof.* UIPS aims to maximize the expected return $V(\pi_\vartheta)$. Therefore, we can utilize Theorem 7.3 by setting $f = -V(\pi_\vartheta)$. Since $f^* = -V_{\max}$ and the expected reward is always non-negative, it follows that $f(x_1) - f^* \leq V_{\max}$.

We first introduce some additional notations. Let $\rho_\vartheta^*(\boldsymbol{x}, a) = \frac{\pi_\vartheta(a|\boldsymbol{x})}{\beta^*(a|\boldsymbol{x})}$ denote the propensity score under ground-truth logging policy, and $\hat{\rho}_\vartheta(\boldsymbol{x}, a) = \frac{\pi_\vartheta(a|\boldsymbol{x})}{\hat{\beta}(a|\boldsymbol{x})} \phi_{\boldsymbol{x},a}^*$ represet the propensity score of UIPS. Recall that $\phi_{\boldsymbol{x},a}^*$ is derived through solving the optimization problem in Eq (8). Let $g_\vartheta(\boldsymbol{x}, a) = \frac{\partial \pi_\vartheta(a|\boldsymbol{x})}{\partial \vartheta}$. The true off-policy policy gradient is computed as follows:

$$\nabla V(\pi_\vartheta) = \mathbb{E}_{\beta^*}[\rho^* r g_\vartheta],$$

For UIPS, the approximate policy gradient in each batch with batch size as $B$ is:

$$\nabla \hat{V}_{\text{UIPS}}(\pi_\vartheta) = \frac{1}{B} \sum_{i=1}^{B} \hat{\rho}_i r_i g_\vartheta^i,$$

which is an unbiased estimate of :

$$\nabla V_{\text{UIPS}}(\pi_\vartheta) = \mathbb{E}_{\beta^*}[\hat{\rho} r g_\vartheta].$$

To utilize Theorem 7.3, we set $\nabla f = -\nabla V(\pi_\vartheta)$, $\nabla \tilde{f} = -\nabla V_{\text{UIPS}}(\pi_\vartheta)$, and $\zeta = \nabla \hat{V}_{\text{UIPS}}(\pi_\vartheta)$. We will now demonstrate that the assumptions in Eq. (21) can be satisfied.

Let $\varphi_\vartheta = \max\left\{\left|\frac{\hat{\rho}_\vartheta(\boldsymbol{x},a)}{\rho_\vartheta^*(\boldsymbol{x},a)} - 1\right|\right\}$, We first have:

$$\|b(\vartheta)\| = \|\nabla V_{\text{UIPS}}(\pi_\vartheta) - \nabla V(\pi_\vartheta)\|$$

$$= \|\mathbb{E}_{\beta^*}[(\hat{\rho} - \rho^*) r g_\vartheta]\| = \|\mathbb{E}_{\beta^*}[\rho^*(\frac{\hat{\rho}}{\rho^*} - 1) r g_\vartheta]\|$$

$$\leq \varphi_\vartheta \|\mathbb{E}_{\beta^*}[\rho^* r g_\vartheta]\| \tag{23}$$

And next we show that $0 < \varphi_\vartheta < 1$.

$$\varphi_\vartheta = \max\left\{\left|\frac{\hat{\rho}_\vartheta(\boldsymbol{x}, a)}{\rho_\vartheta^*(\boldsymbol{x}, a)} - 1\right|\right\} = \max\left\{\left|\frac{\beta^*(a|\boldsymbol{x})}{\hat{\beta}(a|\boldsymbol{x})} \cdot \phi_{\boldsymbol{x},a}^* - 1\right|\right\} \tag{24}$$

Recall from Eq.(17) in proof of Theorem 3.2, we have :

$$\phi_{\boldsymbol{x},a}^* = \min\left(\frac{\lambda}{\lambda\frac{\boldsymbol{B}_{\boldsymbol{x},a}^-}{\hat{\beta}(a|\boldsymbol{x})} + \frac{\pi_\vartheta(a|\boldsymbol{x})^2}{\hat{\beta}(a|\boldsymbol{x})\boldsymbol{B}_{\boldsymbol{x},a}^-}}, \frac{2\hat{\beta}(a|\boldsymbol{x})}{\boldsymbol{B}_{\boldsymbol{x},a}^+ + \boldsymbol{B}_{\boldsymbol{x},a}^-}\right)$$

where $\boldsymbol{B}_{\boldsymbol{x},a}^- := \frac{\hat{Z}\exp(-\gamma U_{\boldsymbol{x},a})}{Z^*}\hat{\beta}(a|\boldsymbol{x})$, and $\boldsymbol{B}_{\boldsymbol{x},a}^+ := \frac{\hat{Z}\exp(\gamma U_{\boldsymbol{x},a})}{Z^*}\hat{\beta}(a|\boldsymbol{x})$. Thus we have:

$$\frac{\beta^*(a|\boldsymbol{x})}{\hat{\beta}(a|\boldsymbol{x})} \cdot \phi_{\boldsymbol{x},a}^* = \min\left(\frac{\lambda}{\lambda\frac{\boldsymbol{B}_{\boldsymbol{x},a}^-}{\beta^*(a|\boldsymbol{x})} + \frac{\pi_\vartheta(a|\boldsymbol{x})^2}{\beta^*(a|\boldsymbol{x})\boldsymbol{B}_{\boldsymbol{x},a}^-}}, \frac{2}{\frac{\boldsymbol{B}_{\boldsymbol{x},a}^+}{\beta^*(a|\boldsymbol{x})} + \frac{\boldsymbol{B}_{\boldsymbol{x},a}^-}{\beta^*(a|\boldsymbol{x})}}\right) \tag{25}$$

Since $\boldsymbol{B}_{\boldsymbol{x},a}^+ \geq \beta^*(a|\boldsymbol{x})$, thus $0 \leq \frac{\beta^*(a|\boldsymbol{x})}{\hat{\beta}(a|\boldsymbol{x})} \cdot \phi_{\boldsymbol{x},a}^* \leq 2$, implying $0 < \varphi_\vartheta < 1$.

Also, since $\nabla \hat{V}_{\text{UIPS}}(\pi_\vartheta)$ is an unbiased estimate of $\nabla V_{\text{UIPS}}(\pi_\vartheta)$, we have:

$$\mathbb{E}[\delta(\vartheta)] = \mathbb{E}[\nabla \hat{V}_{\text{UIPS}}(\pi_\vartheta) - \nabla V_{\text{UIPS}}(\pi_\vartheta)] = \boldsymbol{0} \tag{26}$$

Finally, we have the following,

$$\mathbb{E}[\|\delta(\vartheta)\|^2] = \mathbb{E}\left[\|\nabla\hat{V}_{\text{UIPS}}(\pi_\vartheta) - \nabla V_{\text{UIPS}}(\pi_\vartheta)\|^2\right] = \mathbb{E}\left[\left\|\frac{1}{B}\sum_{i=1}^{B}\left(\hat{\rho}_i r_i g_\vartheta^i - \mathbb{E}_{\beta^*}[\hat{\rho}r g_\vartheta]\right)\right\|^2\right]$$

$$\leq \frac{1}{B^2}\mathbb{E}\left[\left(\sum_{i=1}^{B}\|\hat{\rho}_i r_i g_\vartheta^i - E_{\beta^*}[\hat{\rho}r g_\vartheta]\|\right)^2\right] \leq \frac{1}{B}\mathbb{E}\left[\sum_{i=1}^{B}\|\hat{\rho}_i r_i g_\vartheta^i - \mathbb{E}_{\beta^*}[\hat{\rho}r g_\vartheta]\|^2\right]$$

$$\leq \mathbb{E}_{\beta^*}[\|\hat{\rho}r g_\vartheta - \mathbb{E}_{\beta^*}[\hat{\rho}r g_\vartheta]\|^2] \overset{(1)}{=} \mathbb{E}_{\beta^*}[\|\hat{\rho}r g_\vartheta\|^2] - \|\mathbb{E}_{\beta^*}[\hat{\rho}r g_\vartheta]\|^2$$

$$\leq \mathbb{E}_{\beta^*}[\|\hat{\rho}r g_\vartheta\|^2] \leq G_{\max}^2\Phi \tag{27}$$

where the equality labeled as (1) is due to $E[\|Y - E[Y]\|^2] = E[\|Y\|^2] - \|E[Y]\|^2$.

Hence, by applying Theorem 7.3, we have:

$$\frac{1}{K}\sum_{k=1}^{K}\mathbb{E}[\|\nabla V(\pi_{\vartheta_k})\|^2] \leq \frac{2LV_{\max}}{K(1-\varphi_{\max})} + \left(L + \frac{2V_{\max}}{(1-\varphi_{\max})}\right)\frac{G_{\max}\sqrt{\Phi}}{\sqrt{K}} \tag{28}$$

□