# OpenReview forum: "Uncertainty-Aware Instance Reweighting for Off-Policy Learning"
_NeurIPS.cc/2023/Conference — NeurIPS 2023 poster_

### Official Review · Reviewer_19wU · 2023-06-29

**Soundness:** 3 good
**Presentation:** 3 good
**Contribution:** 2 fair
**Rating:** 5
**Confidence:** 4

**Summary:**

This paper delves into the issue of off-policy learning, the objective of which is to devise a new action selection policy based solely on the logged feedback derived from a logging policy. The paper pays particular attention to scenarios in which the logging policy remains unidentified and its estimation proves challenging. Under these conditions, common estimators, like IPS, may lose beneficial attributes such as unbiasedness. To address this complication, the paper introduces a new off-policy learning (OPL) approach called Uncertainty-aware off-policy learning. This new framework aims to optimize the uncertainty-aware objective function by employing a novel weighting scheme that is tuned by minimizing an upper bound of MSE in estimation. A local convergence based on the proposed method is also shown. Experimental results indicate that this proposed framework outperforms a range of benchmark methods on both semi-synthetic and real-world recommendation datasets.

**Strengths:**

- The paper addresses the practically relevant problem of dealing with uncertainty in logging policy estimation in off-policy learning.


- The paper proposes a reasonable and conceptually straightforward method to handle the issue of uncertain logging policies, providing theoretical guarantees regarding estimation and local convergence.


- The paper presents comprehensive experiments, not just basic performance comparisons, but also experiments on off-policy evaluation (OPE) and critical hyperparameters (some of which are included in the appendix).


**Weaknesses:**

- Given that several papers already exist on the topic of distributionally robust off-policy learning (OPL), as discussed in the paper, the formulation of a problem addressing the uncertainty of logging policies may not be groundbreaking, even though I understand that their motivations differ somewhat.


- In the experiments, the issue might also be tackled by simply applying calibration during the estimation of the logging policy, as seen in the following paper:

Aniruddh Raghu, Omer Gottesman, Yao Liu, Matthieu Komorowski, Aldo Faisal, Finale Doshi-Velez, and Emma Brunskill. Behaviour Policy Estimation in Off-Policy Policy Evaluation: Calibration Matters. https://arxiv.org/pdf/1807.01066.pdf


- In most of the experiments, CE performs quite well and is not substantially outperformed by UIPS. Therefore, considering the current experiment results, I may not use UIPS in practice and would rather rely on CE, which is much easier to implement (there is no need to estimate the logging policy when using CE), and does not require the tuning of additional hyperparameters as with UIPS.


- Related to the previous point, in most experiments, the second-best methods for each metric and dataset perform very similarly to UIPS. I am not sure how essential it truly is to address uncertainty in logging policy estimation. I understand that the results are statistically significant, but results can be deemed significant even with a slight performance difference if the sample size is sufficient. In this context, my focus is on the performance difference.


**Questions:**

- How do baseline methods perform in the experiments when they are combined with a calibrated logging policy estimator? Some additional results about this would be useful.


- Could you provide the results relative to the performance of the (true) logging policy? This enables us to see how much improvements the methods bring compared to the logging policy.


- When does UIPS become really crucial? That is, are there any situations where UIPS performs well while all other methods do not work satisfactory. In the current experiments, the second-best methods perform very similarly to UIPS in all datasets and metrics. Moreover, CE performs reasonably and stably for a range of metrics and datasets, which makes it a really good choice in practice indeed.


**Limitations:**

The paper touches on the tightness of the bound as a limitation and future work in the last section.

---

> ### Author Rebuttal · Authors · 2023-08-09
>
> # Reply to  Reviewer 19wU
>
> We thank the reviewer for pointing out the related work on calibration, and for posing valuable questions that have assisted in clarifying crucial arguments.
>
>
> > [Q1] "Difference between UIPS and the line work of distributionally robust off-policy learning (OPL)."
>
> In Section 7.6, we highlighted the distinct differences between UIPS and the line of work on distributionally robust RL in terms of the source of uncertainty, the motivation for utilizing uncertainty, and the techniques employed to handle uncertainties. Furthermore, the experiments conducted in Section 7.6 demonstrate that directly adapting methods from distributionally robust OPL to handle inaccurately estimated logging probabilities leads to bad performance, thus it is not a preferred and suited solution.
>
> > [Q2] "Comparison to Calibration Methods."
>
> The paper [1] found that the accuracy of off-policy evaluation (OPE) strongly depends on the calibration of the estimated logging probabilities. The research specifically highlights that AppoxKNN exhibits the lowest calibration error, leading to the most accurate OPE. However, the paper does not explore how to calibrate the estimated logging policy models for better OPE.
>
> Following the reviewer’s suggestion, we include two new baselines: 1) ApproxKNN following [1]; 2) IPS-C-TS: IPS that combined with calibrated logging probabilities via temperature scaling [1]. We adopted temperature scaling for calibration for two reasons: 1) The logging policy is inherently a probability distribution over actions, and 2) It has been widely acknowledged as one of the most effective calibration methods in multiple classification settings.
>
> The average performance and standard deviations of ApproxKNN, IPS-C-TS and UIPS on both real-world and synthetic datasets are reported in **Table 2 and 3 in the PDF attached in the global response**.
>
> We found that both ApproxKNN and IPS-C-TS generally achieved better performance than BIPS-Cap, implying the effectiveness of calibration. However, UIPS still consistently outperformed both ApproxKNN and IPS-C-TS, particularly on real-world datasets.
>
> The main reason is that calibration primarily focuses on adjusting the predicted probabilities to ensure **on average** the model's predictions are reliable and accurate. In contrast, UIPS specifically handles the impact from each individual sample in policy learning. Moreover, a perfectly calibrated model is clearly beneficial for IPS, but a perfect model for IPS is not necessarily calibrated: a scaled version of ground-truth logging model is well-suited for IPS, but terrible in calibration. Hence, small calibration error could lead to big IPS error and therefore a poorly learnt policy.
>
> [1] Raghu A, Gottesman O, Liu Y, et al. Behaviour Policy Estimation in Off-Policy Policy Evaluation: Calibration Matters[J].
> [2] Guo C, Pleiss G, Sun Y, et al. On calibration of modern neural networks. ICML 2017.
>
>
> > [Q3] "CE performs quite well and is not substantially outperformed by UIPS.  "
>
> We first want to clarify that UIPS consistently exhibits strong performance over CE, particularly on the real-world datasets. The following table shows the relative improvement ratio of UIPS over CE on three real-world datasets regarding Recall@K, Precison@K and NDCG@K.
>
> |                                  | Yahoo | Coat | KuaiRec |
> |----------------------------------|-------|------|---------|
> | improvement ratio on Recall@K    | 1.7%  | 2.8% | 3.6%    |
> | improvement ratio on Precison@K | 1.9%  | 3.0% | 4.2%    |
> | improvement ratio on NDCG@K     | 3.3%  | 1.0% | 4.1%    |
> |                                  |       |      |         |
>
> We can observe that on KuaiRec dataset, characterized by a large action space and sparse interactions (which is common in real-world scenarios), UIPS achieves approximately a 4% improvement over CE in terms of Recall@K, NDCG@K, and Precision@K metrics.
>
> As shown by recent literature [2,3,4]  referred in answer to CQ3, an improvement at this scale is regarded as being significant for our adopted metrics. In particular, an improvement ratio of around 2% in these offline metrics can lead to enhanced online performance for algorithms, resulting in increased GMV /transactions or longer user staytime.
>
> > [Q4] "Performance difference to the best baseline."
>
> Please find our answer to CQ3 in general response to all reviewers.
>
>
> > [Q5] "Performance of the (true) logging policy."
>
> Please find our answer to CQ2 in the general response to all reviewers.
>
>
> > [Q6] "When does UIPS become really crucial? That is, are there any situations where UIPS performs well while all other methods do not work satisfactory."
>
> Thank the reviewer for posing this intriguing question. Empirically, Table 1 and Table 4 demonstrate that UIPS offers distinct advantages in scenarios where the ground-truth logging policy is skewed (indicated by smaller $\tau$ values) or when dealing with larger action spaces and sparse interactions (such as the KuaiRec dataset). This is primarily due to the fact that in such scenarios, the accurate estimation of logging probabilities becomes challenging, amplifying the adverse impact of inaccurate logging policies on overall performance. Considering that in most real-world scenarios, the logging policy tends to be skewed, accompanied by large action spaces and sparse interactions, UIPS holds practical applicability and relevance.
>
> Above empirical observation is also supported by theoretical findings presented in Theorem 3.4.
> This theorem suggests that with small and inaccurate estimated logging probabilities,  particularly when $\beta^*(a|\boldsymbol{x}) \geq 2 \hat{\beta}(a|\boldsymbol{x})$, UIPS can still be guaranteed to converge to a stationary point with the ground-truth policy gradient being zero.
> However, the convergence of BIPS is unknown.

---

> > ### Comment · Reviewer_19wU · 2023-08-11
> >
> > I appreciate the authors' clarifications. Most of my main concerns were addressed nicely. I still think that CE is quite impressive given its simplicity and effectiveness, and thus it might be preferred in practice compared to UIPS (of course the datasets used in the experiments are substantially smaller compared to those of the industry, so it might not be the problem in the empirical analysis.) However, I also acknowledge the importance of studying how well we can do at best under such uncertainty in research and I can increase my score to 5 to indicate that at least I am no longer on the negative side.
> >
> > > Empirically, Table 1 and Table 4 demonstrate that UIPS offers distinct advantages in scenarios where the ground-truth logging policy is skewed (indicated by smaller values) or when dealing with larger action spaces and sparse interactions (such as the KuaiRec dataset).
> >
> > This actually seems to imply, at first glance, that the advantage of UIPS comes from the fact that it unintentionally deals with high variance of typical estimators such as IPS and SNIPS, rather than via dealing with the uncertainty in logging policy estimation. However, the author indeed compared the variance reduction method such as Shrinkage from [Su et al. 20], so at least empirically, dealing with the uncertainty seems to have an additional positive effect on the policy performance. I think this is a very interesting point, and I would be nice to add this discussion in the revision.

---

> > > ### Author Response · Authors · 2023-08-14
> > > **Reply to Official Comment by Reviewer 19wU (Part 1)**
> > >
> > > We highly appreciate the reviewer’s timely feedback and acknowledgement that our responses were helpful in addressing most of the reviewer’s concerns!
> > >
> > > > “I still think that CE is quite impressive given its simplicity and effectiveness, and thus it might be preferred in practice compared to UIPS.”
> > >
> > > We would like to highlight the practical benefits of UIPS over CE with the following additional notes.
> > >
> > > Firstly, we should emphasize the remarkable and statistically significant improvements of UIPS over CE in all our experiments, as discussed in our previous response to reviewer’s question Q3. An improvement at this scale, as indicated by recent literature [1,2], is very likely to lead to a substantial increase in GMV/transactions of the platform, resulting in billions of profits in practical industry applications. As a result, we firmly believe UIPS has great attractiveness and applied value in practice.
> > >
> > > Furthermore, recent work [3] has demonstrated the benefits of off-policy algorithms over CE in an industry recommender system with an action space in the orders of millions. Their findings specifically indicate that directly learning from the logged feedback (i.e., CE) is subject to biases caused by solely observing feedback on logged recommendations, resulting in the 'richer get richer' effect or popularity bias. Off-policy correction methods, such as BIPS in the study, effectively mitigate these biases.
> > >
> > > As an off-policy learning algorithm, UIPS naturally inherits the aforementioned benefits. To verify this, the table below depicts the frequency at which different algorithms tend to recommend the most popular items in the logged training set of the KuaiRec dataset. A higher recommendation ratio signifies that the algorithm is more influenced by the logging/popularity bias present in the training data, and thus amplifying the "rich get richer" phenomenon.
> > >
> > > |                                             | CE | BIPS | POXM | UIPS |
> > > |---------------------------------------------|----|------|------|------|
> > > | recommendation_ratio of top-10% popular items | 0.445  | 0.296    | 0.476  | 0.297    |
> > > | recommendation_ratio of top-20% popular items | 0.749  | 0.617    | 0.748    | 0.585    |
> > >
> > > Recall that POXM is the best baseline on the KuaiRec dataset. While UIPS significantly outperformed BIPS in terms of recommendation accuracy (Table 1 and 4), it is noteworthy that UIPS also exhibits a tendency to recommend less popular items compared to CE, highlighting its effectiveness in mitigating the 'richer get richer' effect while maintaining the quality of recommendation.
> > >
> > > Lastly, UIPS actually does not incur significant computational overhead as discussed in Section 7.1 of the appendix. The estimation of logging policy can also be further simplified, by parameterizing the learning policy and logging policy within one network and learning them in a simultaneous way [3].
> > >
> > >
> > > [1]Zheng et al. "Multi-Objective Personalized Product Retrieval in Taobao Search. KDD2021
> > >
> > > [2]Li et al. Embedding-based product retrieval in taobao search. KDD2021
> > >
> > > [3] Chen et al. Top-k off-policy correction for a REINFORCE recommender system. WSDM 2019.

---

> > > ### Author Response · Authors · 2023-08-14
> > > **Reply to Official Comment by Reviewer 19wU (Part 2)**
> > >
> > > >  “This actually seems to imply, at first glance, that the advantage of UIPS comes from the fact that it unintentionally deals with high variance of typical estimators such as IPS and SNIPS, rather than via dealing with the uncertainty in logging policy estimation. However, the author indeed compared the variance reduction method such as Shrinkage from [Su et al. 20], so at least empirically, dealing with the uncertainty seems to have an additional positive effect on the policy performance. ”
> > >
> > > When using the estimated logging policy,  samples with either **high estimation uncertainty** or **small estimated probability** tend to introduce high variance and high bias, thereby impeding subsequent off-policy learning. This is demonstrated in Proposition 2.1 of our paper. Figure 1 further illustrates that these two factors are usually accompanied, exacerbating their detrimental effects.
> > >
> > > As a result, variance reduction methods, such as Shrinkage from [Su et al. 20], which solely handle small estimated probabilities, cannot handle all situations and thus performed worse than UIPS. In contrast, UIPS effectively handles **both high estimation uncertainty and small estimated probability** through incorporating an uncertainty-aware sample weight to minimize the mean squared error (MSE) of the estimator to its ground-truth value (line 155-161), leading to its strong performance. Again, as we explained in the rebuttal, the design of UIPS is top-down: from the principle of minimizing the MSE of the offline estimated policy value, to estimating the per-sample weights to control impact from samples with high estimation uncertainty and small estimated probability.
> > >
> > > This also explains why UIPS works especially well in scenarios where the ground-truth logging policy is skewed (e.g., smaller $\tau$ value in Table 1) or when dealing with larger action spaces and sparse interactions (e.g., on our real-world datasets). In order to achieve good performance in such situations, effective handling of both high estimation uncertainty and small estimated probability becomes crucial.

---

> > > > ### Comment · Reviewer_19wU · 2023-08-14
> > > >
> > > > Thank you for the further clarifications, which are useful. If I could find an online A/B test result that demonstrates the effectiveness of UIPS in terms of improving the recommendation metrics and GMV in an actual large-scale recommender system, I would be convinced about these points.

---

> > > > > ### Author Response · Authors · 2023-08-14
> > > > >
> > > > > We sincerely appreciate the reviewer's invaluable time and effort in providing invaluable suggestions that have greatly contributed to improving the quality of our paper!
> > > > >
> > > > > Unfortunately, we currently do not have the privilege to run any online experiments with real user populations, which need resources clearly beyond our reach. However, considering that 1) we have thoroughly tested UIPS on several popularly used offline evaluation benchmarks and obtained consistent and encouraging improvements there, and 2) the adopted offline metrics align well with the online performance as evidenced in recent literature, we firmly believe the practical value of UIPS.
> > > > >
> > > > > The publication of UIPS will definitely help us find resources, either through our efforts or potential industry collaborators, to follow the reviewer's suggestions and endeavor to evaluate UIPS via online experiments in future studies. We are very excited about its outlook.

---

### Official Review · Reviewer_YnY8 · 2023-07-05

**Soundness:** 3 good
**Presentation:** 3 good
**Contribution:** 2 fair
**Rating:** 7
**Confidence:** 3

**Summary:**

The paper considers a scenario in off-policy evaluation where we don't have access to the action probabilities of the logging policy, which we need to compute the propensities in IPS. Prior work would estimate these probabilities from data, but would ignore the uncertainties associated with these estimates. In this work the authors propose to re-weight the propensities based on thess uncertainties. The exact form of the weights is derived by minimizing an upper bound on the MSE. The resulting method consistently beats SOTA baselines on both toy and real datasets.

**Strengths:**

A novel IPS variant that is grounded in and backed by theory (i.e. is derived by minimizing the MSE of the estimator), and is designed to solve a concrete problem in existing methods.

Solid experimental methodology, sufficient details on the experimental setup to enable reproducibility.

Strong results: the proposed method beats a broad list of SOTA baselines on both toy and real datasets.

**Weaknesses:**

I have doubts re. the importance of the problem that the method is solving: we can simply log the propensities, right? How common is it in practice that these propensities are not available?

At the same time, I see that in some experiments the method improves upon using GT propensities. This would be a good selling point of the method, but it goes against the initial motivation. How can a method that takes uncertainty into account be better than having _no_ uncertainty? This warrants more discussion in the paper, and makes me wonder if there are alternative explanations to why the method works.

In general, I've found the manuscript to be rather dense and hard to read in places. I would've liked to see more intuition and/or pedagogical examples to understand what the method does exactly. E.g. on line 158: "UIPS assigns them with an increasing weight [..] as uncertainty increases" -- why does it make sense?

The quality of the write-up could be improved: a few grammar mistakes, figure captions could be more informative, some opaque references to prior work could be expanded to make the paper more self-contained.

The related work could be stronger: lines 301 - 320 largely repeat the introduction. What about other methods in on-policy/off-policy that exploit the uncertainties? UCB in bandits, for example?

**Questions:**

How sensitive is the method to the uncertainty estimation approach? Does something simple like MC-Dropout produce similar results?

Also see a few questions in the weaknesses section.

**Limitations:**

Little discussion of limitations: e.g. what about additional cost due to having to estimate uncertainties? Limited societal impact.

---

> ### Author Rebuttal · Authors · 2023-08-09
>
> # Response to  Reviewer YnY8
>
> We thank the reviewer for the positive comments on our work and valuable suggestions. We have clarified several important arguments as outlined below. And we will diligently address other suggestions by further polishing our paper, making the figure captions more informative, and enhancing the references accordingly.
>
> > [Q1] " How common is it in practice that these logging propensities are not available?"
>
> The absence of ground-truth logging probabilities and taking the estimated logging policy for off-policy learning has been a long-standing practice in the off-policy learning literature [1-3].
> There are several reasons that hinder the recording of logging probabilities.
>
> First, in some situations, such as the healthcare domain discussed in [1] or industrial recommender systems as discussed in [2], access to the ground-truth logging policy is not feasible.
>
> Another important reason is due to legacy issues, i.e., the probabilities were not logged when collecting data (e.g., space efficiency was prioritized when designing the logging system). However, even if one is willing to bear the high costs in time and resources to re-collect data with “everything” logged, relying solely on newly-collected data provides only a partial depiction of users' preferences afterwards. Hence, in order to gain a clearer understanding of users, leveraging their historical interaction logs becomes crucial, necessitating the estimation of logging probabilities.
>
> Moreover, in certain cases, there may be security concerns where individuals/companies are unwilling to disclose the logging policy to prevent potential adversarial attacks, while these public resources may hold significant value. As a result, estimating the logging policy is still an important and necessary effort in practice.
>
> [1] Raghu et al. "Behaviour policy estimation in off-policy policy evaluation: Calibration matters." arxiv 2018
>
> [2] Chen et al. Top-k off-policy correction for a REINFORCE recommender system. WSDM 2019.
>
> [3] Strehl et al. Learning from logged implicit exploration data. NeurIPS 2010.
>
>
>
> > [Q2] "Why do UIPS improve upon using GT propensities ?"
>
> Please find our answer to CQ2 in the general response to all reviewers.
>
> > [Q3] "More intuition and/or pedagogical examples to understand what UIPS does exactly."
>
> Please find our answer to CQ1 in the general response to all reviewers. We will make further revisions to enhance the readability of the corresponding section.
>
>
> > [Q4] " What about other methods in on-policy/off-policy that exploit the uncertainties? UCB in bandits, for example?"
>
> In the context of on-policy RL/bandits, the use of uncertainty aims to strike a balance between exploration and exploitation by adopting an optimistic approach (i.e., UCB in bandits). On the other hand, most research on off-policy RL/bandits tends to be more conservative, employing techniques such as Lower Confidence Bounds (LCB) in bandits or penalizing samples with high uncertainty. But those principles are fundamentally different from what we developed in UIPS, which directly minimize the mean square error of off-policy evaluation. The closed-form solution of the resulting per-instance weight in UIPS reflects how uncertainty contributes to the policy evaluation error.
>
> Our UIPS-O and UIPS-P baselines leverage uncertainties using the two aforementioned general principles respectively. However, empirical findings indicate that blindly penalizing or boosting instances based on uncertainty leads to inferior performance compared to UIPS, as they do not directly suggest how uncertainty in the estimated logging probability is related to policy evaluation.
>
> We thank the reviewer once again for your suggestion regarding the related work. We will incorporate the aforementioned discussion to enhance the related work section.
>
> > [Q5] "How sensitive is the method to the uncertainty estimation approach? Does something simple like MC-Dropout produce similar results?"
>
> Our framework is agnostic to the uncertainty estimation methods, as long as the estimated uncertainty is reliable. In the paper, we conducted experiments using the uncertainty estimation framework described in [4] due to its computational efficiency and theoretical soundness. But alternative methods for estimating uncertainties can be readily incorporated into our framework. Inspecting the impact of different uncertainty estimation methods on the quality of the policy evaluation as well as the resulting policy optimization in UIPS is an important and interesting future direction.
>
> [4] Xu P, Wen Z, Zhao H, et al. Neural Contextual Bandits with Deep Representation and Shallow Exploration. ICLR 2021.
>
>
> > [Q6] "what about additional cost due to having to estimate uncertainties?"
>
> The computational complexity of UIPS is discussed in Section 7.1 in appendix. Given the logged dataset containing $N$ samples, $A$ actions and latent dimension $d$, the computational cost of precomputing uncertainties of the logging probabilities is  $O(Nd^2 + d^3)$, where $O(d^3)$ is for matrix inverse and $O(Nd^2)$ is for calculating uncertainties in samples.
>
> Note that calculating logging probability for each sample, which is essential for both UIPS and all IPS-type algorithms, takes $O(NAd)$ time. But considering that the dimension $d$ is typically much smaller than the action size $A$ and sample size $N$, and with precomputed logging probabilities and uncertainties, UIPS can efficiently calculate the sample weight in $O(1)$ time during off-policy learning. Therefore,  we can conclude that  **UIPS does not introduce significant computational overhead compared to the original IPS**.

---

> > ### Comment · Reviewer_YnY8 · 2023-08-15
> >
> > I thank the authors for their response.
> >
> > > relying solely on newly-collected data provides only a partial depiction of users' preferences afterwards
> >
> > Nit: while I understand the additional cost argument, I don't fully follow the "partial depiction" argument: wouldn't newer data provide a more up-to-date depiction of users' preferences?
> >
> > In general, I appreciate the author's arguments for the problem importance, and I think future readers would benefit from a short summary of those in the introduction.
> >
> > Unfortunately, I struggled to follow the intuition for why the relationship between the uncertainty and the sample weight is non-linear, as provided in the common response. Unpacking the argument further and being more rigorous could help. At the same time, the method is theoretically grounded -- while a clear, intuitive interpretation would be useful, I do not consider it to be essential.
> >
> > Overall, the authors have addressed many of my concerns/questions, hence I increase my score.

---

> > > ### Author Response · Authors · 2023-08-16
> > > **# Reply to Official Comment by Reviewer YnY8**
> > >
> > >
> > >
> > > We genuinely appreciate the reviewer's dedicated time, efforts, and invaluable input in responding to our submission and rebuttals. We are delighted to know that our responses effectively addressed most of the reviewer's concerns. Furthermore, we extend our gratitude to the reviewer for raising the recommendation.
> > >
> > > Following the reviewer's suggestion, we will incorporate necessary discussions about the problem's importance into the introduction and expand our current discussion about its derived closed form solution of the per-sample weight in the method section.
> > >
> > > Here are some additional notes on the questions mentioned in the reviewer’s latest comment.
> > >
> > > > “I don't fully follow the "partial depiction" argument: wouldn't newer data provide a more up-to-date depiction of users' preferences?”
> > >
> > > We agree that newer data does have the advantage in providing a more up-to-date depiction of users' preference. However, there are two potential drawbacks in solely relying on newer data. First, it would overlook aspects or patterns that were present in the past but are no longer captured in the recent data. Second, to get a more comprehensive picture, a longer window for data collection is needed, which however slows down model update and system optimization. Hence, effectively leveraging historical data together with any newer data is a more economic and preferred way to understand user preferences in large practical systems [1].
> > >
> > > [1] Pi Q, Zhou G, Zhang Y, et al. Search-based user interest modeling with lifelong sequential behavior data for click-through rate prediction. CIKM 2020.
> > >
> > > > “why the relationship between the uncertainty and the sample weight is non-linear”
> > >
> > > This is also something particularly interesting to us: the closed-form solution for the per-sample weight is rigorously derived based on the minimax optimization problem in Eq (8); and this optimization problem is formulated based on the principle of minimizing MSE of value estimation. Hence, this nonlinearity cannot be manually instructed beforehand. As our empirical study suggested, simply boosting or penalizing samples based on the uncertainty in their estimated logging probabilities did not work out, which further confirmed the validity of our derivation. This motivates us to look further into this new perspective in sample importance in off-policy learning.

---

### Official Review · Reviewer_QiPS · 2023-07-06

**Soundness:** 2 fair
**Presentation:** 2 fair
**Contribution:** 2 fair
**Rating:** 5
**Confidence:** 4

**Summary:**

This paper proposes an Uncertainty-aware Inverse Propensity Score estimator (UIPS) for off-policy learning, taking into account the uncertainty in the estimated logging policy. The authors demonstrate that the commonly used method of estimating the logging policy can lead to biased estimators, particularly for samples with small estimated logging probabilities. UIPS addresses this issue by reweighting the propensity scores based on the uncertainty of the estimated logging policy. The paper provides a theoretical analysis of the convergence properties of UIPS and presents experimental results on synthetic and real-world recommendation datasets, comparing against state-of-the-art baselines.

**Strengths:**

● The paper addresses an important problem in off-policy learning and proposes a novel method, UIPS, to improve the quality of the discovered policy.
● The authors provide a comprehensive theoretical analysis of UIPS, including a convergence guarantee.
● The experimental results demonstrate the effectiveness of UIPS compared to some baselines on both synthetic and real-world datasets.

**Weaknesses:**

● There remain some issues unsolved in the paper, such as the availability of the logging policy. See the questions for details.
● There are some related works that are not mentioned in this paper. In off-policy RL, several papers work on behavior-agnostic instance reweighting [1,2]. They compute the prioritization weight without the need of obtaining a behavior policy. There are also papers that discuss the importance ratio term when applying RL to recommendation systems [3,4].
● Introducing another neural network to estimate $\beta^*$ will increase the system complexity and the computational cost during training and testing. This may hinder the practical application of the algorithm.
● The synthetic dataset and the offline evaluation can give biased evaluation results of the algorithms.

[1] Sinha, Samarth, et al. "Experience replay with likelihood-free importance weights." Learning for Dynamics and Control Conference. PMLR, 2022.
[2] Liu, Xu-Hui, et al. "Regret minimization experience replay in off-policy reinforcement learning." Advances in Neural Information Processing Systems 34 (2021).
[3] Cai, Qingpeng, et al. "Reinforcing User Retention in a Billion Scale Short Video Recommender System." arXiv preprint arXiv:2302.01724 (2023).
[4] Chen, Minmin, et al. "Off-policy actor-critic for recommender systems." Proceedings of the 16th ACM Conference on Recommender Systems. 2022.


**Questions:**

1.How is this paper related to behavior-agnostic methods [1,2,5]? With a GAN-like estimator, these methods no longer reconstruct all those behavior policies. They may also be regarded as baselines to compare with.
2.Why are the probabilities $\beta^*(a|x)$ not recorded in the data? With stochastic logging policies, it is easy to store probabilities together with state and actions when generating data. With deterministic logging policies, a common practice is to sample actions from a Gaussian distribution, with policy output as mean and a certain standard deviation. The action probability will also be available.
3.Why do UIPS-O and UIPS-P lead to poor performance?

[5] Nachum, Ofir, et al. "Dualdice: Behavior-agnostic estimation of discounted stationary distribution corrections." Advances in neural information processing systems 32 (2019).

**Limitations:**

The usage of off-policy correction and uncertainty-based reweighting is limited to policy-based techniques based on the REINFORCE trick. Such techniques can have higher variance than value-based techniques such as TD3 and SAC, and may lead to unstable training.

---

> ### Author Rebuttal · Authors · 2023-08-09
>
> # Response to Reviewer QiPS
> We thank the reviewer for valuable suggestions provided, which help clarify important arguments and enhance the overall quality of the paper.
>
> > [Q1] "Why are the probabilities not recorded in the data?"
>
> The absence of ground-truth logging probabilities and taking the estimated logging policy for off-policy learning have been a long-standing assumption in the off-policy learning literature [6-8]. There are several reasons that hinder the recording of logging probabilities.
>
>  First, in many situations, such as the healthcare domain discussed in [6] or industrial recommender systems as discussed in [7], access to the ground-truth logging policy is not feasible.
>
> Another important reason is due to legacy issues, i.e., the probabilities were not logged when collecting data (e.g., space efficiency was prioritized when designing the users' interaction logging system). However, even if one is willing to bear the high costs in time and resources to re-collect data with “everything” logged, relying solely on newly-collected data provides only a partial depiction of users' preferences afterwards. Hence, in order to gain a clearer understanding of users, leveraging their historical interaction logs becomes crucial, necessitating the estimation of logging probabilities.
>
> Moreover, in certain cases, there may be security concerns where individuals/companies are unwilling to disclose the logging policy to prevent potential adversarial attacks, while these public resources may hold significant value. As a result, estimating the logging policy is still an important and necessary effort in practice.
>
> [6] Raghu et al. "Behaviour policy estimation in off-policy policy evaluation: Calibration matters." arxiv 2018
>
> [7] Chen et al. Top-k off-policy correction for a REINFORCE recommender system. WSDM 2019.
>
> [8] Strehl et al. Learning from logged implicit exploration data. NeurIPS 2010.
>
>
> > [Q2]"Comparison with the behavior-agnostic methods[1,2]".
>
> The main goal of work in [1,2] is to prioritize instances in the replay buffer for better TD learning, rather than accounting for uncertainty in the estimated logging policy for improved off-policy learning. However, we do acknowledge that their proposed solution for directly estimating the propensity ratio has the potential benefit of avoiding estimating the logging policy. To compare its effectiveness, we also included a new baseline called IPS-LFIW, which implements the approach proposed in [1] to directly estimate the propensity ratio for off-policy learning.
>
> The average performance and standard deviations of  IPS-LFIW and UIPS on three synthetic datasets are reported in **Table 1 in the PDF attached in the global response**.
>
> Notably, UIPS consistently outperformed IPS-LFIW with statistically significant improvements.
> One major reason for the worse performance of IPS-LFIW is that it does not consider the accuracy of the estimated propensity ratio, in a direct analogy to failing to handle uncertainty in the estimated logging probabilities in existing IPS-type algorithms.
>
> Furthermore, another advantage of UIPS over the work in [1,2] is that UIPS provides a theoretical guarantee regarding the performance of the learnt policy (Theorem 3.4). In contrast, the behavior-agnostic methods in [1,2] do not offer such a guarantee.
>
>
> > [Q3] "Difference between UIPS and the DICE line of work."
>
> The difference has been discussed in Section 7.5 in appendix. To briefly recap, we demonstrated that in the contextual bandit setting, DualDICE degenerates to the IPS estimator that approximates the unknown ground-truth logging policy with its empirical estimate from the given logged dataset.
>
> Denoting the adapted algorithm from DualDICE as DICE-S, we compared its performance against UIPS in Table 9 and 10 in appendix. We found DICE-S underperformed UIPS significantly in all datasets.
>
> > [Q4] "The relevant papers on RL for recommendation systems [3-4]."
>
> We will incorporate the discussion of them in the related work section. However, it is worth noting that none of the aforementioned works attempt to account for the inaccuracy of the estimated logging policy. In particular, Chen et al. [4] directly used the estimated logging probabilities as the ground-truth in their IPS estimator.
>
> > [Q5] "Introducing another neural network to estimate $\beta^*$ will increase the system complexity and the computational cost during training and testing."
>
> We thank the reviewer for pointing out the place that unfortunately caused misunderstanding.
> For UIPS, both the estimated logging policy and its associated uncertainty can be pre-computed. The uncertainty can be directly estimated using the same model employed for estimating the logging policy (line 162 to 170), with no requirement for an additional neural network.
>
> During training, with the precomputed logging probabilities and uncertainties, UIPS calculates the per-sample weight in O(1) time, and thus it incurs no additional computational cost and does not require an extra network. The evaluation is performed on the learnt policy. Further details regarding the computational cost can be found in Section 7.1.
>
> > [Q6] "The synthetic dataset and the offline evaluation can give biased evaluation results of the algorithms."
>
> While training under non-uniform training datasets (collected under a specific logging policy), all algorithms are evaluated on the **unbiased test dataset**, either from randomized controlled trials (Yahoo & Coat) or a full-observed interaction dataset (KuaiRec, synthetic datasets).
> Consequently, all algorithms are evaluated in an unbiased manner. And such evaluation setting is one of the referred procedures in off-policy learning.
>
> If the reviewer has any further concerns, please let us know. We are more than willing to engage in further discussion.
>
>
> > [Q6] "Why do UIPS-O and UIPS-P lead to poor performance?"
>
> Please find our answer to CQ1 in the general response to all reviewers.

---

> > ### Comment · Reviewer_QiPS · 2023-08-16
> >
> > Thanks for the detailed rebuttal. I am more pleased with the paper, and I wish you luck.

---

> > > ### Author Response · Authors · 2023-08-17
> > > **# Reply to Official Comment by Reviewer QiPS**
> > >
> > > We sincerely thank the reviewer for invaluable time and efforts in handling our submission. We are delighted to learn that our explanations in the rebuttal have addressed the reviewer’s concerns and the reviewer is satisfied with our submission. We are also excited about the theoretical validity and empirical effectiveness of our proposed UIPS algorithm for off-policy learning, and thus are eager to share it with the community.
> > >
> > > Given the rebuttal period is coming to its end, we kindly inquire if there is any additional guidance or request that is necessary for the reviewer to consider increasing the evaluation of our work. Thank you.

---

### Official Review · Reviewer_ha2F · 2023-07-16

**Soundness:** 3 good
**Presentation:** 3 good
**Contribution:** 2 fair
**Rating:** 6
**Confidence:** 3

**Summary:**

This paper proposes UIPS, a method that models the uncertainty of the estimated logging policy to improve off-policy learning. It assigns weights to each observation instance instead of simply dropping those with high uncertainty. The paper deduces the optimal form of weights from minimizing the upper bound of the resulting estimator’s MSE. Then it gets the improved policy by a two-step iterative optimization. This method is evaluated on both synthetic data and three real-world datasets in which UIPS outperforms multiple baselines.

**Strengths:**

The paper addresses an important problem in policy optimization and proposes an innovative way to handle the uncertainty in the logged data. It derives a closed-form solution for the upper bound of MSE and proves the convergence of the method. It also conducts extensive evaluation on synthetic and real-world dataset to demonstrate the effectiveness of UIPS.

**Weaknesses:**

1. For both synthetic and real-world data, although UIPS achieves the best results, it does not outperform the second-best method by a significant extent. It's not clear how this method will be useful in application.
2. The paper lacks a more detailed study and comparison with existing work on handling uncertainty. More explanation is needed on why UIPS is better compared to other methods.
3.  Some deduction is not clear. For example, in-between steps are needed to show the "log trick" used in formula (2). The figure illustration is not very clear either. For example, it's confusing to use ``Log(item freq)" on the X-axis. In Tables 3 and 4, the metrics are not very clear.

**Questions:**

1. How many iterations and how much time does the algorithm run in the evaluation? Does the result verify Theorem 3.4?
2. For Figure 1, the conclusion "items with lower frequencies in the logged dataset have lower estimated logging probabilities" is not a universal trend but only applies for frequency less than 7. How do you explain this observation, and does this affect your other results?

**Limitations:**

The authors are upfront about their limitations and listed future directions to address them.

---

> ### Author Rebuttal · Authors · 2023-08-09
>
> # Reply to  Reviewer ha2F
>
> We appreciate the reviewer's positive feedback, insightful questions, and suggestions for improving the paper. We will incorporate the suggested revisions, including providing a detailed derivation step for the 'log trick' and offering further explanations on the metrics used.
>
> >[Q1] "Performance difference to the best baseline".
>
> Please find our answer to CQ3 in the general response to all reviewers.
>
> >[Q2] "The paper lacks a more detailed study and comparison with existing work on handling uncertainty. "
>
> To the best of our knowledge, our work is the first to explicitly model uncertainty in the estimated logging policy for improved off-policy learning. We have noticed that some work on offline RL also utilizes uncertainties to address the OOD issues or extrapolating errors. But the main idea of those studies is to penalize states/state-action pairs with high uncertainties, which is fundamentally different from the principle in UIPS. Please also refer to our answer in [CQ1].
>
> To understand the impact of these different choices in leveraging uncertainty, we introduced the baseline UIPS-P, which always penalizes samples with high uncertainty, and UIPS-O, which always boosts samples with high uncertainty. The clearly worse performance of UIPS-P and UIPS-O suggests that blindly reweighting through uncertainties is not effective, regardless of the scale of propensity scores.
>
> If the reviewer is aware of any other relevant work that leverages the uncertainty of estimated logging policy for off-policy learning, please let us know. We are more than happy to discuss this further.
>
>
> > [Q3] " Use Log(item freq)" on the X-axis in Figure 1."
>
> The range of values for X is immensely broad in the real-world KuaiRec dataset, spanning multiple orders of magnitude. Consequently, we employ a logarithmic scale to ensure the readability of the figure.
>
> > [Q4] In Tables 3 and 4, the metrics are not very clear.
>
> Let us first briefly recap the motivation behind the experiments in Table 3. UIPS is divided into two steps:
>
> - Derive the optimal instance weight $\phi_{\boldsymbol{x},a}$, so that $\hat{V}\_{\rm UIPS}(\pi_{\boldsymbol{\vartheta}})$ in Eq. (4) approaches the ground-truth $V(\pi_{\boldsymbol{\vartheta}})$ as closely as possible.
> - Update $\pi_{\boldsymbol{\vartheta}}$ by maximizing $\hat{V}\_{\rm UIPS}(\pi_{\boldsymbol{\vartheta}})$.
>
> Thus, Table 3 evaluates the mean square error (MSE) of $\hat{V}_{{\rm UIPS}}$ and the estimators used in the related baselines in approximating the ground-truth $V$. A smaller MSE indicates a higher accuracy. The results reveal that incorporating uncertainty makes the UIPS estimator the most accurate.
>
> In Table 4,  we employed widely adopted offline metrics in recommendation algorithms, namely Recall@K, NDCG@K, and Precision@K, to evaluate the learnt policy $\pi_{\vartheta}$ for recommendation. Their definitions are described in line 233-235, with provided references.
> Due to space constraints, we are unable to present the detailed definitions here. However, we are more than willing to discuss further details during the discussion stage if the reviewer expresses interest.
>
> >[Q5]"How many iterations and how much time does the algorithm run in the evaluation? Does the result verify Theorem 3.4?"
>
> The objective of Theorem 3.4 is to demonstrate the optimality of the learned policy, rather than its convergence rate. The theorem suggests that even without direct access to the true policy gradient (due to the unknown ground-truth logging policy), UIPS converges to a stationary point where the true policy gradient approaches zero.
>
> The computational complexity of UIPS is discussed in Section 7.1 in appendix.  Notably, both the estimated logging policy and its associated uncertainty can be pre-computed, resulting in no additional computational cost during the off-policy learning process of UIPS. Empirically, we also found that UIPS achieved the optimal performance using approximately the same number of epochs as in the BIPS-Cap baseline.
>
> >[Q6] "For Figure 1, the conclusion "items with lower frequencies in the logged dataset have lower estimated logging probabilities" is not a universal trend but only applies for frequency less than 7. How do you explain this observation, and does this affect your other results?"
>
> Proposition 2.1 in the paper suggests that samples with either **high estimation uncertainty** or **small estimated probability** tend to introduce large errors in off-policy learning.  And Figure 1 suggests that these two factors are usually accompanied, exacerbating such errors.
> However, we should also emphasize that UIPS does not have any threshold to define what is small in the estimated logging probability, or assume any monotonic relation among a sample’s estimation uncertainty, estimated logging probability, and observation frequency. The estimated uncertainty is leveraged in Eq (8) for policy learning, and a particular way for uncertainty estimation provided in line 162 to 170. Hence, the reviewer’s observation in Figure 1 does NOT affect the application or performance of UIPS.
>
> On the other hand, there could be many reasons why we did not see a universal trend in Figure 1. For example, in the model's learnt embedding space, some lower frequency items might be similar to some high frequency ones and therefore their estimated logging probabilities are not smaller than those with higher observation frequencies (e.g., around log frequency 7). But again, as long as we can quantify the associated estimation uncertainty, the proposed UIPS solution can be applied.

---

> > ### Comment · Reviewer_ha2F · 2023-08-12
> >
> > While my overall decision of the paper remains  the same, the authors' response is quite sufficient and is appreciated. They more or less address my concerns.

---

> > > ### Author Response · Authors · 2023-08-14
> > > **Reply to Official Comment by Reviewer ha2F**
> > >
> > > Thank you for your timely feedback. We are very glad to find that the reviewer is satisfied with our submission and found our explanations in the rebuttal sufficient. Ourselves are also excited about this work, because of its potential in providing a theoretically justified off-policy learning solution, especially when learning from offline data with no knowledge about the logging policy.
> > >
> > > We are more than happy to know if there is anything that is necessary and may potentially  convince the reviewer to increase the recommendation, which would help us improve both the quality and visibility of this work.

---

### Author Rebuttal · Authors · 2023-08-09

# General Response

We thank all reviewers for their insightful comments and suggestions, which will significantly help us  strengthen our paper. In the following, we will first respond to the common suggestions from all reviewers, and then respond to each reviewer individually.

> [CQ1] "Explain what UIPS exactly does and why do UIPS-O and UIPS-P lead to poor performance?"

UIPS minimizes the mean square error of the estimated value of a learnt policy, via estimating a per-sample weight $\phi\_{\boldsymbol{x},a}$ using bi-level optimization defined in Eq (8). Eq (8) nicely leads to a closed-form of $\phi\_{\boldsymbol{x},a}$ (in Theorem 3.2), with very intuitive and insightful physical meanings (in line 155 to 161):

-  For samples whose largest possible propensity scores are under control: i.e., $ \frac{\pi_{\boldsymbol{\vartheta}}(a|\boldsymbol{x})}{\min
 \boldsymbol{B}_{\boldsymbol{x},a}} < \sqrt{\lambda}$, higher uncertainty in the estimated logging probability implies smaller values of $\pi / \hat{\beta}$ and even smaller values of $\pi(a|\boldsymbol{x})$.
This suggests samples of this type with positive rewards are underestimated, and the degree of underestimation increases with larger uncertainty.
 UIPS thus chooses to increase their weights as uncertainty increases, to emphasize these long-tail positive samples.
-  Conversely, for samples with large propensity scores, UIPS decreases the weights of these samples as the uncertainty increases, so as to prevent their distortion in policy learning.

We should emphasize that these insights were purely extracted by the closed-form solution of $\phi\_{\boldsymbol{x},a}$, rather than manually injected beforehand.

The learning problem induced by UIPS also has a theoretical guarantee on the learnt policy (Theorem 3.4), which suggests that with a high probability UIPS can converge to a stationary point where the ground-truth policy gradient is zero.

In contrast, UIPS-P and UIPS-O blindly penalize or boost samples based on uncertainties, without considering their impact on the accuracy of the resulting estimator and the learnt policy.
As a result, they either overlook long-tail positive samples with high uncertainty or become distorted by samples with high uncertainties. This ultimately leads to their inferior performance.

> [CQ2]"Provide the results relative to the performance of the (true) logging policy, and explain why UIPS improves upon using GT propensities on some datasets."

We have included the IPS-GT baseline in our evaluation on synthetic datasets, which represents the performance of an IPS estimator utilizing the ground-truth logging probabilities.

We can observe from Table 1 that UIPS achieved similar and even better performance than IPS-GT when the ground-truth logging policy is skewed, specifically when $\tau=0.5$ and $\tau=1$. This is because IPS-GT suffers from high variance due to small logging probabilities associated with a skewed ground-truth policy. However, UIPS achieves a better bias-variance trade-off by effectively controlling the negative impact of these high-variance samples.

When the ground-truth logging policy is smoother (e.g., $\tau=2$), the variance of the IPS estimator becomes much smaller, and off-policy correction with the ground-truth logging probabilities leads to better model performance. But UIPS still outperformed all baselines without accessing the ground-truth logging policy.

> [CQ3] “Clarification on the performance.”

To evaluate the significance of improvement, we performed a t-test between the performance of UIPS and the best baseline on all datasets over 10 random trials created by distinct random seeds. Note that the hypothesis testing was conducted at the trial level rather than the instance level.

Table 1 (synthetic datasets) and Table 4 (real-world datasets) demonstrate that **the best baseline varies across datasets** and also **under different metrics**, i.e., no consistent best baselines. In contrast, **UIPS consistently outperformed the best baseline across all metrics with a high level of statistical significance.** This proves the practical generality and applicability of UIPS.

Furthermore, the adopted offline metrics, namely Recall@K, NDCG@K, and Precision@K, have been demonstrated to align well with the online performance of recommendation algorithms [1]. As shown in recent literature [2,3,4], similar improvements on these offline metrics as UIPS has achieved, already suggest enhanced  online performance, such as increase in GMV/transactions or longer staytime.

Finally,  the benefit of UIPS is also been theoretically guaranteed (Theorem 3.4).

[1]Wang X, et al. How well do offline metrics predict online performance of product ranking models? SIGIR2023.

[2]Zheng et al. "Multi-Objective Personalized Product Retrieval in Taobao Search. KDD2021

[3]Li et al. Embedding-based product retrieval in taobao search. KDD2021

[4]Zhang et al. Disentangled Representation for Diversified Recommendations. WSDM2023

---

### Decision · Program_Chairs · 2023-09-21

**Decision:**

Accept (poster)

**Comment:**

The authors proposed to improve the standard IPS estimator via taking the estimation error of the propensity score into consideration. It is achieved by introducing an instance weight to the estimator and optimizing the instance weight to minimize the mean square error of the newly proposed estimator. All of the reviewers found the proposed method sound and easy to follow and hence I recommend an acceptance. However, I still hope the authors can include several more discussions on the related work. For example, I believe for the stochastic bandit case, IPS estimator is minimax optimal, and I hope the authors can make more explicit comparison on the constant terms between the newly proposed estimator and IPS estimator to demonstrate the effectiveness on these simple cases. The discussion on the differences between the proposed estimator and doubly robust estimator is also encouraged.